# XCTFormer: Leveraging Cross-Channel and Cross-Time Dependencies for Enhanced Time-Series Analysis

**Israel Zexer**                                                                              *zexer@post.bgu.ac.il*
*The Stein Faculty of Computer and Information Science*
*Ben-Gurion University of the Negev*

**Omri Azencot**                                                                            *azencot@bgu.ac.il*
*The Stein Faculty of Computer and Information Science*
*Ben-Gurion University of the Negev*

**Reviewed on OpenReview:** *https: // openreview. net/ forum? id= TEfyR4t0Tw*

## Abstract

Multivariate time-series analysis involves extracting informative representations from sequences of multiple interdependent variables, supporting tasks such as forecasting, imputation, and anomaly detection. In real-world scenarios, these variables are typically collected from a shared context or underlying phenomenon, suggesting the presence of latent dependencies across time and channels that can be leveraged to improve performance. However, recent findings show that channel-independent (CI) models, which assume no inter-variable dependencies, often outperform channel-dependent (CD) models that explicitly model such relationships. This surprising result indicates that current CD models may not fully exploit their potential due to limitations in how dependencies are captured. Recent studies have revisited channel dependence modeling with various approaches; however, these methods often employ indirect modeling strategies, which can lead to meaningful dependencies being overlooked. To address this issue, we introduce **XCTFormer**, a transformer-based channel-dependent (CD) model that explicitly captures cross-temporal and cross-channel dependencies via an enhanced attention mechanism. The model operates in a *token-to-token* fashion, modeling pairwise dependencies between every pair of tokens across time and channels. The architecture comprises (i) a data processing module, (ii) a novel Cross-Relational Attention Block (CRAB) that increases capacity and expressiveness, and (iii) an optional Dependency Compression Plugin (DeCoP) that improves scalability. Through extensive experiments on three time-series benchmarks, we show that **XCTFormer** achieves strong results compared to widely recognized baselines; in particular, it attains state-of-the-art performance on the imputation task, outperforming the second-best method by an average of 20.8% in MSE and 15.3% in MAE. Our code is publicly available at `https://github.com/azencot-group/XCTFormer`.

## 1 Introduction

Forecasting, anomaly detection, and imputation are critical tasks across a wide range of real-world domains (Jin et al., 2024). For instance, forecasting is utilized in energy management, weather prediction, healthcare, and more (Mystakidis et al., 2024; Brunet et al., 2023; Duarte et al., 2021). Time-series analysis plays a vital role in extracting key information from sequential data to facilitate these tasks. The effectiveness of this information extraction is crucial, as it directly impacts the performance of subsequent time-series tasks (Trirat et al., 2024). Accurate time-series analysis enables organizations to enhance decision-making and optimize resource allocation (Bui et al., 2018; Wang et al., 2024b), highlighting the importance of the information extraction component as a key area of research.

Time-series data can be modeled using two main approaches (Han et al., 2024). Univariate approaches treat each channel independently, disregarding any potential relationships between them. In contrast,

*multivariate* approaches take into account not only the temporal behavior within each channel but also potential dependencies across channels. In real-world scenarios, multivariate datasets are often derived from a common underlying process or phenomenon, which typically leads to dependencies among the features (Chen et al., 2024). Incorporating relevant signals enhances representation quality and improves accuracy in downstream tasks (Isik et al., 2025; Domingos, 2012). As a result, multivariate models are generally expected to outperform univariate models by leveraging both cross-channel dependencies and temporal dynamics. Therefore, time-series analysis can benefit significantly from richer representations when cross-channel dependencies are utilized.

However, recent work in time-series forecasting by Han et al. (2024) challenged this assumption by showing that channel-independent (CI) models, which treat multivariate time-series as separate univariate channels and ignore potential inter-channel correlations, outperform channel-dependent (CD) models that explicitly model such dependencies. They attribute this surprising outcome to a trade-off between capacity, defined as a model's ability to fit complex patterns, and robustness, defined as its ability to remain accurate in the presence of noise, input variation, or distribution shifts. While CD models gain capacity by incorporating cross-channel information, this often comes at the expense of robustness, making them more sensitive to distribution shifts. In contrast, CI models sacrifice some capacity by ignoring cross-channel dependencies, thereby improving robustness and generalization accuracy. These findings challenge the common belief that adding relevant information typically improves representation quality and accuracy, revealing a gap between channel-dependent methods and their unrealized potential. Motivated by these findings, we seek in this paper to address the following question:

> *How should we model sequential cross-channel information to realize its potential?*

Recent research has revisited channel dependence with cross-channel modeling approaches that often outperform channel-independent (CI) baselines. iTransformer (Liu et al., 2024) targets cross-channel dependencies by treating each channel as a token and applying a Transformer on the token sequence. CrossFormer (Zhang & Yan, 2023) and CARD (Wang et al., 2024c) address both cross-channel and cross-time relationships, by employing a two-stage pipeline for sequence modeling and channel processing. Despite recent advancements, most methods model dependencies across different channels and time *indirectly*, thereby potentially overlooking important interactions. Additionally, cross-channel dependencies are often unknown in advance, as the underlying generative process is typically unknown. These dependencies may also change over time (Zhao & Shen, 2024), raising the need for simultaneous cross-channel cross-time modeling. To address these challenges, we propose a *direct* modeling strategy with a *token-to-token* approach that explicitly captures each token's pairwise dependencies across all channels and time-steps. This potentially minimizes essential information loss associated with existing indirect models. To accomplish this, we introduce **XCTFormer**, a Transformer-based framework that models all pairwise dependencies directly within a single attention block, token-to-token, effectively identifying the most relevant dependencies for downstream tasks.

The backbone of the **XCTFormer** consists of three novel components: (i) a data processing unit, (ii) the Cross-Relational Attention Block (CRAB), and (iii) the Dependency Compression Plugin (DeCoP). First, we independently patch each channel and tokenize the data. Next, we flatten the channel and time dimensions, which allows CRAB and DeCoP to capture all pairwise dependencies in a token-to-token manner. CRAB extends the standard attention block (Vaswani et al., 2017) with two key modifications to improve expressivity and robustness. First, it introduces a learnable, non-boolean masking mechanism that supplements conventional binary masks by weighting dependencies according to their learned importance. This allows the model to focus on the most crucial dependencies for the downstream task. Second, CRAB replaces the standard softmax function with a new normalization technique that retains the properties needed for attention activation (Saratchandran et al., 2025) while allowing negative weights. This extension increases the model's expressiveness by enabling it to capture a wider range of relationships, as suggested by Lv et al. (2024). Lastly, DeCoP is an optional module that partly modifies CRAB with the addition of a learnable compression mechanism, designed to enhance scalability for datasets with numerous channels. It addresses the memory limitations imposed by the transformer's quadratic attention mechanism. DeCoP compresses the quadratic attention into a linear form while minimizing information loss through a learnable compression transformation.

We evaluated XCTFormer against various baseline models on multiple downstream tasks, including forecasting, anomaly detection, and imputation, demonstrating strong results. Our main contributions are:

1. We identify a key limitation in the current literature on time-series modeling: while analysis methods have advanced substantially, little emphasis has been placed on explicitly capturing both cross-channel and cross-time dependencies in a unified manner. Most existing approaches either model temporal patterns or inter-channel relations separately, which restricts their ability to exploit the full structure of multivariate time-series data.

2. To address this gap, we propose XCTFormer, a general-purpose framework that models all pairwise cross-channel and cross-time dependencies directly through token-to-token mappings. XCTFormer integrates two complementary components: (i) CRAB, which enhances expressiveness by learning importance-aware attention masks and allowing signed attention activations, and (ii) DeCoP, which mitigates scalability bottlenecks on high-dimensional data through learnable compression while minimizing information loss.

3. We evaluate our approach on three core time series tasks, forecasting, anomaly detection, and imputation, obtaining consistent improvements or competitive performance against strong baselines. In particular, we achieve state-of-the-art (SoTA) performance in the imputation task, with average reductions in MSE and MAE of 20.8% and 15.3%, respectively. We also observe notable gains in forecasting accuracy and anomaly detection performance. Furthermore, we introduce a synthetic dataset, evaluated on the forecasting task, to evaluate our model on capturing cross-variate and cross-time relationships and to test its robustness to spurious correlations (Appendix D).

## 2 Related Work

**From Classical Methods to Deep Architectures.** Multivariate time-series analysis has progressed from traditional statistical models like ARIMA (Box & Jenkins, 1970), which often struggle to capture nonlinear dynamics, to deep neural approaches such as LSTM (Hochreiter & Schmidhuber, 1997) and TCN (Franceschi et al., 2019). While these deep models improve expressiveness, they may still fall short in modeling very long-range dependencies. More recently, time-series tasks have utilized both simple MLP-based architectures (Zeng et al., 2023; Wang et al., 2024a; Nochumsohn et al., 2025b) and Transformer-based models (Zhou et al., 2021; Nie et al., 2023; Liu et al., 2024; Zhang & Yan, 2023; Wang et al., 2024c). Broadly, these models adopt either a channel-independent (CI) strategy, treating each variable separately, or a channel-dependent (CD) approach that explicitly leverages cross-variable structure.

**Early CD designs: temporal focus with implicit cross-channel modeling.** Early CD models emphasized efficient temporal modeling and attention computation. These methods implicitly incorporated cross-channel information by generating tokens representing all channels at the same or nearby time-steps, typically using 1D convolutions, before applying cross-time attention (Li et al., 2019b; Zhou et al., 2021; Wu et al., 2021; Liu et al., 2022b; Zhou et al., 2022). However, since inter-channel relationships were not explicitly embedded, these approaches failed to fully leverage cross-channel dependencies (Zhang & Yan, 2023). Consequently, the attention mechanism struggled to recover missing structure, leading to suboptimal representations (Liu et al., 2024).

**CI baselines and channel as token formulations.** On the CI side, PatchTST partitions each channel into overlapping time patches, treating these patches as tokens. These channel tokens are then passed to a stacked transformer architecture that exclusively models cross-time dependencies (Nie et al., 2023). Linear models, when applied independently to each channel, have also demonstrated competitive performance (Zeng et al., 2023; Das et al., 2023). MTLinear (Nochumsohn et al., 2025b) is CI: it first clusters channels and then trains a predictor for each cluster to mitigate conflicts in the multi-task objective, but cross-channel dependencies are not explicitly modeled. To reintroduce cross-channel interactions, iTransformer represents each entire channel as a single token, enabling self-attention to operate across variables (Liu et al., 2024). LEDDAM (Yu et al., 2024) takes a different approach by decomposing each series into trend and seasonal

components, processing the seasonal part via parallel cross-channel and cross-time pathways before combining them. However, it still lacks a unified mechanism that jointly models both dimensions within its attention module.

**Two-stage explicit cross-time and cross-channel modeling.** CrossFormer addresses the limitations of earlier models by dividing each channel into equal-length segments and embedding these segments individually to better preserve semantic information (Zhang & Yan, 2023). This approach, along with CARD (Wang et al., 2024c), utilizes a two-stage attention scheme: first attending along the temporal dimension, then explicitly across channels. While this sequential treatment is effective, it captures cross-channel temporal dependencies only *indirectly*, which may result in limited expressiveness.

**Time-series foundation models.** Time-series foundation models (TSFMs) have attracted growing interest as unified architectures for zero-shot and few-shot forecasting across multiple datasets. They are pretrained on diverse time-series corpora to learn general-purpose temporal patterns that transfer across domains. Most TSFMs follow channel-independent designs (Das et al., 2024; Ansari et al., 2024; Shi et al., 2025; Auer et al., 2025; Nochumsohn et al., 2025a), handling multivariate inputs by processing each variable independently as a univariate series. This choice improves scalability and helps pretraining remain broadly applicable across datasets with varying numbers and types of variables. But it may fail to fully leverage cross-variable dependencies that are crucial in many real-world multivariate systems. Recent efforts have begun to address this cross-channel challenge. For example, Chronos 2 (Ansari et al., 2025) introduces group attention to share information within sets of related series, while Moirai-1 (Woo et al., 2024) proposes an any-variate architecture that flattens multivariate time series into a single token sequence, allowing it to handle an arbitrary number of variables and jointly model cross-channel structure. Overall, these works highlight that effectively modeling cross-channel structure remains a key challenge, also within the time-series foundation model paradigm.

## 3 Vanilla Transformer Attention

To facilitate a clear understanding of our proposed modifications, we first outline the standard transformer attention mechanism (Vaswani et al., 2017). Consider an input sequence $X \in \mathbb{R}^{N \times D_i}$, where $N$ denotes the sequence length and $D_i$ the per-token input feature dimension. In our case, the same sequence serves to form queries, keys, and values. The attention block projects $X$ with learnable matrices (with $D_m$ being the per-head attention dimension):

$$W_q, W_k, W_v \in \mathbb{R}^{D_i \times D_m}, \qquad Q = XW_q, \ K = XW_k, \ V = XW_v, \quad Q, K, V \in \mathbb{R}^{N \times D_m}.$$

Scaled dot-product scores quantify pairwise query–key affinity:

$$A = \frac{QK^\top}{\sqrt{D_m}} \in \mathbb{R}^{N \times N}.$$

An optional mask $M \in \mathbb{R}^{N \times N}$ encodes disallowed positions (e.g., padding or future time-steps) via

$$M_{ij} = \begin{cases} 0, & \text{allowed} \\ -\infty, & \text{blocked} \end{cases}$$

We then convert scores into attention weights row-wise and aggregate values accordingly:

$$W = \text{Softmax}(A + M) \in \mathbb{R}^{N \times N}, \qquad O = WV \in \mathbb{R}^{N \times D_m}.$$

Before normalization, the optional mask $M$ is added to the score matrix $A$. Applying a row-wise softmax to $A + M$ effectively assigns zero weight to blocked entries; hence, the mask serves as a selection mechanism that suppresses specific relationships (e.g., to prevent information leakage from future time steps). The resulting attention matrix $W$ is nonnegative with each row summing to one, yielding a probability-like distribution over keys for each query. Consequently, the output $O$ is a row-wise weighted combination of the value vectors, governed by these attention weights, representing the attention block's output.

# 4 XCTFormer

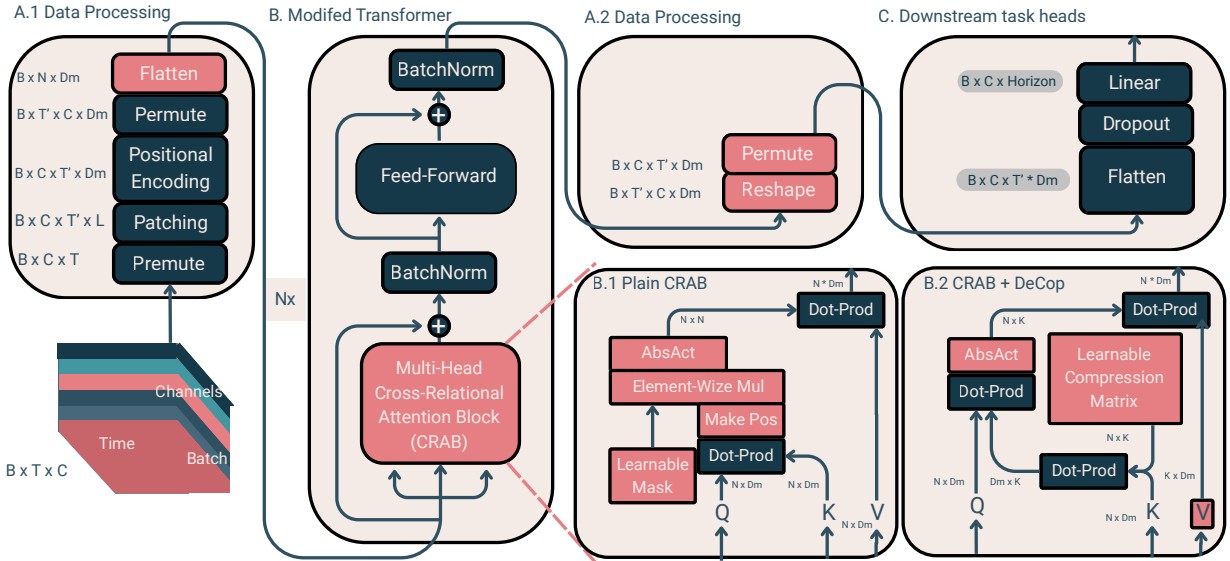

Figure 1: XCTFormer model overview. Multivariate inputs are divided into patches per channel, tokenized, and then passed as a flattened time-and-channel sequence through stacked Transformers with CRAB attention. CRAB utilizes a learnable mask and a signed, non-softmax normalization to model direct token-to-token dependencies. The optional DeCoP module enhances scalability by compressing pairwise attention into a compact representation, reducing memory and compute requirements; since the compressed attention loses the original pairwise structure, the learnable mask is not applied when DeCoP is used. In the diagram, panel B.1 illustrates the CRAB flow and panel B.2 illustrates the CRAB with DeCoP flow.

To model all pairwise dependencies through *direct token-to-token* modeling, we present **XCTFormer**, a Transformer-based, general-purpose, encoder-only time-series model. XCTFormer comprises a universal backbone and a task-specific head. The backbone has three components: (i) a data-processing unit, (ii) a Cross-Relational Attention Block (CRAB), and (iii) a Dependency Compression Plugin (DeCoP). Figure 1 summarizes the pipeline: panel A tokenizes the input and flattens across channels and patches to form a unified sequence that exposes all pairwise dependencies to the Transformer; Panel B applies a stack of our modified Transformer equipped with CRAB (Sec. 4.2) and the optional DeCoP module (Sec. 4.3); Panel C maps the resulting representations to predictions via a task-specific head.

## 4.1 Data Processing

To effectively capture the diverse and unknown dependency structures present in multivariate time-series, XCTFormer is designed to explicitly model all pairwise cross-channel and cross-temporal relationships. For each token, we define potential pairwise dependencies across channels and time points throughout the entire time-series. These dependencies can take one of four potential forms: (i) self-lag relationships, where past values of the same channel may influence future states; (ii) cross-channel synchronous relationships, where channels at identical time points may influence one another; (iii) cross-channel lagged relationships, where other channels may exert temporal influence through their historical values; and (iv) forward-in-time relationships, where current values may influence subsequent values within the same or different channels. For visual representation of these dependencies, see Figure 2.

Modeling dependencies at the level of individual measurements is both computationally expensive and impractical, as single measurements lack meaning without temporal context (Zeng et al., 2023). To address this, we adopt a patching strategy (Nie et al., 2023), segmenting each channel independently into short temporal patches that capture local patterns. We project each patch through a learnable linear layer and add a learnable positional encoding along the time axis for each channel, generating tokens. Finally, we

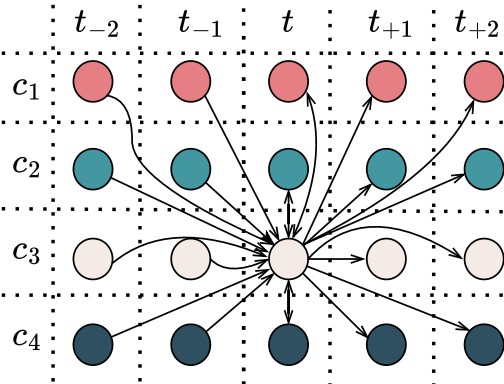

Figure 2: Potential cross-channel and temporal dependencies for token at channel 3 at time $t$.

permute the data dimensions so that the patch sequence comes first, then flatten the tokens across the patch and channel dimensions to create a unified sequence. This enables the Transformer to model all pairwise dependencies (see Panel A, Figure 1). We apply this permutation to simplify the structure of the attention mask, making it easier to analyze further (see App. C.1).

### 4.2 Cross-Relational Attention Block (CRAB)

CRAB modifies the standard attention (Vaswani et al., 2017) block with two complementary components designed to increase models' expressivity: (i) a learnable, non-Boolean relational mask designed to learn important dependencies, and (ii) a signed, absolute-sum normalization activation function that replaces the original softmax, designed to increase expressiveness by allowing negative values, inspired by Lv et al. (2024) findings. For a visual representation of the model, refer to Panel B.1, Fig. 1.

Our **learnable non-Boolean mask** is designed to learn the most dominant dependencies between tokens. We apply this mask to a *positive-transformed* attention score matrix. Starting from the score matrix $A \in \mathbb{R}^{N \times N}$, we remove sign information via a global shift,

$$A_+ = A - \min(A) .$$

Then, we initialize a learnable real-valued mask $M \in \mathbb{R}^{N \times N}$ with zero mean and standard deviation $\sqrt{2/N}$ following He initialization (He et al., 2015). We apply $M$ in an element-wise fashion:

$$A = M \circ A_+ .$$

Thus, this shift removes sign while preserving the *relative ranking* and pairwise differences of the original scores, while the learnable mask $M$ sets their signs and reweighs magnitudes. The produced output is then passed to the activation function.

Our **modified attention activation function** replaces softmax with a row-wise normalization that yields *signed* attention weights. Allowing negative weights can increase the model's expressive power (Lv et al., 2024). To ensure stable training, our proposed activation must preserve the stability property that underlies softmax's success (Saratchandran et al., 2025): maintaining a bounded Frobenius norm of $\sqrt{N}$ for the produced activation matrix,

$$\left\| \text{Activation}(A) \right\|_F \leq \sqrt{N} .$$

For an attention-score matrix $A \in \mathbb{R}^{N \times N}$, we define our activation function as a normalization of values by the absolute sum of the corresponding row. The AbsAct function is defined as:

$$\text{AbsAct}(A_{ij}) = \frac{A_{ij} + \varepsilon}{\sum_{k=1}^{N} |A_{ik} + \varepsilon| + \delta}, \quad \forall i, j \in \{1, \ldots, N\} \tag{1}$$

where $\varepsilon = 1 \times 10^{-4}$ and $\delta = 1 \times 10^{-8}$ are numerical stabilizers. The parameter $\varepsilon$ shifts each attention score before normalization, while $\delta$ adds a positive margin to ensure the denominator remains non-zero. Our activation function satisfies the bounded-norm constraint, ensuring stable training (see Appendix A.1 for a formal proof). Additionally, allowing negative attention weights enables the model to capture a wider range of dependencies, thereby increasing its expressiveness (see Appendix C.4 for additional analysis). Note that since AbsAct divides each entry by its row's absolute-value sum, the standard $1/\sqrt{d_k}$ scaling of attention scores cancels out and is therefore omitted (see Appendix A.2 for a formal derivation).

### 4.3 Dependency Compression Plugin (DeCoP)

Since we model all pairwise cross-channel and cross-time dependencies (Figure 2) using the attention mechanism, memory and compute scale as $\mathcal{O}(N^2)$, where $N$ is the total number of modeled relations. Applying such attention to datasets with many channels can exceed hardware capacity, potentially leading to running failures. To address this limitation, we introduce *DeCoP*, a plugin which compresses each row of the attention matrix $A \in \mathbb{R}^{N \times N}$ with a learnable transformation, yielding a compressed matrix $A_c \in \mathbb{R}^{N \times k}$ with $k \ll N$. This reduces the dominant cost from quadratic to linear in $N$ for fixed $k$, improving scalability regardless of dataset size. For a visual plot of the model, we refer to Panel B.2, Fig. 1. DeCoP is defined as follows:

DeCoP introduces a learnable compression transform whose parameters are initialized with He initialization (He et al., 2015) and fine-tuned during training. Our compressor is a matrix, $C \in \mathbb{R}^{N \times k}$ such that $k \ll N$. Let $Q = XW_q$, $K = XW_k$ with $Q, K \in \mathbb{R}^{N \times D_m}$. We utilize $C$ in the attention computation as follows:

$$A_c = Q\left(K^\top C\right) \in \mathbb{R}^{N \times k} \ .$$

Due to the associative property of matrix multiplication, computing $Q(K^\top C)$ is equivalent to computing $(QK^\top)C$, enabling us to obtain a compressed version of the full token-to-token attention without materializing the quadratic $N \times N$ attention matrix. While vanilla attention incurs $O(N^2 D_m)$ cost for computing $QK^\top$, DeCoP's reordered computation achieves $O(ND_m k)$ complexity, scaling linearly in $N$ when $k \ll N$ while preserving essential attention relationships through the compressed representation. For a full complexity analysis, see Appendix A.3.

Finally, we also need to modify $V$ calculations as the attention dimension is reduced to $k$. The modification is defined as follows:

$$W_v \in \mathbb{R}^{k \times N} \ , \qquad V = W_v X, \quad V \in \mathbb{R}^{k \times D_m} \ .$$

The new $V$ represents the values corresponding to the compressed attention dependencies. The final output is calculated as follows:

$$W_c = \text{AbsAct}(A_c) \in \mathbb{R}^{N \times k} \ , \qquad O = W_c V \in \mathbb{R}^{N \times D_m} \ .$$

Note that when DeCoP is applied, the learnable mask $M$ from CRAB (Sec. 4.2) is not used. Since the compressed attention $A_c \in \mathbb{R}^{N \times k}$ does not preserve the direct pairwise structure of the full $N \times N$ matrix, element-wise masking of individual token-to-token relationships is no longer applicable; instead, AbsAct is applied directly to the compressed scores $A_c$.

## 5 Experiments

We evaluate the proposed XCTFormer across three fundamental time-series tasks: long-term forecasting, imputation, and anomaly detection. Our experiments use well-established benchmark datasets commonly used in prior work to ensure a fair comparison with existing approaches. Across all experiments, we apply

DeCoP (Sec. 4.3) to datasets with more than 60 channels; datasets with 60 or fewer channels use the plain CRAB module without DeCoP (Sec. 4.2). For each task, we present the experimental setup and datasets and report comparative results against strong baselines. The following subsections detail our experiments for each time-series task. App. B provides formal task formulations, extended training and evaluation details, including hyperparameter search protocol and values.

## 5.1 Long-Term Forecasting

Time-series forecasting aims to predict future values from historical observations. We evaluate our model on seven widely used multivariate datasets, comprising four ETT subsets (ETTm1, ETTm2, ETTh1, ETTh2), Weather, Electricity (ECL), and Traffic, following Autoformer (Wu et al., 2021). We adopt the TimesNet setup (Wu et al., 2023) with a lookback window of 96 time-steps and forecasting horizons $\{96, 192, 336, 720\}$. We compare against twelve widely recognized forecasting models: (i) Transformer-based: Autoformer (Wu et al., 2021), FEDformer (Zhou et al., 2022), Crossformer (Zhang & Yan, 2023), PatchTST (Nie et al., 2023), iTransformer (Liu et al., 2024); (ii) Linear/MLP-based: DLinear (Zeng et al., 2023), TiDE (Das et al., 2023), TimeMixer (Wang et al., 2024a), MTLinear (Nochumsohn et al., 2025b); (iii) Hybrid Transformer and Linear: LeDDAM (Yu et al., 2024); (iv) TCN-based: SCINet (Liu et al., 2022a), TimesNet (Wu et al., 2023). We also include a synthetic dataset whose target channel is constructed from lagged cross-variate signals. The dataset also includes distractor channels (random walks) to test robustness to spurious correlations. Models are evaluated in the multivariate-to-single (MS) setting, which allows to specifically test each model's capacity to capture cross-channel and cross-time dependencies. To ensure a fair comparison, all models are trained using the same hyperparameters as ETTm1 (Appendix D).

Table 1: Average long-term forecasting results comparison. Synthetic$^{\dagger}$ is evaluated under multivariate-to-single (MS) setting while all other datasets use multivariate (M). We compare extensive competitive models under different prediction lengths. *Avg* is averaged from all four prediction lengths, that $\{96, 192, 336, 720\}$.

| Models | XCTFormer (Ours) | | TimeMixer++ (ICLR 2025) | | MTLinear (AISTATS 2025) | | Leddam (ICML 2024) | | TimeMixer (ICLR 2024) | | iTransformer (ICLR 2024) | | PatchTST (ICLR 2023) | | Crossformer (ICLR 2023) | | TiDE (TMLR 2023) | | TimesNet (ICLR 2023) | | DLinear (AAAI 2023) | | SCINet (NeurIPS 2022) | | FEDformer (ICML 2022) | | Autoformer (NeurIPS 2021) | |
|---|---|---|---|---|---|---|---|---|---|---|---|---|---|---|---|---|---|---|---|---|---|---|---|---|---|---|---|---|
| Metric | MSE | MAE | MSE | MAE | MSE | MAE | MSE | MAE | MSE | MAE | MSE | MAE | MSE | MAE | MSE | MAE | MSE | MAE | MSE | MAE | MSE | MAE | MSE | MAE | MSE | MAE | MSE | MAE |
| ETT(Avg) | **0.364** | **0.386** | **0.348** | **0.377** | 0.373 | 0.387 | 0.367 | 0.387 | 0.367 | 0.389 | 0.384 | 0.404 | 0.397 | 0.406 | 0.685 | 0.578 | 0.482 | 0.470 | 0.391 | 0.404 | 0.446 | 0.447 | 0.689 | 0.597 | 0.421 | 0.433 | 0.465 | 0.459 |
| Weather | **0.237** | **0.267** | **0.226** | **0.262** | 0.238 | 0.276 | 0.242 | 0.272 | 0.240 | 0.272 | 0.258 | 0.278 | 0.265 | 0.285 | 0.264 | 0.320 | 0.270 | 0.320 | 0.259 | 0.286 | 0.265 | 0.315 | 0.292 | 0.363 | 0.309 | 0.360 | 0.338 | 0.382 |
| ECL$^P$ | **0.166** | **0.263** | **0.165** | **0.253** | 0.198 | 0.283 | 0.168 | 0.263 | 0.182 | 0.273 | 0.178 | 0.270 | 0.216 | 0.318 | 0.244 | 0.334 | 0.252 | 0.344 | 0.193 | 0.304 | 0.225 | 0.319 | 0.268 | 0.365 | 0.213 | 0.327 | 0.227 | 0.364 |
| Traffic$^P$ | 0.435 | 0.287 | **0.416** | **0.264** | 0.621 | 0.372 | 0.467 | 0.294 | 0.485 | 0.297 | **0.428** | **0.282** | 0.529 | 0.341 | 0.667 | 0.426 | 0.760 | 0.473 | 0.620 | 0.336 | 0.625 | 0.383 | 0.804 | 0.509 | 0.609 | 0.376 | 0.628 | 0.379 |
| Synthetic$^{\dagger}$ | **0.040** | **0.155** | – | – | 1.099 | 0.835 | 0.079 | 0.211 | 0.070 | 0.197 | 0.254 | 0.390 | 0.277 | 0.408 | **0.025** | **0.124** | 1.196 | 0.893 | 0.445 | 0.523 | 0.480 | 0.559 | 0.097 | 0.239 | 2.476 | 1.265 | 1.419 | 0.949 |
| 1$^{st}$ Count | 0 | 0 | 4 | 4 | 0 | 0 | 0 | 0 | 0 | 0 | 0 | 0 | 0 | 0 | 1 | 1 | 0 | 0 | 0 | 0 | 0 | 0 | 0 | 0 | 0 | 0 | 0 | 0 |
| Avg FLOPs | 3.33E+09 | | – | | 7.59E+06 | | 8.64E+08 | | 8.72E+08 | | 1.34E+09 | | 8.28E+08 | | 2.31E+09 | | 1.31E+09 | | 1.23E+11 | | 1.01E+07 | | 1.41E+09 | | 2.01E+09 | | 5.26E+07 | |
| Avg Params | 5.08E+06 | | – | | 7.71E+05 | | 2.85E+06 | | 1.34E+05 | | 2.30E+06 | | 7.10E+05 | | 7.87E+06 | | 8.53E+06 | | 5.74E+07 | | 6.50E+04 | | 7.14E+07 | | 1.69E+07 | | 2.15E+05 | |

$^{\dagger}$ Evaluated under multivariate-to-single (MS) setting; hyperparameters chosen identical to ETTm1 for all models. See Appendix D for dataset construction details.
$^1$ Reported MTLinear results reflect the per-dataset best of MTNLinear and MTDLinear.
$^P$ DeCoP was enabled for XCTFormer on this dataset.

**Results.** As shown in Table 1, XCTFormer delivers strong forecasting results compared to widely recognized baselines across diverse benchmarks, achieving second-best performance on 8 out of 10 evaluation metrics. These results highlight the model's effectiveness in capturing cross-channel dependencies in the long-term forecasting task. On the synthetic dataset, where the target is generated from lagged cross-variate signals, XCTFormer ranks second (Avg MSE = 0.040) while using $55\times$ fewer parameters and $364\times$ fewer FLOPs than the top-ranked model. These results suggest that XCTFormer models cross-channel and temporal dependencies effectively in this setting, while being less affected by distractor channels, indicating robustness to spurious correlations. Full results and additional details are provided in Appendix D.

## 5.2 Imputation

Time-series imputation reconstructs missing values from observed data. We evaluate our model on six widely used multivariate datasets: four ETT subsets (ETTm1, ETTm2, ETTh1, ETTh2) (Zhou et al., 2021), Electricity (ECL), and Weather. We adopt the TimeMixer++ setup, using a lookback window of 1024 time steps and applying random missing-mask rates of $\{12.5\%, 25\%, 37.5\%, 50\%\}$. We compare against eleven

Table 2: To evaluate our model performance on imputation, we randomly mask $\{12.5\%, 25\%, 37.5\%, 50\%\}$ of the time points in a time series of length 1024. The final results are averaged across these 4 different masking ratios. Red indicates best performance (lowest error), blue indicates second best.

| Models | XCTFormer (Ours) | | TimeMixer++ (ICLR 2025) | | TimeMixer (ICLR 2024) | | iTransformer (ICLR 2024) | | PatchTST (ICLR 2023) | | Crossformer (ICLR 2023) | | FEDformer (ICML 2022) | | TIDE (TMLR 2023) | | DLinear (AAAI 2023) | | TimesNet (ICLR 2023) | | MICN (ICLR 2023) | | Autoformer (NeurIPS 2021) | |
|---|---|---|---|---|---|---|---|---|---|---|---|---|---|---|---|---|---|---|---|---|---|---|---|---|
| Metric | MSE | MAE | MSE | MAE | MSE | MAE | MSE | MAE | MSE | MAE | MSE | MAE | MSE | MAE | MSE | MAE | MSE | MAE | MSE | MAE | MSE | MAE | MSE | MAE |
| ETT(Avg) | 0.050 | 0.141 | 0.055 | 0.154 | 0.097 | 0.220 | 0.096 | 0.205 | 0.120 | 0.225 | 0.150 | 0.258 | 0.124 | 0.230 | 0.314 | 0.366 | 0.115 | 0.229 | 0.079 | 0.182 | 0.119 | 0.234 | 0.104 | 0.215 |
| Weather | 0.032 | 0.049 | 0.049 | 0.078 | 0.091 | 0.114 | 0.095 | 0.102 | 0.082 | 0.149 | 0.150 | 0.111 | 0.064 | 0.139 | 0.063 | 0.131 | 0.071 | 0.107 | 0.061 | 0.098 | 0.075 | 0.126 | 0.066 | 0.107 |
| ECL[p] | 0.046 | 0.140 | 0.109 | 0.197 | 0.142 | 0.261 | 0.140 | 0.223 | 0.129 | 0.198 | 0.125 | 0.204 | 0.181 | 0.314 | 0.182 | 0.202 | 0.080 | 0.200 | 0.135 | 0.255 | 0.138 | 0.246 | 0.141 | 0.234 |

[p] DeCoP was enabled for XCTFormer on this dataset.

Table 3: F1 scores (%) for anomaly detection across five benchmark datasets. Red indicates best (highest), blue indicates second best.

| Models | XCTFormer (Ours) | TimeMixer++ (ICLR 2025) | iTransformer (ICLR 2024) | TiDE (TMLR 2023) | TimesNet (ICLR 2023) | FEDformer (ICML 2022) | LightTS (AAAI 2022) | ETSformer (ICML 2022) | DLinear (AAAI 2023) | Stationary (NeurIPS 2022) | LSSL (NeurIPS 2022) | Autoformer (NeurIPS 2021) | Pyraformer (ICLR 2022) | Anomaly* (ICLR 2022) | Informer (AAAI 2021) | Reformer (ICLR 2020) | TCN (2018) | LogTrans (NeurIPS 2019) | Transformer (NeurIPS 2017) | LSTM (1997) |
|---|---|---|---|---|---|---|---|---|---|---|---|---|---|---|---|---|---|---|---|---|
| Metric | F1 | F1 | F1 | F1 | F1 | F1 | F1 | F1 | F1 | F1 | F1 | F1 | F1 | F1 | F1 | F1 | F1 | F1 | F1 | F1 |
| SMD | 84.21 | 86.50 | 71.15 | 68.91 | 85.81 | 85.08 | 82.53 | 83.13 | 77.10 | 84.62 | 71.31 | 85.11 | 83.04 | 85.49 | 81.65 | 75.32 | 81.49 | 76.21 | 79.56 | 71.41 |
| MSL | 79.05 | 85.82 | 72.54 | 70.18 | 85.15 | 78.57 | 78.95 | 85.03 | 84.88 | 77.50 | 82.53 | 79.05 | 84.86 | 83.31 | 84.06 | 84.40 | 78.60 | 79.57 | 78.68 | 81.93 |
| SMAP | 86.68 | 73.10 | 66.87 | 64.00 | 71.52 | 70.76 | 69.21 | 69.50 | 69.26 | 71.09 | 66.90 | 71.12 | 71.09 | 71.18 | 69.92 | 70.40 | 70.45 | 69.97 | 69.70 | 70.48 |
| SWaT | 92.60 | 94.64 | 79.18 | 76.73 | 91.74 | 93.19 | 93.33 | 84.91 | 87.52 | 79.88 | 85.76 | 92.74 | 91.78 | 83.10 | 81.43 | 82.80 | 85.09 | 80.52 | 80.37 | 84.34 |
| PSM | 95.30 | 97.60 | 95.17 | 92.50 | 97.47 | 97.23 | 97.15 | 91.76 | 93.55 | 97.29 | 77.20 | 93.29 | 82.08 | 79.40 | 77.10 | 73.61 | 70.57 | 76.74 | 76.07 | 81.67 |
| Avg | 87.57 | 87.47 | 76.98 | 74.46 | 86.34 | 84.97 | 84.23 | 82.87 | 82.46 | 82.08 | 76.74 | 84.26 | 82.57 | 80.50 | 78.83 | 77.31 | 77.24 | 76.60 | 76.88 | 77.97 |

* Anomaly Transformer (Xu et al., 2022); for fair comparison, only reconstruction error is used as the anomaly criterion.

widely recognized models: (i) Transformer-based: Autoformer (Wu et al., 2021), FEDformer (Zhou et al., 2022), Crossformer (Zhang & Yan, 2023), PatchTST (Nie et al., 2023), iTransformer (Liu et al., 2024); (ii) MLP-based: DLinear (Zeng et al., 2023), TiDE (Das et al., 2023), TimeMixer (Wang et al., 2024a); (iii) Convolutional-based: SCINet (Liu et al., 2022a), TimesNet (Wu et al., 2023), MICN (Wang et al., 2023).

**Results.**    As shown in Table 2, XCTFormer delivers state-of-the-art (SoTA) imputation results in comparison to competing baselines across diverse benchmarks. With the best performance on all 6 evaluation metrics, particularly, our approach outperforms the second-best baseline by an average of 20.8% on MSE and 15.3% on MAE across all datasets, highlighting the model's effectiveness in capturing cross-channel dependencies.

## 5.3   Anomaly Detection

Anomaly detection seeks to identify unusual or abnormal patterns in time-series, often corresponding to faults, attacks, or rare operational modes. We evaluate on five widely used benchmarks: SMD (Server Machine Dataset, (Su et al., 2019)), SWaT (Secure Water Treatment, (Mathur & Tippenhauer, 2016)), PSM (Pooled Server Metrics, (Abdulaal et al., 2021)), and NASA telemetry datasets MSL and SMAP (Hundman et al., 2018). We compare against nineteen widely used models: (i) RNN/TCN: LSTM (Hochreiter & Schmidhuber, 1997), TCN (Franceschi et al., 2019); (ii) Transformer-based: Transformer (Vaswani et al., 2017), LogTrans (Li et al., 2019a), Reformer (Kitaev et al., 2020), Informer (Zhou et al., 2021), Pyraformer (Liu et al., 2022b), Autoformer (Wu et al., 2021), FEDformer (Zhou et al., 2022), ETSformer (Woo et al., 2022), Stationary (Non-stationary Transformer) (Liu et al., 2022c), Anomaly Transformer (Xu et al., 2022), LightTS (Zhang et al., 2022), iTransformer (Liu et al., 2024); (iii) State-space: LSSL (Gu et al., 2022); (iv) Linear/MLP: DLinear (Zeng et al., 2023), TiDE (Das et al., 2023); (v) Convolutional/Mixer: TimesNet (Wu et al., 2023), TimeMixer++ (Wang et al., 2025).

**Results.**    As shown in Table 3, XCTFormer performs competitively against the considered strong baselines (for detailed comparison table refer to Appendix E.6). Our model achieves a high $F_1$ score, suggesting it captures cross-channel dependencies effectively for anomaly detection.

# 6 Analysis

**Ablation Study.** To evaluate each component's contribution, we conducted an ablation study across three fundamental time-series tasks: long-term forecasting, imputation, and anomaly detection. Our methodology involved two categories of experiments: (i) *component-wise analysis*, where we systematically removed or altered individual architectural modifications introduced to the vanilla Transformer to isolate each component's impact. Modifications include: removing the learnable mask, reverting the activation function from our proposed approach back to the standard softmax and both; (ii) *dependency modeling analysis*, where we examined variants that model only cross-channel dependencies (inspired by iTransformer (Liu et al., 2024)) or only temporal relationships (similar to PatchTST (Nie et al., 2023)) to validate the necessity of our integrated cross-channel and cross-time modeling strategy. Results presented in Table 4 report the average performance metrics across all datasets and experimental configurations specific to each task for every model variation. The full XCTFormer consistently outperforms all variants across all three tasks and evaluation metrics. These findings validate our architectural design choices and provide empirical evidence that each proposed component contributes meaningfully to the model's overall performance across diverse time-series applications. For detailed experimental configurations and full results, refer to Appendix E.2.

Table 4: Ablation study results across different tasks, evaluated with different XCTFormer variations.

| | Long-term Forecasting | | Imputation | | Anomaly Detection | | | XCTFormer vs Others |
|---|---|---|---|---|---|---|---|---|
| | MSE | MAE | MSE | MAE | Precision | Recall | F-Score | (%) |
| XCTFormer (Original) | **0.328** | **0.337** | **0.044** | **0.124** | **92.1** | **83.7** | **87.6** | - |
| W/o mask[†] | 0.351 | 0.369 | 0.051 | 0.131 | 90.6 | 74.5 | 81.1 | 8.0% |
| Original softmax activation | 0.359 | 0.364 | 0.053 | 0.143 | 90.3 | 75.4 | 81.5 | 9.7% |
| Vanilla transformer | 0.361 | 0.364 | 0.060 | 0.149 | 90.9 | 76.1 | 82.0 | 11.2% |
| Sequence modeling | 0.341 | 0.343 | 0.052 | 0.131 | 91.2 | 78.8 | 83.9 | 5.6% |
| Channel modeling | 0.341 | 0.348 | 0.081 | 0.174 | 91.2 | 76.1 | 82.3 | 14.2% |

[†] W/o mask averages exclude DeCoP datasets (ECL, Traffic) where the learnable mask is not applicable.

**Robustness Across Random Seeds.** To evaluate the stability and reliability of XCTFormer, we assessed its performance across different random initializations. Neural models are often sensitive to parameter initialization randomness and the order of training samples, leading to variability in results. To address this, we trained XCTFormer using the optimal hyperparameters selected by validation on five distinct random seeds (2021 to 2025). For each of the three primary time-series tasks: long-term forecasting, anomaly detection, and imputation, we report both the mean and standard deviation of the relevant performance metric, providing a more robust estimate of model effectiveness. We further quantify robustness using a confidence score, calculated from the coefficient of variation (Reed et al., 2002), which reflects the model's precision and repeatability. In this context, a lower standard deviation indicates greater consistency and, therefore, higher reliability. Summarized seed robustness results are presented in Table 5. For more information on confidence score calculation and the complete analysis tables, refer to Appendix E.3.

Table 5: Averaged metrics of trained models, evaluated on five different seeds (2021–2025) across all datasets, are reported for each metric and time-series task, along with the corresponding confidence interval.

| Task | Metric | Mean $\pm$ Avg. Std | Confidence Score (%) |
|---|---|---|---|
| Long-Term Forecasting | MSE | $0.330 \pm 0.003$ | 99.05% |
| Long-Term Forecasting | MAE | $0.339 \pm 0.002$ | 99.28% |
| Imputation | MSE | $0.047 \pm 0.005$ | 89.14% |
| Imputation | MAE | $0.128 \pm 0.009$ | 93.10% |
| Anomaly Detection | Precision | $91.382 \pm 0.868$ | 99.05% |
| Anomaly Detection | Recall | $79.660 \pm 4.298$ | 94.60% |
| Anomaly Detection | F1 | $84.600 \pm 3.100$ | 96.34% |

## 7 Limitations

XCTFormer explicitly models all pairwise channel-time dependencies via a unified attention block, improving expressiveness and delivering strong performance relative to well-established baselines. However, this design also introduces practical limitations. Flattening time channel tokens makes attention quadratic in the number of tokens, increasing memory and runtime as the lookback length and channel count grow. DeCoP mitigates this cost by compressing attention to a linear form, but it still scales with sequence length and dimensionality and adds a decent parameter overhead. Finally, the gains are not uniform across datasets and tasks, with some settings showing smaller improvements or higher variance, suggesting that the presented pairwise modeling strategy is sensitive to the underlying dependency structure and may offer limited benefits when cross-channel relations are weak or difficult to capture.

## 8 Conclusion

In this paper, we address a fundamental paradox in multivariate time-series analysis: although leveraging cross-channel structure should improve performance, recent findings show that channel-independent models often outperform channel-dependent models. This counterintuitive result suggests that existing channel-dependent methods do not fully exploit cross-channel information. We argue that this limitation arises from current approaches that model cross-channel and cross-time dependencies *indirectly*, thereby overlooking interactions. To bridge this gap, we introduce **XCTFormer**, which revisits channel dependence through direct, token-by-token modeling. Instead of treating channels and time steps as separate entities processed through multi-stage pipelines, XCTFormer treats each channel-time data point as an individual token and models all pairwise dependencies within a unified attention mechanism, which is important for capturing time-evolving dependencies. Through the Cross-Relational Attention Block (CRAB) with learnable masking (when DeCoP is not applied) and an enhanced attention activation function, XCTFormer improves expressivity while maintaining robustness, and the optional Dependency Compression Plugin (DeCoP) supports scalability on high-dimensional datasets. Extensive evaluation across forecasting, anomaly detection, and imputation highlights XCTFormer's effectiveness: it delivers state-of-the-art imputation accuracy, with average error reductions of 20.8% in MSE and 15.3% in MAE, while also achieving strong performance gains in forecasting and anomaly detection. Additional evaluation on a synthetic dataset, where the target depends on lagged cross-variate signals with distractor channels, suggests that XCTFormer effectively captures cross-channel and cross-time dependencies while showing robustness to spurious correlations. At the same time, this direct modeling strategy introduces practical limitations: Unified token-to-token attention scales quadratically with the number of time-channel tokens, and while DeCoP reduces this cost, the parameter count still grows linearly and remains non-negligible. In addition, gains are not uniform across datasets and tasks, with some settings showing smaller improvements or higher variance, suggesting sensitivity to the underlying dependency structure. This motivates further research into more robust channel-time modeling strategies that balance expressiveness, efficiency, and consistency across diverse datasets and tasks. Despite the limitations presented, our proposed direct modeling approach represents a substantial step toward a more comprehensive capture of dependencies and toward realizing the full modeling potential of multivariate time-series data.

### Acknowledgments

This research was partially supported by the Lynn and William Frankel Center of The Stein Faculty of Computer and Information Science, Ben-Gurion University of the Negev, ISF grants 668/21 and 1299/25, an ISF equipment grant, and by the Israeli Council for Higher Education (CHE) via the Data Science Research Center, Ben-Gurion University of the Negev, Israel.

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

# A   Appendix: Extended Notes on XCTFormer

## A.1   Theoretical validity of the proposed activation function

*Proof.* We show that the proposed AbsAct activation function satisfies the sufficient stability criterion of Saratchandran et al. (2025), namely that the Frobenius norm of the produced matrix is bounded by the square root of the number of rows:

$$\|\mathbf{Activation}(A)\|_F \ \leq \ \sqrt{N}$$

Given a matrix $A = (a_{ij}) \in \mathbb{R}^{N \times N}$, and using the activation defined in Section 4, we have:

$$\mathbf{AbsAct}\left( \begin{bmatrix} a_{11} & \cdots & a_{1n} \\ \vdots & \ddots & \vdots \\ a_{n1} & \cdots & a_{nn} \end{bmatrix} \right) = \begin{bmatrix} \dfrac{a_{11}}{\sum_{j=1}^{n}|a_{1j}|} & \cdots & \dfrac{a_{1n}}{\sum_{j=1}^{n}|a_{1j}|} \\ \vdots & \ddots & \vdots \\ \dfrac{a_{n1}}{\sum_{j=1}^{n}|a_{nj}|} & \cdots & \dfrac{a_{nn}}{\sum_{j=1}^{n}|a_{nj}|} \end{bmatrix}. \tag{2}$$

*Note.* In practice, we add a small positive offset and a denominator stabilizer to prevent division by zero. We set $\tilde{a}_{ij} = a_{ij} + 10^{-4}$ element-wise, then normalize each row by $\sum_{j=1}^{n}|\tilde{a}_{ij}| + 10^{-8}$, i.e., use $\tilde{a}_{ij} / \left( \sum_{k=1}^{n} |\tilde{a}_{ik}| + 10^{-8} \right)$. These constants ($10^{-4}$ and $10^{-8}$) are included only for numerical stability and are omitted from the proof for simplicity.

By definition of the Frobenius Norm:

$$\|\mathbf{AbsAct}(A)\|_F^2 = \sum_{i=1}^{N} \sum_{j=1}^{n} \left( \frac{a_{ij}}{\sum_{k=1}^{n}|a_{ik}|} \right)^2 \tag{3}$$

$$= \sum_{i=1}^{N} \frac{\sum_{j=1}^{n} a_{ij}^2}{\left( \sum_{k=1}^{n}|a_{ik}| \right)^2} \tag{4}$$

$$\leq \sum_{i=1}^{N} \frac{\left( \sum_{j=1}^{n}|a_{ij}| \right)^2}{\left( \sum_{k=1}^{n}|a_{ik}| \right)^2} \quad \left( \text{since } \sum_{j} x_j^2 \leq \left( \sum_{j} |x_j| \right)^2 \right) \tag{5}$$

$$= \sum_{i=1}^{N} 1 \ = \ N. \tag{6}$$

Hence, $\|\mathbf{AbsAct}(A)\|_F \leq \sqrt{N}$.
*Note.* The same bound holds when DeCoP is applied: the matrix still has $N$ rows, and after rowwise $\ell_1$ normalization each row's Frobenius norm is strictly less than one. Consequently, the sum of the Frobenius norm across all rows does not exceed $N$.

$\square$

## A.2   Scaling invariance of AbsAct

In standard attention, the score matrix is scaled by $1/\sqrt{d_k}$ before applying softmax to keep dot-product magnitudes controlled and avoid overly sharp attention distributions. Under AbsAct, this scaling is unnecessary. Let $\alpha > 0$ be any positive scalar (e.g. $\alpha = 1/\sqrt{d_k}$). For any row $i$ with $\sum_{k=1}^{N} |A_{ik}| > 0$, we have for every entry $(i, j)$:

$$\text{AbsAct}(\alpha A)_{ij} = \frac{\alpha A_{ij}}{\sum_{k=1}^{N}|\alpha A_{ik}|} = \frac{\alpha A_{ij}}{\alpha \sum_{k=1}^{N}|A_{ik}|} = \frac{A_{ij}}{\sum_{k=1}^{N}|A_{ik}|} = \text{AbsAct}(A)_{ij}. \tag{7}$$

Hence, AbsAct is homogeneous of degree zero for positive scalars:

$$\text{AbsAct}(\alpha A) = \text{AbsAct}(A), \qquad \forall \alpha > 0,$$

whenever the row-wise denominator is nonzero. Because the $1/\sqrt{d_k}$ factor is a positive scalar applied uniformly to every entry, it cancels exactly and can be omitted without affecting the output.

### A.3 DeCoP: Complexity Analysis

We show that DeCoP reduces the attention cost from *quadratic* in the sequence length $N$ to *linear* (for fixed $k \ll N$), thereby lowering both compute and memory costs.

**Setup.** Let $X \in \mathbb{R}^{N \times D_i}$ and $W_q, W_k \in \mathbb{R}^{D_i \times D_m}$. Define

$$Q = XW_q \in \mathbb{R}^{N \times D_m}, \qquad K = XW_k \in \mathbb{R}^{N \times D_m}.$$

DeCoP introduces a learnable compression matrix $C \in \mathbb{R}^{N \times k}$ with $k \ll N$.

**DeCoP computation (no $N \times N$ intermediate).**

$$S = K^\top C \in \mathbb{R}^{D_m \times k}, \qquad A = QS \in \mathbb{R}^{N \times k}.$$

When DeCoP is applied, the learnable non-boolean mask from CRAB (Sec. 4.2) is not used. Since the compressed attention $A_c \in \mathbb{R}^{N \times k}$ does not retain the direct pairwise structure of the full $N \times N$ attention matrix, element-wise masking of individual token-to-token relationships is no longer applicable.

**Total complexity (after).**

$$
\begin{aligned}
\text{Total} &= O(ND_i D_m) && (\text{form } Q) \\
&+ O(ND_i D_m) && (\text{form } K) \\
&+ O(ND_m k) && (S = K^\top C) \\
&+ O(ND_m k) && (A = QS) \\
&= O\big(N[2D_i D_m + 2D_m k]\big) = O\big(2ND_i D_m + 2ND_m k\big) \\
&\overset{(D_i = D_m)}{=} \boxed{O\big(N(D_m^2 + D_m k)\big)}.
\end{aligned}
$$

**Growth in $N$.** For fixed $D_i, D_m, k$ with $k \ll N$, the cost is linear in $N$.

**Memory.** Store $Q, K$: $O(ND_m)$; $A$ and $A_+$: $O(Nk)$; temporary $S$: $O(D_m k)$; parameters $C$: $O(Nk)$, $M$: $O(Nk)$. No $N \times N$ matrix is materialized.

# B  Appendix: Implementation Details

## B.1  Time-Series Tasks Formulation and Metrics

This section presents the mathematical formulation and evaluation metrics for three fundamental time-series tasks: long-term forecasting, anomaly detection, and imputation. Each task addresses distinct challenges in temporal data analysis while sharing common architectural foundations.

### B.1.1  Long-Term Forecasting

Long-term forecasting aims to predict future values of a multivariate time-series given historical observations. Given a historical sequence $\mathbf{X} \in \mathbb{R}^{L \times C}$ where $L$ is the lookback window length and $C$ is the number of channels, the objective is to predict the future sequence $\mathbf{Y} \in \mathbb{R}^{H \times C}$ where $H$ is the prediction horizon.

**Training Objective:** The model minimizes the Mean Squared Error (MSE) loss:

$$\mathcal{L}_{forecast} = \frac{1}{N} \sum_{i=1}^{N} ||\mathbf{Y}_i - \hat{\mathbf{Y}}_i||^2 \tag{8}$$

where $N$ is the number of training samples, $\mathbf{Y}_i$ is the ground truth, and $\hat{\mathbf{Y}}_i$ is the predicted sequence.

**Evaluation Metrics:** Performance is assessed using multiple regression metrics:

- **Mean Absolute Error (MAE):** $MAE = \frac{1}{NH} \sum_{i=1}^{N} \sum_{t=1}^{H} |\mathbf{Y}_{i,t} - \hat{\mathbf{Y}}_{i,t}|$
- **Mean Squared Error (MSE):** $MSE = \frac{1}{NH} \sum_{i=1}^{N} \sum_{t=1}^{H} (\mathbf{Y}_{i,t} - \hat{\mathbf{Y}}_{i,t})^2$

### B.1.2  Imputation

Time-series imputation reconstructs missing values in partially observed sequences. Let $\mathbf{X} \in \mathbb{R}^{L \times C}$ be the ground-truth sequence and $\mathbf{M} \in \{0,1\}^{L \times C}$ a binary mask where $M_{t,c} = 0$ marks a missing entry; the observed input is obtained via an element-wise product $\mathbf{X}^{\text{obs}} = \mathbf{X} \odot \mathbf{M}$.

**Problem Formulation.** An imputation model with parameters $\psi$ reconstructs the complete sequence:

$$\widehat{\mathbf{X}} = \text{Imputer}_\psi\big(\mathbf{X}^{\text{obs}}, \mathbf{M}\big). \tag{9}$$

**Training Objective.** During training, artificial masks are sampled with mask rate $p$. The reconstruction loss is computed *only on masked positions*:

$$\mathcal{L}_{\text{impute}} = \frac{1}{|m|} \sum_{(t,c) \in m} \big(\mathbf{X}_{t,c} - \widehat{\mathbf{X}}_{t,c}\big)^2, \qquad m = \{(t,c) : M_{t,c} = 0\}. \tag{10}$$

**Evaluation Protocol.**

1. Apply a random mask (rate $p$) to test sequences to obtain $\mathbf{X}^{\text{obs}}$ and $\mathbf{M}$.

2. Impute the missing entries: $\mathbf{X}^{\text{filled}} = \mathbf{X}^{\text{obs}} \odot \mathbf{M} + \widehat{\mathbf{X}} \odot (1 - \mathbf{M})$.

3. Compute metrics *exclusively on the masked set $m$*.

**Evaluation Metrics (masked-only).** All metrics are calculated *only on masked values* $(t,c) \in m$:

$$\text{MAE}_{\text{mask}} = \frac{1}{|m|} \sum_{(t,c) \in m} \big|\mathbf{X}_{t,c} - \widehat{\mathbf{X}}_{t,c}\big|, \tag{11}$$

$$\text{MSE}_{\text{mask}} = \frac{1}{|m|} \sum_{(t,c) \in m} \big(\mathbf{X}_{t,c} - \widehat{\mathbf{X}}_{t,c}\big)^2. \tag{12}$$

### B.1.3 Anomaly Detection

Anomaly detection identifies unusual patterns or outliers in time-series data using a reconstruction-based approach. The model learns to reconstruct normal patterns and flags samples with high reconstruction errors as anomalous.

**Problem Formulation:** Given a time-series $\mathbf{X} \in \mathbb{R}^{L \times C}$, the model $g_\phi$ learns to reconstruct the input:

$$\hat{\mathbf{X}} = g_\phi(\mathbf{X}) \tag{13}$$

The anomaly score is computed as the reconstruction error: $s = ||\mathbf{X} - \hat{\mathbf{X}}||^2$

**Training Objective:** The model is trained exclusively on normal data using reconstruction loss:

$$\mathcal{L}_{recon} = \frac{1}{N} \sum_{i=1}^{N} ||\mathbf{X}_i - g_\phi(\mathbf{X}_i)||^2 \tag{14}$$

**Detection Mechanism:** We use a percentile-based threshold over the pooled anomaly-score distribution. Let $\mathcal{S} = \{s_{\text{train}}\} \cup \{s_{\text{test}}\}$. For a target anomaly rate $\alpha$, the threshold is

$$\tau = \text{quantile}(\mathcal{S}, 1 - \alpha).$$

In our experiments we set $\alpha = 0.01$ (1%) for all datasets, except SMD (Su et al., 2019), where $\alpha = 0.005$ (0.5%). A sample with score $s$ is labeled as

$$\text{label} = \begin{cases} 1, & \text{if } s > \tau \text{ (anomaly)}, \\ 0, & \text{if } s \leq \tau \text{ (normal)}. \end{cases}$$

**Evaluation Metrics:** Performance is measured using binary classification metrics:

- **Precision:** $P = \frac{TP}{TP+FP}$ (proportion of correctly identified anomalies)

- **Recall:** $R = \frac{TP}{TP+FN}$ (proportion of actual anomalies detected)

- **F1-Score:** $F_1 = 2 \cdot \frac{P \times R}{P+R}$ (harmonic mean of precision and recall)

- **Accuracy:** $Acc = \frac{TP+TN}{TP+TN+FP+FN}$ (overall classification correctness)

where $TP$, $TN$, $FP$, and $FN$ represent true positives, true negatives, false positives, and false negatives, respectively.

## B.2 Experiment Datasets And Evaluation Setups

We evaluate long-term forecasting on seven widely used multivariate datasets (Weather, Electricity, Traffic, ETTh1, ETTh2, ETTm1, ETTm2). For forecasting, we follow the TimesNet (Wu et al., 2023) setup with a look-back window $L = 96$ and horizons $H \in \{96, 192, 336, 720\}$; dataset specifications appear in Table 6. For time-series imputation, we use the same datasets as forecasting, except for Traffic, and follow the TimeMixer++ (Wang et al., 2025) setup with $L = 1024$ and masking ratios $p \in \{12.5\%, 25\%, 37.5\%, 50\%\}$, refer to Table 7. Anomaly detection focuses on identifying fine-grained patterns. To assess this, we selected the following datasets: SMD (Server Machine Dataset, (Su et al., 2019)), SWaT (Secure Water Treatment, (Mathur & Tippenhauer, 2016)), PSM (Pooled Server Metrics, (Abdulaal et al., 2021)), and NASA telemetry datasets MSL and SMAP (Hundman et al., 2018). The details of the datasets used for anomaly detection are provided in Table 8. For long-term forecasting, we additionally evaluate on a synthetic dataset where the target is derived from lagged cross-channel signals; full construction details appear in Appendix D.

Table 6: Benchmark datasets and evaluation settings for long-term forecasting.

| Dataset | Dim | Look-back | Prediction Horizons | Dataset Size | Frequency | Information |
|---------|-----|-----------|---------------------|--------------|-----------|-------------|
| ETTm1 | 7 | 96 | {96, 192, 336, 720} | (34465, 11521, 11521) | 15 min | Temperature |
| ETTm2 | 7 | 96 | {96, 192, 336, 720} | (34465, 11521, 11521) | 15 min | Temperature |
| ETTh1 | 7 | 96 | {96, 192, 336, 720} | (8545, 2881, 2881) | 15 min | Temperature |
| ETTh2 | 7 | 96 | {96, 192, 336, 720} | (8545, 2881, 2881) | 15 min | Temperature |
| Weather | 21 | 96 | {96, 192, 336, 720} | (36792, 5271, 10540) | 10 min | Weather |
| ECL | 321 | 96 | {96, 192, 336, 720} | (18317, 2633, 5261) | Hourly | Electricity |
| Traffic | 862 | 96 | {96, 192, 336, 720} | (12185, 1757, 3509) | Hourly | Transportation |

Table 7: Benchmark datasets and evaluation settings for time-series imputation.

| Dataset | Dim | Look-back | Imputation Mask Ratios | Dataset Size | Frequency | Information |
|---------|-----|-----------|------------------------|--------------|-----------|-------------|
| ETTm1 | 7 | 1024 | [12.5%, 25%, 37.5%, 50%] | (34465, 11521, 11521) | 15 min | Temperature |
| ETTm2 | 7 | 1024 | [12.5%, 25%, 37.5%, 50%] | (34465, 11521, 11521) | 15 min | Temperature |
| ETTh1 | 7 | 1024 | [12.5%, 25%, 37.5%, 50%] | (8545, 2881, 2881) | 15 min | Temperature |
| ETTh2 | 7 | 1024 | [12.5%, 25%, 37.5%, 50%] | (8545, 2881, 2881) | 15 min | Temperature |
| Weather | 21 | 1024 | [12.5%, 25%, 37.5%, 50%] | (36792, 5271, 10540) | 10 min | Weather |
| ECL | 321 | 1024 | [12.5%, 25%, 37.5%, 50%] | (18317, 2633, 5261) | Hourly | Electricity |

Table 8: Dataset detailed descriptions for anomaly detection. The dataset size is organized in (Train, Validation, Test).

| Dataset | Dim | Series Length | Dataset Size | Information |
|---------|-----|---------------|--------------|-------------|
| SMD | 38 | 100 | (566724, 141681, 708420) | Server machines |
| MSL | 55 | 100 | (44653, 11664, 73729) | Spacecraft telemetry (Mars) |
| SMAP | 25 | 100 | (108146, 27037, 427617) | Spacecraft telemetry |
| SWaT | 51 | 100 | (396000, 99000, 449919) | Water treatment ICS |
| PSM | 25 | 100 | (105984, 26497, 87841) | Server metrics |

## B.3 Training Details.

All experiments were implemented in PyTorch (Paszke et al., 2019) and run on NVIDIA RTX 3090 GPUs. We fix the random seed to 2021. We applied a RevIN transformation (Kim et al., 2022) to mitigate distributional shifts in the data. Models are trained with the Adam optimizer (Kingma & Ba, 2015) using mean squared error (MSE) loss, together with a `OneCycleLR` scheduler. At a high level, `OneCycleLR` first increases the learning rate from its initial value to a peak (while inversely adjusting momentum), and then gradually anneals it to a small value for the remainder of training; this promotes fast early progress and stable late-stage convergence. We set `pct_start = 0.4`, allocating 40% of the total training steps to the warm-up/increase phase and 60% to the annealing phase. For each run, we select the checkpoint with the lowest validation MSE and report the corresponding test performance in the tables.

**Hyper-Parameter Search.** We tuned hyperparameters with Optuna and chose the configuration yielding the lowest validation **MSE**. The selected configuration is tuned per dataset and task, and then kept fixed across all forecasting horizons or imputation mask ratios, as reported in Table 9.

Table 9: Hyperparameter settings for **XCTFormer** per dataset per time-series task

| | Data Processing | | Transformer | | | | | | XCTFormer | | Training | | |
|---|---|---|---|---|---|---|---|---|---|---|---|---|---|
| Dataset | patch_len | stride | e_layers | n_heads | d_model | d_ff | dropout | fc_dropout | attn_dropout | k | batch_size | learning_rate | epochs |
| **Long-term time-series Forecasting** | | | | | | | | | | | | | |
| ETTh1 | 16 | 8 | 1 | 1 | 8 | 16 | 0.2 | 0.3 | 0.6 | - | 32 | 0.001 | 10 |
| ETTh2 | 16 | 8 | 3 | 1 | 30 | 60 | 0.1 | 0.2 | 0.8 | - | 32 | 0.01 | 10 |
| ETTm1 | 16 | 8 | 2 | 4 | 32 | 64 | 0.1 | 0.05 | 0.8 | - | 32 | 0.005 | 10 |
| ETTm2 | 16 | 8 | 1 | 1 | 224 | 448 | 0.1 | 0.05 | 0.8 | - | 32 | 0.005 | 10 |
| Weather | 16 | 8 | 3 | 2 | 248 | 496 | 0.1 | 0.05 | 0.8 | - | 32 | 0.0005 | 10 |
| Traffic | 16 | 8 | 3 | 4 | 248 | 496 | 0.1 | 0.05 | 0.6 | 192 | 8 | 0.001 | 10 |
| ECL | 16 | 8 | 3 | 1 | 248 | 496 | 0.1 | 0.05 | 0.5 | 64 | 32 | 0.005 | 10 |
| **Imputation** | | | | | | | | | | | | | |
| ETTh1 | 16 | 8 | 2 | 1 | 64 | 128 | 0.1 | 0.05 | 0.5 | - | 32 | 0.01 | 10 |
| ETTh2 | 64 | 32 | 3 | 1 | 160 | 320 | 0.1 | 0.05 | 0.3 | - | 32 | 0.005 | 10 |
| ETTm1 | 16 | 8 | 3 | 4 | 96 | 192 | 0.1 | 0.05 | 0.1 | - | 32 | 0.005 | 10 |
| ETTm2 | 16 | 8 | 2 | 1 | 128 | 256 | 0.1 | 0.05 | 0.5 | - | 32 | 0.001 | 10 |
| Weather | 16 | 8 | 3 | 1 | 192 | 384 | 0.1 | 0.05 | 0.8 | - | 32 | 0.001 | 10 |
| ECL | 64 | 32 | 2 | 2 | 192 | 384 | 0.1 | 0.05 | 0.7 | 128 | 32 | 0.005 | 10 |
| **Anomaly Detection** | | | | | | | | | | | | | |
| MSL | 16 | 8 | 2 | 4 | 256 | 512 | 0.1 | 0.05 | 0.7 | - | 128 | 0.01 | 10 |
| PSM | 16 | 8 | 2 | 1 | 256 | 512 | 0.1 | 0.05 | 0.8 | - | 128 | 0.001 | 10 |
| SMAP | 16 | 8 | 3 | 1 | 256 | 128 | 0.1 | 0.05 | 0.3 | - | 128 | 0.005 | 10 |
| SMD | 16 | 8 | 2 | 1 | 168 | 336 | 0.1 | 0.05 | 0.3 | - | 128 | 0.001 | 10 |
| SWaT | 16 | 8 | 1 | 2 | 216 | 432 | 0.1 | 0.05 | 0.4 | - | 128 | 0.0005 | 10 |

**Hyper-Parameter Search Space.** We searched over a bounded hyperparameter space per dataset and task, while fixing a few coupled settings to reduce degrees of freedom (we set stride = patch_len/2 and d_ff = 2 × d_model). For ETTh1/ETTh2 in forecasting, where we observed stronger overfitting, we narrowed the d_model range and explicitly tuned dropout to improve generalization. The search area is represented in Table 10.

Table 10: Hyperparameter Search Space

| Task | Datasets | lr | att_dropout | n_heads | e_layers | d_model | patch_len | dropout | fc_dropout | k |
|---|---|---|---|---|---|---|---|---|---|---|
| Long-term Forecast | ETTh1, ETTh2 | $\{5e\text{-}4, 1e\text{-}3,$ | $[0.1, 0.8]_{0.1}$ | $\{1, 2, 4\}$ | $[1, 3]$ | $[4, 64]_2$ | – | $[0.1, 0.3]_{0.05}$ | $[0.05, 0.3]_{0.05}$ | – |
| | Others | $5e\text{-}3, 1e\text{-}2\}$ | | | | $[8, 256]_8$ | – | – | – | $[64, 256]^{\dagger}_{64}$ |
| Imputation | All | | | | | $[32, 256]_{32}$ | $\{16, 64, 128\}$ | – | – | $[64, 256]^{\dagger}_{64}$ |
| Anomaly Detection | All | | | | | $[8, 256]_8$ | – | – | – | – |

**Notation:** $[a, b]_s$ denotes integer/float range from $a$ to $b$ with step $s$; $\{...\}$ denotes categorical choices; – indicates parameter not used. [†]**k** was searched only for ECL and Traffic datasets.

## B.4 Baseline Implementation Details

For baselines evaluated under the same experimental setting as our main study, we directly used the reported results from the TimeMixer++ Wang et al. (2025) paper when available, and otherwise reported from the corresponding original papers.

For the synthetic dataset, all compared models were trained using the ETTm1 hyperparameters. Each model was run using the code and hyperparameters from its official repository where available; otherwise, the implementation was taken from `https://github.com/thuml/Time-Series-Library`.

### B.5 Technical Evaluation Note

We compute final metrics for imputation and forecasting as weighted averages across batches to account for varying batch sizes during evaluation. This adjustment is necessary because the last batch in an epoch may contain fewer samples than the standard batch size. When computing performance metrics by simply averaging across batches without considering batch sizes, smaller batches receive disproportionate weight in the final metric calculation, leading to biased performance estimates that do not accurately reflect true model performance across the entire dataset. In many older works, researchers addressed this problem by setting the drop_last=True parameter in PyTorch's DataLoader, which discards the final incomplete batch to ensure identical batch sizes. However, this approach wastes data and can be particularly problematic for smaller datasets, where discarding samples reduces available training or evaluation data. In recent works, it is more common to solve this problem by setting drop_last=False and computing weighted averages, where each batch's metric contribution is weighted by its actual size, ensuring that the final averaged metric accurately represents performance across all samples in the dataset without discarding any data.

### B.6 Computational Profiling

To assess the computational cost of each model, we measure both floating-point operations (FLOPs) and trainable parameter counts across all datasets. Full per-dataset results are reported in Tables 27 and 28.

**FLOPs.** For all real-world datasets, FLOPs are computed using `fvcore`'s `FlopCountAnalysis` Research (2021), which performs a symbolic forward pass to trace the model's computation graph and counts multiply-accumulate operations (MACs) across all layers, including custom modules.

**Trainable parameters.** Parameter counts are obtained via PyTorch's native enumeration: `sum(p.numel() for p in model.parameters() if p.requires_grad)`, ensuring all learnable weights are captured regardless of layer type.

**Excluded models.** TimeMixer++ is excluded because its source code is not publicly available. Crossformer values on ETTh2 and ETTm2 are unavailable (N/A) due to configuration incompatibility with those datasets.

## C   Appendix: Extended Analysis

### C.1   Interpretable Learned Masks Analysis.

This analysis applies to configurations without DeCoP, where the learnable mask is active (see Sec. 4.3).

CRAB (Sec. 4.2) introduces a learnable non-boolean mask that learns the most dominant cross-channel and temporal dependencies. The mask learns dominant dependencies by directly modulating the strength of attention values during training. Specifically, the mask multiplies attention weights element-wise, with higher absolute values amplifying the corresponding attention relationships and values near zero suppressing them. Through gradient-based optimization, the mask automatically discovers which cross-channel and temporal interactions are most informative for the downstream task, effectively learning a data-driven weighting scheme that prioritizes the most predictive dependencies.

Examining the learned mask can therefore provide data-specific insights about these dependency structures, as the mask values directly reflect relationship dominance, with higher absolute values indicating stronger learned dependencies. In this section, we explain how to interpret the attention mask as a foundation for further analysis so it can be leveraged for different analytical needs.

Following data processing (Sec. 4.1), the input data is first permuted so that the patch-sequence dimension is placed before the channel dimension, and then the sequence and channel dimensions are flattened. This creates an attention mask structure that can be visualized as a grid of squares where each square represents cross-channel relationships between pairs of time steps (refer to Figure 3 for visual representation), with the main diagonal squares capturing cross-channel interactions within the same time step and off-diagonal squares revealing temporal cross-channel dependencies.

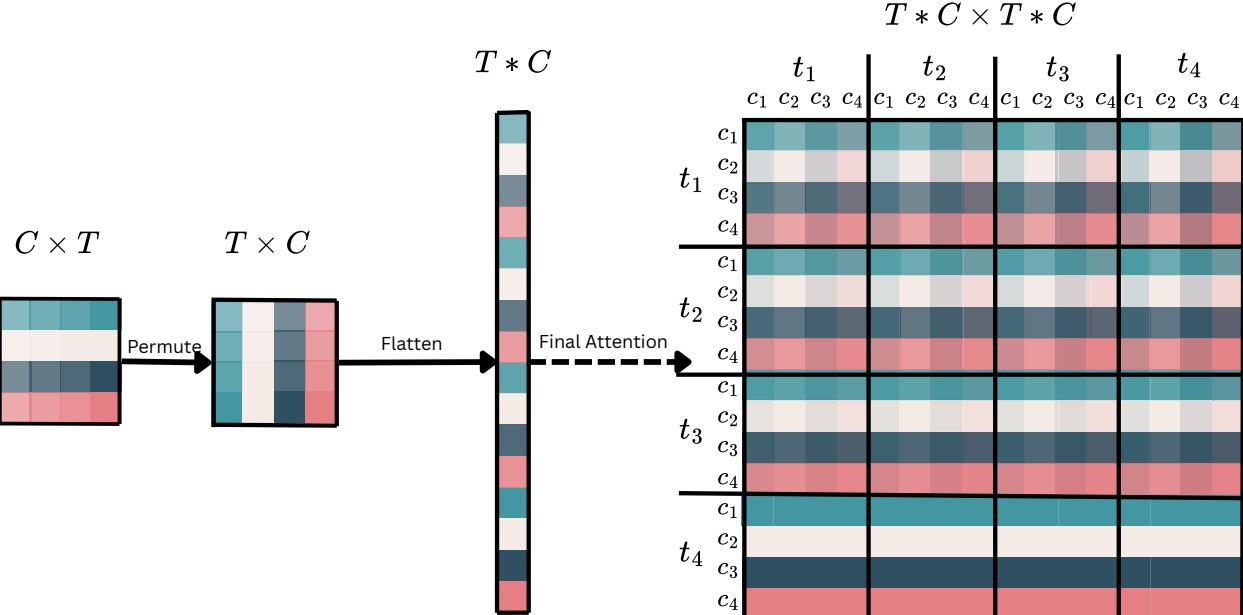

Figure 3: Interpretable Learned Mask Structure: The data permutation step places the patch sequence dimension first, creating an attention mask that can be visualized as a grid of squares where each square represents cross-channel relationships between pairs of time-steps. Note: batch and data dimensions are excluded from this diagram for clarity.

**Analysis of Learnable Masks on ETTm1 Dataset**   Figure 4 analyzes learnable attention masks trained on the ETTm1 dataset across two forecasting scenarios: 96→96 (top row) and 96→192 (bottom row). Each row displays three visualizations: initial random masks initiated from a normal distribution (left), learned

patterns after training (middle), and corresponding heatmaps quantifying cross-channel dependency strength (right).

**Data Processing and Architecture.** The ETTm1 dataset was processed using patches of length 16 with a stride of 8, generating 12 patches across ETTm1's 7 channels. This configuration produces an $84 \times 84$ attention matrix ($12 \times 7 = 84$ dimensions) that captures both temporal and cross-channel relationships.

**Heatmap Interpretation.** The dependency strength heatmaps are derived from the trained masks by averaging the absolute values within each cross-channel grid. Since masks are applied to attention weights, higher absolute mask values correspond to more dominant dependencies, with darker red regions in the heatmap indicating stronger cross-channel relationships between specific time steps.

**Key Findings.** The trained masks exhibit several notable patterns. First, they develop structured grid formations that align precisely with the 12-patch architecture, suggesting the model learns systematic cross-channel dependencies. By examining the heatmaps from both configurations, we observe a high density of dominant dependencies along the main diagonal. This diagonal concentration indicates that the model learns strong self-attention patterns, in which each time step primarily attends to itself and its immediate temporal neighbors. Such patterns suggest that the most informative relationships for forecasting are local temporal dependencies, in which recent observations carry the greatest predictive power for future values. This finding aligns with the intuitive understanding that in time-series analysis, temporally proximate data points are typically more relevant than distant historical information.

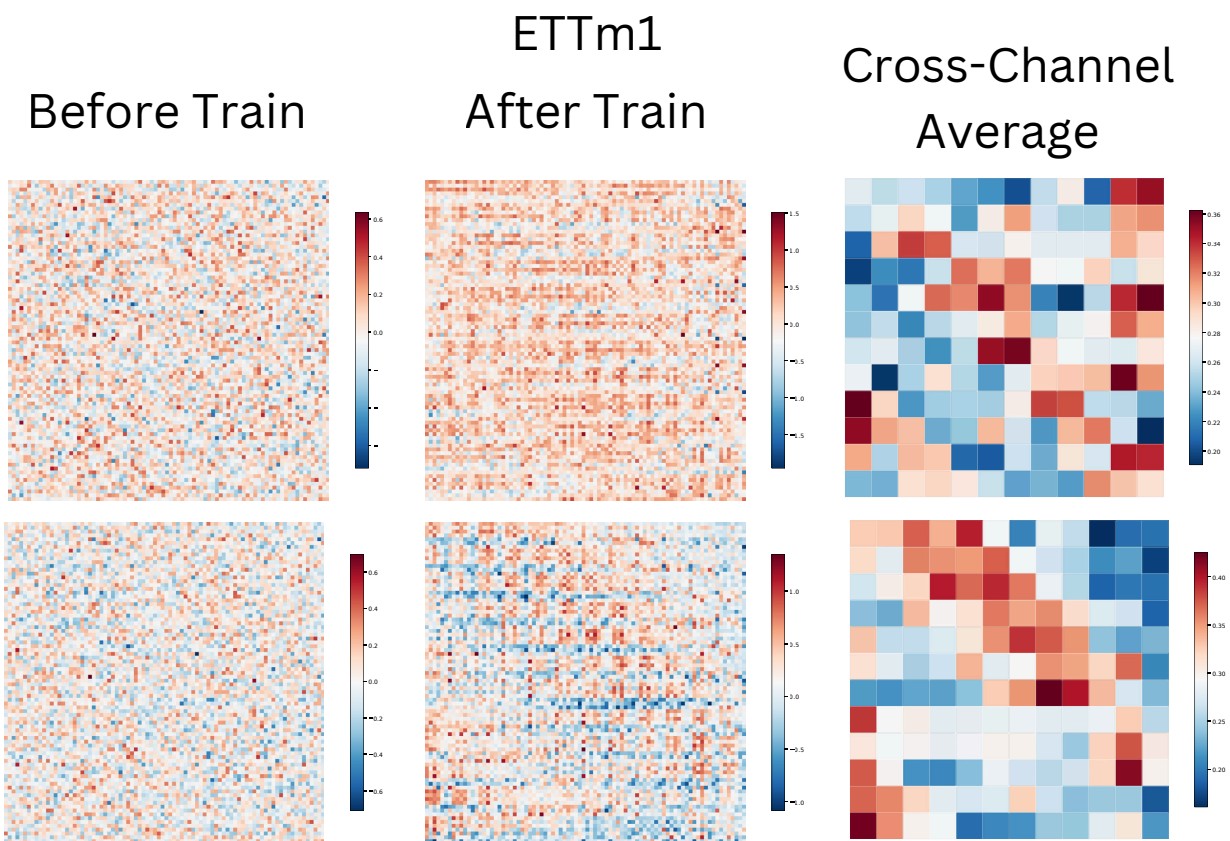

Figure 4: Analysis of learnable attention masks on ETTm1 dataset. Top row: 96→96 forecasting; bottom row: 96→192 forecasting. Left column: initial random masks; middle column: learned structured patterns after training; right column: heatmaps of cross-channel dependency strength derived from trained masks. The heatmaps visualize the strength of cross-channel dependencies between time points, with darker red regions indicating stronger relationships.

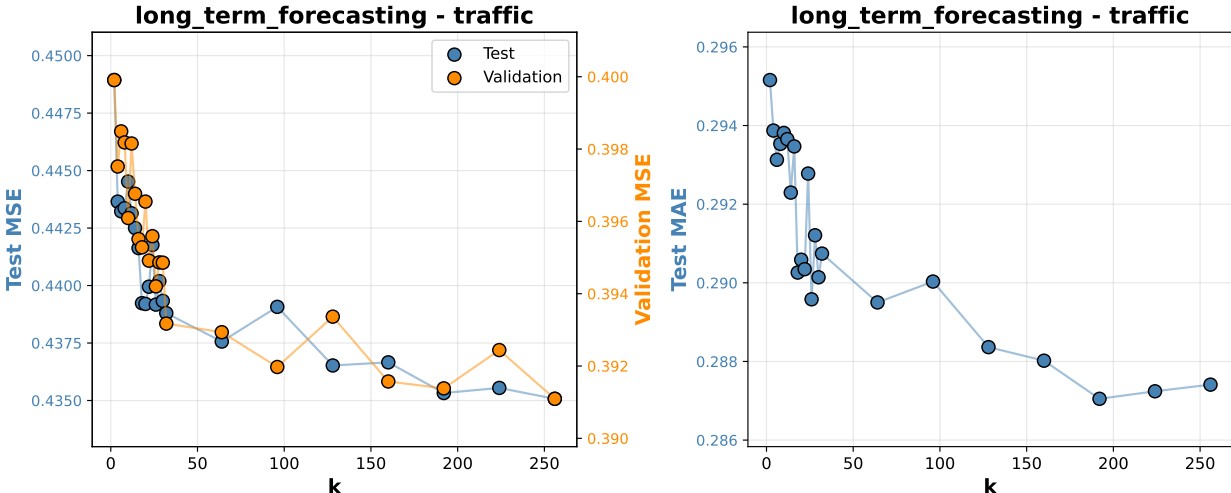

Figure 5: **DeCoP $k$ sensitivity on Traffic (forecasting).** Average MAE/MSE across horizons $\{96, 192, 336, 720\}$ for different compressed representation sizes $k$.

## C.2 DeCoP compression size analysis

As recalled, DeCoP compresses token-to-token interactions into a low-dimensional representation; the choice of $k$ directly controls the expressive capacity of this bottleneck and thus can affect both accuracy and efficiency. In particular, larger $k$ increases the dimensionality of the compressed attention embedding, which summarizes each token's pairwise interactions with all other tokens, enabling richer dependency modeling. However, this comes at a higher compute and memory cost. Therefore, analyzing sensitivity to $k$ is important for providing practical guidance on choosing $k$ that accounts for resource constraints.

Concretely, we tested $k \in [2, 32]$ (step 2) and $k \in [64, 256]$ (step 32). For each $k$, we report the average MAE/MSE across forecasting horizons $96, 192, 336, 720$ and the average MAE/MSE across imputation mask ratios $0.125, 0.25, 0.375, 0.5$. Overall, we observe dataset-dependent behavior. On **Traffic** (forecasting analysis Table 5), performance tends to improve as $k$ increases, indicating that a higher-capacity compressed representation better captures the complex multivariate dependencies in this dataset. In contrast, on **ECL** (forecasting analysis Table 6, imputation analysis Table 7), smaller or mid-range $k$ values are often competitive and occasionally slightly better, suggesting that stronger compression can provide a useful regularizing effect and that further increasing $k$ yields limited additional benefit. Based on these results, we recommend treating $k$ as a *dataset-specific* hyperparameter and tuning it to balance accuracy with compute and memory costs, as larger $k$ values often yield only marginal gains.

## C.3 Patch length and stride analysis.

Patching determines how the input sequence is divided into fixed-length windows, where patch_len sets the window size, and each window is projected into a token embedding. The stride controls the overlap between consecutive windows and thus the density with which the sequence is covered. Together, they affect which information is represented in each token and how many tokens the transformer processes. With a larger `patch_len` or stride, the model uses fewer tokens, but each token must represent a longer window, which can make it harder to preserve meaningful temporal information. With a smaller `patch_len` or stride, tokens represent more local information and overlap increases, but the longer token sequence raises the computational cost of attention. Therefore, `patch_len` and stride define a fundamental accuracy efficiency trade-off. This sensitivity analysis evaluates how robust our method is to this configuration and if dataset-specific tuning may be required.

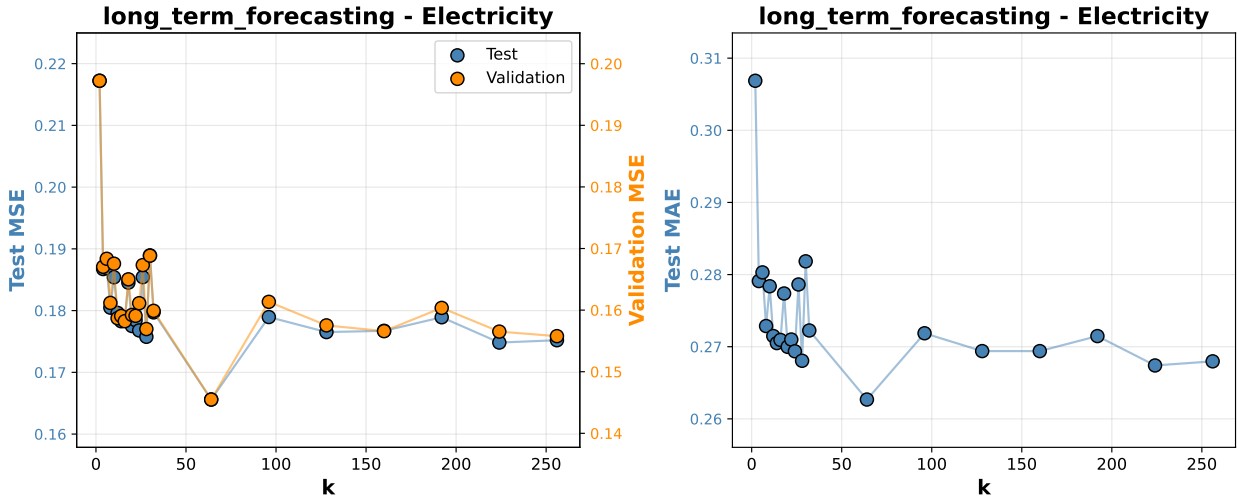

Figure 6: **DeCoP $k$ sensitivity on ECL (forecasting).** Average MAE/MSE across horizons $\{96, 192, 336, 720\}$ for different compressed representation sizes $k$.

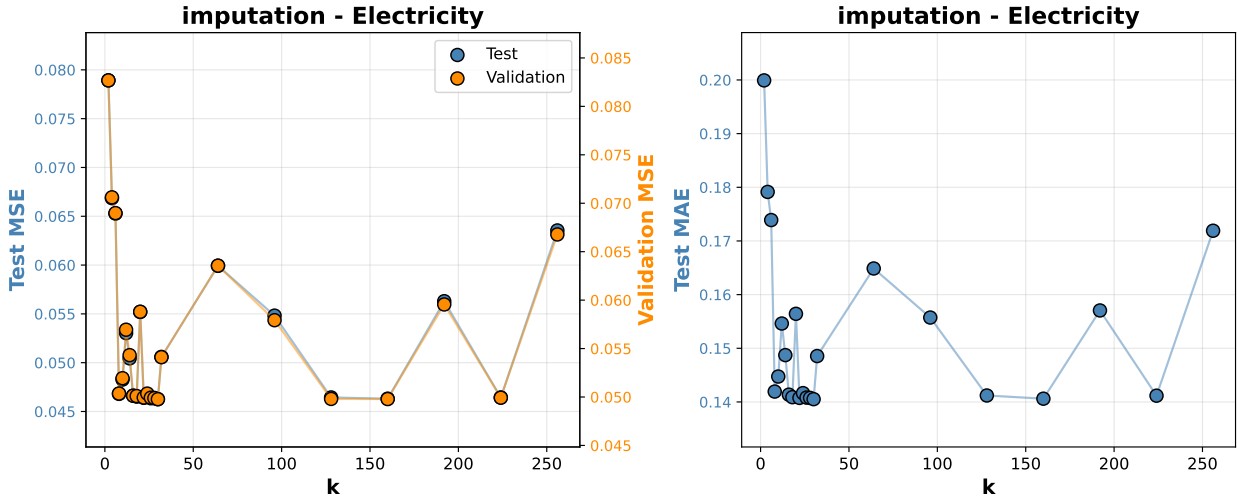

Figure 7: **DeCoP $k$ sensitivity on ECL (imputation).** Average MAE/MSE across mask ratios $\{0.125, 0.25, 0.375, 0.5\}$ for different compressed representation sizes $k$.

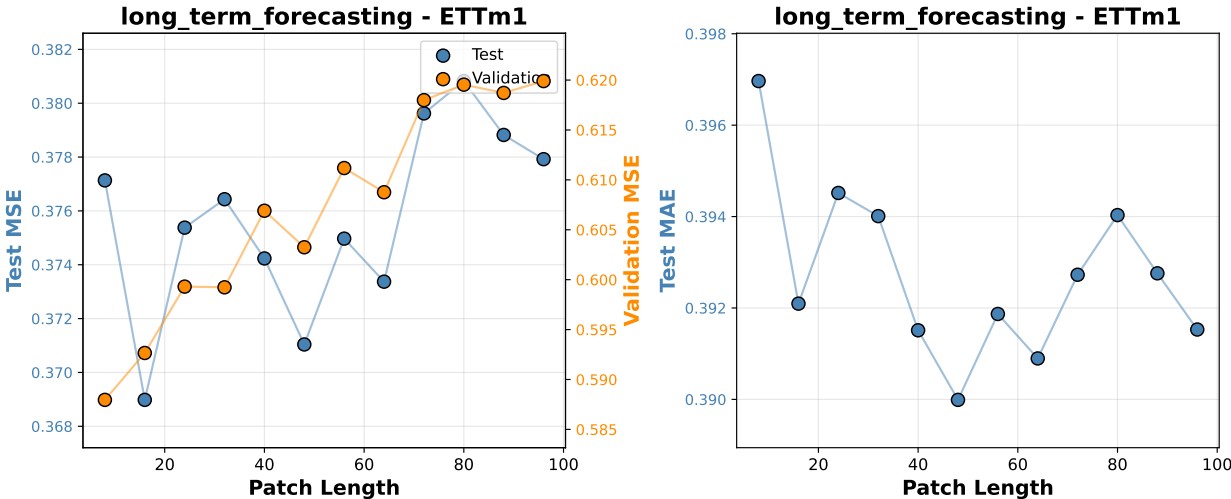

Figure 8: **Patch length sensitivity on ETTm1 (forecasting).** Average validation and test losses across horizons $\{96, 192, 336, 720\}$ for different `patch_len` values with `stride = patch_len`$/2$.

Following PatchTST's (Nie et al., 2023) best practice, we kept the patching configuration fixed in most of our main experiments. For forecasting and anomaly detection, we used `patch_len`$= 16$ and `stride`$= 8$. For imputation, where the lookback was substantially larger ($L$=1024 versus 96 in forecasting and 100 in anomaly detection), we tested a small set of `patch_len` values $16, 64, 128$ with the corresponding `stride`$=$ `patch_len`$/2$. Based on validation error during hyperparameter search, we selected `patch_len`$= 64$ and `stride`$= 32$ for some imputation datasets.

In this subsection, we conducted a patching sensitivity experiment for forecasting by fixing `stride` $=$ `patch_len`$/2$ and testing different values of `patch_len` from 8 to 96 in increments of 8. The plots report the average validation and test losses on the ETTm1 (Figure 8) and Weather (Figure 9) datasets, where the average is computed across horizons $\{96, 192, 336, 720\}$. Overall, performance varies only slightly across different setups, indicating that our method is robust to this hyperparameter. While the effect is small on **ETTm1**, it is more noticeable on **Weather**, yet the differences remain limited to roughly 2-4%, suggesting that the preferred patching configuration can be dataset-dependent without requiring highly precise tuning.

### C.4   Signed-attention mask analysis.

**Distribution of negative weights.**   As recalled from the Method (Sec. 4.2), we do not use the vanilla attention scores directly. Instead, we first apply a positive transformation and then adjust the scores with a learnable, non-Boolean attention mask, which controls both the sign and the magnitude of the resulting attention weights. Concretely, given an attention score matrix $A \in \mathbb{R}^{N \times N}$, we remove sign information via a global shift $A_+ = A - \min(A)$, which ensures $A_+ \geq 0$ element-wise, and then form the signed attention as $A = M \circ A_+$, where $M \in \mathbb{R}^{N \times N}$ is a learnable real-valued mask applied element-wise. Since $A_+$ is non-negative, the sign of each entry in $A$ is determined entirely by the corresponding mask value: $M_{ij} < 0$ produces a negative attention weight, while $M_{ij} > 0$ produces a positive one. We initialize $M$ from a zero-mean Gaussian distribution, so roughly half of its entries are negative at initialization, and consequently, about half of the resulting attention weights are negative as well. Although $M$ is fine-tuned during training, its distribution remains close to normal, so negative weights persist and can contribute throughout optimization. Take a look at Figure 10, which shows the distributions of the mask values and the activated attention weights at the end of the first and last training epochs.

**Ablation: clipping negative weights.**   We further isolate the role of negative weights with a targeted ablation. We keep our signed-attention activation unchanged, and only add a final ReLU that clips all negative

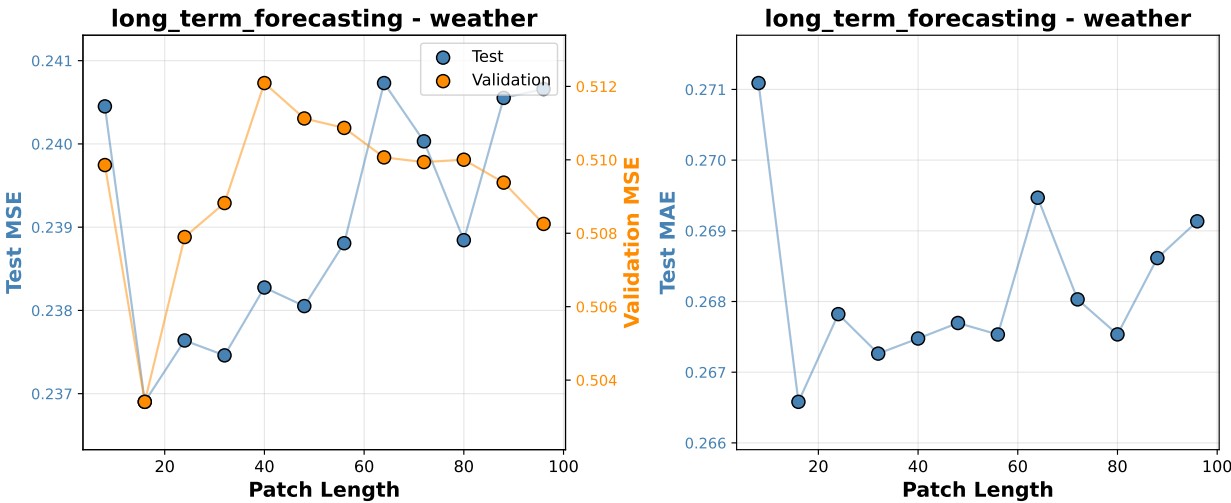

Figure 9: **Patch length sensitivity on Weather (forecasting).** Average validation and test losses across horizons $\{96, 192, 336, 720\}$ for different `patch_len` values with `stride = patch_len/2`.

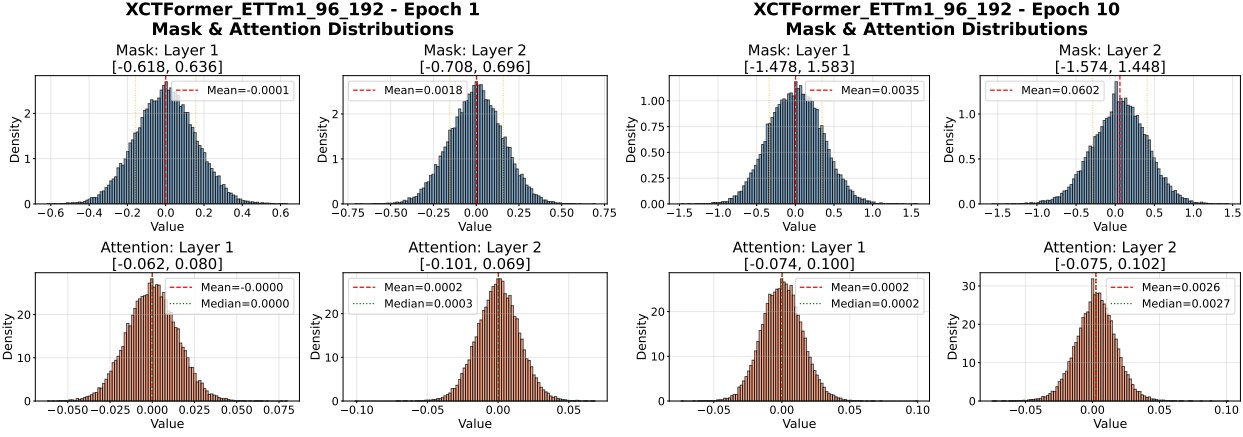

Figure 10: **Distributions of mask and signed-attention weights.** Histograms of the learnable mask values $M$ and the resulting activated attention weights. Results are shown for the forecasting task upon the ETTm1 dataset with lookback $L=96$ and horizon $H=192$. The left panel shows the distributions after the first training epoch, and the right panel after the final (10th) epoch. The distributions remain approximately Gaussian over training, indicating that negative weights persist.

values to zero. This variant is trained with the exact same hyperparameters and experimental settings, covering all forecasting horizons $\{96, 192, 336, 720\}$ and all imputation mask ratios $\{0.125, 0.25, 0.375, 0.5\}$. We then report performance averaged across these settings (see Table 11). Overall, the original model that permits negative weights improves performance by about 1.3% over the clipped variant, suggesting that negative weights provide a consistent but modest performance gain.

## C.5 Empirical Scalability Analysis with Respect to Input Dimension

As noted in the Limitations, XCTFormer is sensitive to input size because it explicitly models all pairwise time and channel relationships, which induces quadratic scaling. We design DeCoP to mitigate this cost by

Table 11: Ablation study results across different tasks, evaluated with different XCTFormer with clipped values.

| | Long-term Forecasting | | Imputation | | Anomaly Detection | | | XCTFormer vs Others |
|---|---|---|---|---|---|---|---|---|
| | MSE | MAE | MSE | MAE | Precision | Recall | F-Score | (%) |
| XCTFormer (Original) | **0.328** | **0.337** | **0.044** | **0.124** | **92.1** | **83.7** | **87.6** | - |
| AbsAct + ReLU | 0.331 | 0.339 | 0.044 | 0.124 | 92.1 | 79.5 | 85.2 | 1.3% |

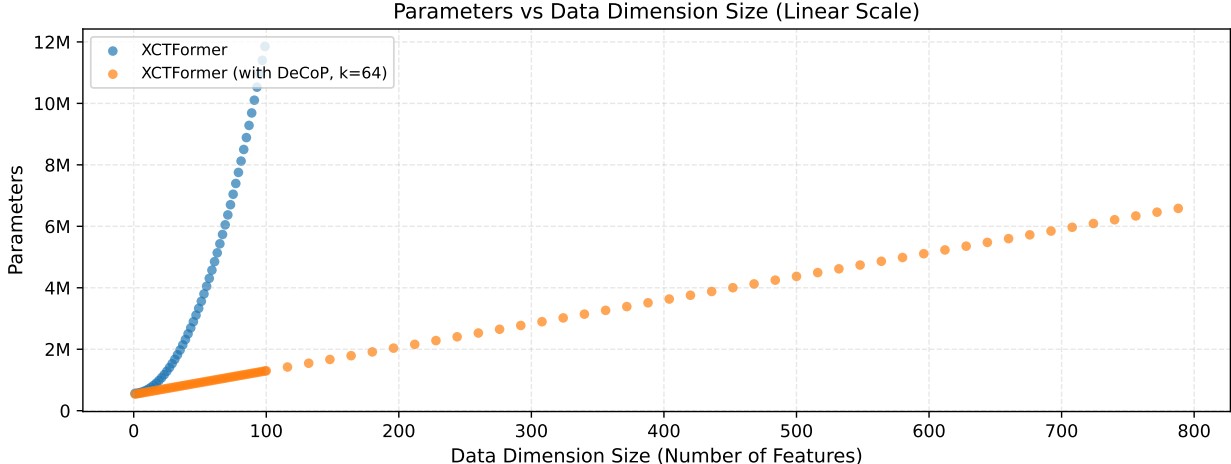

Figure 11: Scalability with respect to channel dimensionality (*n_features*). XCTFormer's parameter count grows quadratically, while the DeCoP variant scales approximately linearly.

compressing per-token dependencies and thereby achieving approximately linear scaling. To verify these theoretical expectations, we conduct an empirical scalability analysis. Because the effective input equals the product of the sequence tokens (after patching) and the number of channels, we vary each factor independently and measure how the parameter amount changes. Table 12 represents the hyperparameter used.

Table 12: Hyperparameter settings for **XCTFormer** scalability analysis

| | Data Processing | | Transformer | | | | | | XCTFormer | | Variable |
|---|---|---|---|---|---|---|---|---|---|---|---|
| Experiment | patch_len | stride | e_layers | n_heads | d_model | d_ff | dropout | fc_dropout | attn_dropout | k | Range |
| | | | | | | | | | | | |
| Dimension Scaling Analysis | | | | | | | | | | | |
| XCTFormer | 16 | 8 | 2 | 4 | 128 | 256 | 0.1 | 0.05 | 0.0 | - | n_features: 1–100 (step 2) |
| XCTFormer (DeCoP) | 16 | 8 | 2 | 4 | 128 | 256 | 0.1 | 0.05 | 0.0 | 64 | n_features: 1–100 (step 2), 100–800 (step 16) |
| Sequence Length Scaling Analysis | | | | | | | | | | | |
| XCTFormer | 16 | 8 | 2 | 4 | 128 | 256 | 0.1 | 0.05 | 0.0 | - | seq_len: 64–1024 (step 32) |
| XCTFormer (DeCoP) | 16 | 8 | 2 | 4 | 128 | 256 | 0.1 | 0.05 | 0.0 | 64 | seq_len: 64–1024 (step 32), 1024–4096 (step 128) |

**Channel scaling.** We fix the lookback sequence length to 96 and vary the number of channels to match the dataset's dimensionality. Specifically, we tested *n_features* from 1 to 100 in steps of 2, and further extended from 100 to 800 in steps of 16 for the DeCoP variant. Because utilizing XCTFormer without DeCoP on larger datasets is impractical, the original XCTFormer is evaluated only over the smaller range. As shown in Fig. 11, XCTFormer exhibits quadratic parameter growth with respect to the number of channels, whereas adding DeCoP yields approximately linear growth.

**Sequence scaling.** We fix the data dimensionality to 10 and vary the sequence length, which determines the number of tokens after patching. We evaluate sequences from 64 to 1024 in steps of 32, and extend from 1024 to 4096 in steps of 128 for the DeCoP variant. Again, XCTFormer is evaluated only on the shorter range.

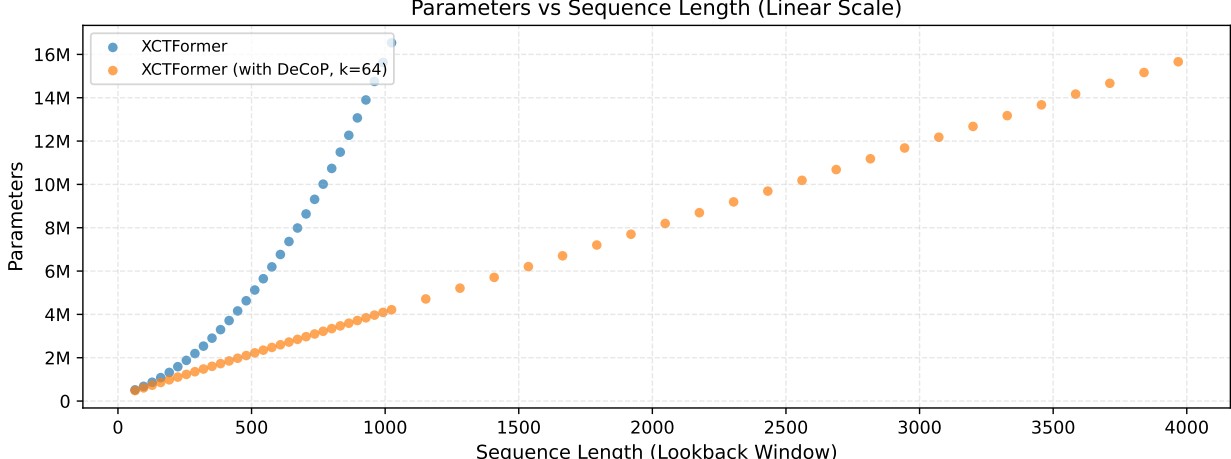

Figure 12: Scalability with respect to sequence length (`seq_len`). XCTFormer exhibits quadratic parameter growth, while the DeCoP variant scales approximately linearly.

Fig. 12 shows the same trend: XCTFormer grows quadratically with sequence length, whereas XCTFormer with DeCoP scales approximately linearly.

## D  Appendix: Synthetic Dataset

### D.1  Overview & Motivation

While existing synthetic datasets are available (Nochumsohn et al., 2024; Nochumsohn & Azencot, 2025; Oreshkin et al., 2026) and generative approaches enable sampling from empirical data (Naiman et al., 2024b;a; Gonen et al., 2025; Fadlon et al., 2025), they do not explicitly target the interplay between temporal and cross-variate dependencies. We propose a new synthetic dataset, evaluated on the forecasting task, to emphasize settings in which forecastability depends jointly on cross-variate and lagged cross-time relationships. The dataset consists of seven channels and is evaluated under the multivariate-to-single (MS) setting: all channels serve as input, but training and evaluation loss is computed only on the 7th channel (the synthetic target). The MS setting is adopted to specifically evaluate each model's ability to forecast data that depends solely on cross-channel and cross-time relationships. This target channel is constructed entirely from lagged signals in the first four channels, making it difficult to predict from its own history alone. Channels 5 and 6 (var_5, var_6) are independent random walks that act as distractors, meant to probe whether a model latches onto spurious correlations, i.e., learns to rely on channels that share no causal relationship with the target, which would degrade generalization.

### D.2  Dataset Construction

The dataset comprises 7 channels and 10,000 time steps. At a high level, the channels fall into three groups:

- **Informative sources** (channels 1–4): four sine waves with distinct amplitudes and frequencies that carry the information needed to predict the target.

- **Distractors** (channels 5–6): two random walks that share no causal relationship with the target. They test whether a model can ignore spurious correlations.

- **Target** (channel 7): constructed entirely from *lagged patches* of the four source channels. At each point in time the target blends patches from different sources at different look-back distances, and the blend weight itself changes over time. Because the target is derived solely from other channels'

past values, it is difficult to predict from its own history alone. Small Gaussian noise is added for realism.

The remainder of this section specifies each construction step in detail.

### D.2.1 Step 1: Source Signals

Table 13 lists all seven channels. The four informative sources are sine waves with varying amplitudes (1.0–5.0), frequencies (0.002–0.03), and phase offsets, ensuring the signals are diverse and not trivially correlated. The two distractor channels are random walks, which are non-stationary and can exhibit spurious short-range correlations with any signal. A model that naively attends to all variates equally would be misled by these distractors.

Table 13: Synthetic dataset variate specifications.

| Index | Name | Type | Role |
|---|---|---|---|
| 1 | var_1 | Sine wave ($A$=1.0, $f$=0.02) | Informative source |
| 2 | var_2 | Sine wave ($A$=3.0, $f$=0.03) | Informative source |
| 3 | var_3 | Sine wave ($A$=2.0, $f$=0.01) | Informative source |
| 4 | var_4 | Sine wave ($A$=5.0, $f$=0.002) | Informative source |
| 5 | var_5 | Random walk ($\sigma$=0.1) | Distractor |
| 6 | var_6 | Random walk ($\sigma$=0.15) | Distractor |
| 7 | target | Patch-dependent blend | Target (predicted) |

### D.2.2 Step 2: Patching

The source signals are divided into non-overlapping patches of length 16 with a stride of 8 (50% overlap). All subsequent operations are defined at the patch level.

### D.2.3 Step 3: Target Construction

The target channel is built by blending patches from two pairs of source variates, each read at a specific lag. The two pairs are:

- **Pair 1**: var_1 (lag$=1$) and var_2 (lag$=2$), contributing weight 0.5.

- **Pair 2**: var_3 (lag$=2$) and var_4 (lag$=3$), contributing weight 0.5.

The different lags ensure that the model cannot simply align patches at a single fixed offset; it must learn to look back different distances for different variates.

Within each pair, the relative contribution of the two sources is controlled by a blend weight $w(k)$ that varies over time. Specifically, $w(k)$ follows a triangle wave cycling between 0 and 1 with a period of 20 patches (160 time steps), creating a non-stationary dependency structure:

$$w(k) = \begin{cases} (k \bmod 20)/10, & \text{if } (k \bmod 20) \leq 10 \\ 2 - (k \bmod 20)/10, & \text{otherwise} \end{cases} \tag{15}$$

That is, $w(k)$ cycles through the values $0, 0.1, 0.2, \ldots, 0.9, 1.0, 0.9, \ldots, 0.1, 0, 0.1, \ldots$ as $k$ increases.

Putting it all together, the target patch at index $k$ is:

$$\text{target}[k] = 0.5 \cdot \big[w(k) \cdot \text{var\_1}[k-1] + (1-w(k)) \cdot \text{var\_2}[k-2]\big] + 0.5 \cdot \big[w(k) \cdot \text{var\_3}[k-2] + (1-w(k)) \cdot \text{var\_4}[k-3]\big] \tag{16}$$

where var_$i[k-\ell]$ denotes the 16-element patch extracted from variate $i$ at patch position $k-\ell$. Overlapping patch regions are averaged.

### D.2.4 Step 4: Noise

Finally, Gaussian noise ($\sigma$=0.02) is added to the target channel to prevent trivial exact recovery and simulate real-world measurement noise.

### D.3 Dataset Splits & Preprocessing

Table 14 summarizes the dataset splits and evaluation configuration.

Table 14: Synthetic dataset splits and evaluation configuration.

| Property | Parameter | Value |
|---|---|---|
| Train / Val / Test split | Proportion | 70% / 10% / 20% |
| Total points | $n$ | 10,000 |
| Normalization | StandardScaler | Fitted on train only |
| Lookback window | `seq_len` | 96 |
| Decoder label length | `label_len` | 48 |
| Prediction horizon | `pred_len` | {96, 192, 336, 720} |
| Training epochs | `train_epochs` | 10 |

### D.4 Results

Table 15 reports per-horizon forecasting results on the synthetic dataset, together with the trainable parameter count and FLOPs for each model on this dataset.

Table 15: Forecasting results on the synthetic dataset (MS setting) across prediction horizons, with computational cost on this dataset. All models use ETTm1 hyperparameters without dataset-specific tuning.

| Model | 96 MSE | 96 MAE | 192 MSE | 192 MAE | 336 MSE | 336 MAE | 720 MSE | 720 MAE | Avg MSE | Avg MAE | Params | FLOPs |
|---|---|---|---|---|---|---|---|---|---|---|---|---|
| **Crossformer** | **0.018** | **0.106** | **0.026** | **0.126** | **0.027** | **0.129** | **0.031** | **0.137** | **0.025** | **0.124** | 1.1E+07 | 1.2E+09 |
| **XCTFormer** (Ours) | 0.038 | 0.151 | _0.043_ | _0.161_ | _0.045_ | _0.165_ | _0.035_ | _0.144_ | _0.040_ | _0.155_ | 2.0E+05 | 3.3E+06 |
| TimeMixer | _0.025_ | _0.123_ | 0.070 | 0.204 | 0.053 | 0.178 | 0.133 | 0.281 | 0.070 | 0.197 | 1.2E+05 | 1.6E+07 |
| Leddam | 0.043 | 0.159 | 0.094 | 0.235 | 0.095 | 0.234 | 0.082 | 0.216 | 0.079 | 0.211 | 2.9E+06 | 3.5E+07 |
| SCINet | 0.130 | 0.277 | 0.092 | 0.235 | 0.085 | 0.225 | 0.079 | 0.218 | 0.097 | 0.239 | 2.8E+05 | 3.4E+06 |
| iTransformer | 0.217 | 0.362 | 0.236 | 0.375 | 0.282 | 0.411 | 0.282 | 0.413 | 0.254 | 0.390 | 2.6E+05 | 2.9E+06 |
| PatchTST | 0.222 | 0.365 | 0.319 | 0.442 | 0.319 | 0.441 | 0.249 | 0.384 | 0.277 | 0.408 | 9.2E+05 | 3.8E+07 |
| TimesNet | 0.554 | 0.597 | 0.471 | 0.538 | 0.404 | 0.499 | 0.350 | 0.459 | 0.445 | 0.523 | 2.7E+06 | 3.8E+09 |
| DLinear | 0.516 | 0.577 | 0.496 | 0.572 | 0.482 | 0.565 | 0.428 | 0.523 | 0.480 | 0.559 | 6.5E+04 | 4.5E+05 |
| MTLinear | 0.670 | 0.640 | 1.189 | 0.881 | 1.388 | 0.975 | 1.149 | 0.845 | 1.099 | 0.835 | 2.0E+05 | 3.4E+05 |
| TiDE | 0.836 | 0.746 | 1.290 | 0.934 | 1.476 | 1.016 | 1.182 | 0.875 | 1.196 | 0.893 | 9.6E+06 | 9.2E+07 |
| Autoformer | 0.573 | 0.605 | 1.753 | 1.094 | 1.743 | 1.079 | 1.606 | 1.019 | 1.419 | 0.949 | 1.2E+05 | 2.4E+07 |
| FEDformer | 1.537 | 1.006 | 2.422 | 1.282 | 4.065 | 1.663 | 1.879 | 1.110 | 2.476 | 1.265 | 1.6E+07 | 1.8E+09 |

The two models that explicitly capture both cross-variate and cross-time patch interactions, namely Crossformer and XCTFormer, achieve the lowest errors, consistent with the dataset's construction in which the target depends on lagged patches from other channels. Crossformer ranks first (Avg MSE = 0.025) and XCTFormer ranks second (Avg MSE = 0.040). However, Crossformer uses approximately 55× more trainable parameters (11M vs. 200K) and 364× more FLOPs (1.2E+09 vs. 3.3E+06) than XCTFormer on this dataset, highlighting a substantial efficiency gap.

We note that no dataset-specific hyperparameter tuning was performed: all models were trained with ETTm1 hyperparameters. Consequently, the reported results may not reflect each model's optimal performance on this dataset.

Channel-independent models (e.g., PatchTST, DLinear) tend to show higher error, which is expected given that the target is difficult to predict from its own history alone. The distractor channels (random walks) do

not appear to noticeably affect the performance of cross-variate models, suggesting these architectures can, to some extent, distinguish informative from non-informative channels.

A visualization of the synthetic dataset is shown in Figure 13, and qualitative forecasting examples on this dataset are provided in Figure 14.

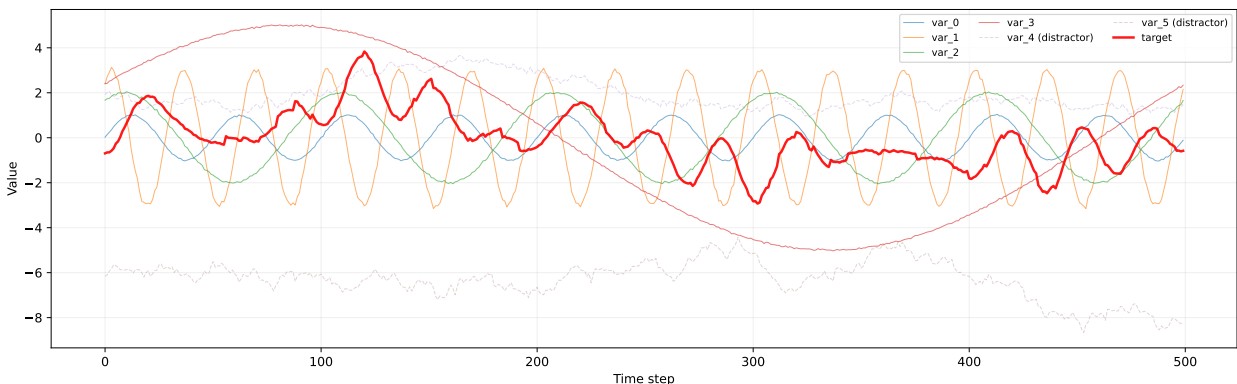

Figure 13: Synthetic dataset visualization: source signals (var_1 through var_6) and the constructed target channel.

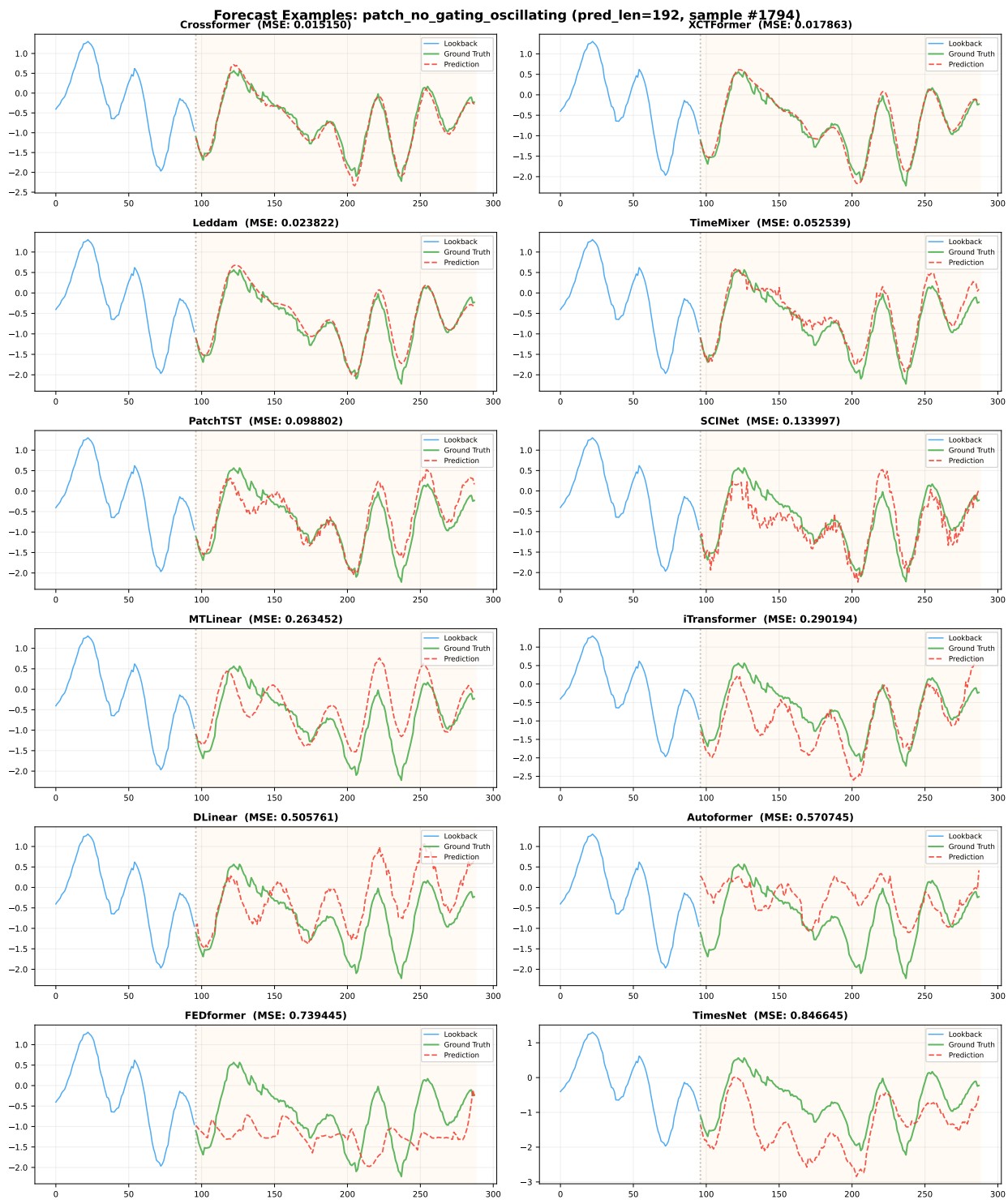

Figure 14: Prediction examples on the synthetic dataset: ground-truth target vs. model forecasts for selected models.

# E    Appendix: Full Results

## E.1    Statistical significance tests

To assess whether the observed performance differences are robust to random initialization, we conducted statistical significance tests for the **forecasting** task against two competitive baselines, *LeDDAM* (Yu et al., 2024) and *iTransformer* Liu et al. (2024), under the same evaluation protocol used throughout the paper. The results are reported in Tables 16 and 17 respectively.

**Experimental setup.**    For each dataset, we evaluated all models on the four standard prediction horizons $\{96, 192, 336, 720\}$. For every method (ours and each baseline), we trained the model **five** times using different random seeds $\{2021, 2022, 2023, 2024, 2025\}$. For our model, we used the hyperparameters reported in Appendix B.3. For the baselines, we used the best hyperparameters provided in their official GitHub repositories. All runs followed the same training procedure as the main experiments, including model selection based on validation loss and reporting test-set errors. We applied a two-sided paired t-test with $\alpha = 0.05$. We mark a result as significant when the mean difference favors our model and $p \leq 0.05$; otherwise, we treat it as inconclusive.

Table 16: Statistical comparison of XCTFormer vs LeDDAM on forecasting datasets. Results averaged over prediction lengths $\{96, 192, 336, 720\}$ across five seeds (2021-2025). Confidence level derived from Welch's t-test (99.9%: $p < 0.001$, 99%: $p < 0.01$, 95%: $p < 0.05$). **Bold** indicates statistically significant better performance.

| Dataset | XCTFormer (Ours) | | LeDDAM | | Confidence Level |
| --- | --- | --- | --- | --- | --- |
| | MSE | MAE | MSE | MAE | |
| ETTh1 | $0.449 \pm 0.002$ | $0.436 \pm 0.001$ | $\mathbf{0.436} \pm 0.0073$ | $\mathbf{0.432} \pm 0.0034$ | 95% |
| ETTh2 | $0.374 \pm 0.007$ | $0.399 \pm 0.004$ | $0.374 \pm 0.0019$ | $0.398 \pm 0.0007$ | n.s. |
| ETTm1 | $\mathbf{0.371} \pm 0.003$ | $\mathbf{0.393} \pm 0.002$ | $0.388 \pm 0.0034$ | $0.398 \pm 0.0023$ | 99% |
| ETTm2 | $\mathbf{0.271} \pm 0.001$ | $\mathbf{0.319} \pm 0.001$ | $0.282 \pm 0.0019$ | $0.326 \pm 0.0009$ | 99.9% |
| ECL$^p$ | $0.176 \pm 0.007$ | $0.270 \pm 0.007$ | $0.171 \pm 0.0042$ | $0.264 \pm 0.0027$ | n.s. |
| Traffic$^p$ | $\mathbf{0.435} \pm 0.001$ | $\mathbf{0.287} \pm 0.001$ | $0.468 \pm 0.0075$ | $0.294 \pm 0.0052$ | 95% |
| Weather | $\mathbf{0.237} \pm 0.001$ | $\mathbf{0.267} \pm 0.001$ | $0.244 \pm 0.0013$ | $0.273 \pm 0.0012$ | 99.9% |

n.s. Not statistically significant ($p \geq 0.10$).
$^p$ DeCoP was enabled for XCTFormer on this dataset.

Table 17: Statistical comparison of XCTFormer vs iTransformer on forecasting datasets. Results averaged over prediction lengths $\{96, 192, 336, 720\}$ across five seeds (2021-2025). Confidence level derived from Welch's t-test (99.9%: $p < 0.001$, 99%: $p < 0.01$, 95%: $p < 0.05$). **Bold** indicates statistically significant better performance.

| Dataset | XCTFormer (Ours) | | iTransformer | | Confidence Level |
| --- | --- | --- | --- | --- | --- |
| | MSE | MAE | MSE | MAE | |
| ETTh1 | $\mathbf{0.449} \pm 0.002$ | $\mathbf{0.436} \pm 0.001$ | $0.457 \pm 0.0014$ | $0.449 \pm 0.0013$ | 99.9% |
| ETTh2 | $\mathbf{0.374} \pm 0.007$ | $\mathbf{0.399} \pm 0.004$ | $0.383 \pm 0.0022$ | $0.407 \pm 0.0011$ | 95% |
| ETTm1 | $\mathbf{0.371} \pm 0.003$ | $\mathbf{0.393} \pm 0.002$ | $0.408 \pm 0.0022$ | $0.412 \pm 0.0013$ | 99.9% |
| ETTm2 | $\mathbf{0.271} \pm 0.001$ | $\mathbf{0.319} \pm 0.001$ | $0.291 \pm 0.0010$ | $0.335 \pm 0.0011$ | 99.9% |
| ECL$^p$ | $0.176 \pm 0.007$ | $0.270 \pm 0.007$ | $0.176 \pm 0.0037$ | $0.267 \pm 0.0026$ | n.s. |
| Traffic$^p$ | $0.435 \pm 0.001$ | $0.287 \pm 0.001$ | $\mathbf{0.430} \pm 0.0010$ | $\mathbf{0.283} \pm 0.0010$ | 99.9% |
| Weather | $\mathbf{0.237} \pm 0.001$ | $\mathbf{0.267} \pm 0.001$ | $0.260 \pm 0.0014$ | $0.281 \pm 0.0016$ | 99.9% |

n.s. Not statistically significant ($p \geq 0.10$).
$^p$ DeCoP was enabled for XCTFormer on this dataset.

**Results.**   Overall, our model achieves strong performance and shows statistically significant improvements across most datasets relative to both baselines. Compared to *LeDDAM*, the results are inconclusive on ECL and ETTh2, while ETTh1 favors LeDDAM. Compared to *iTransformer*, the result is inconclusive on ECL, and Traffic favors iTransformer. These outcomes suggest that gains are often consistent across seeds, but in some datasets, the differences are not statistically significant.

## E.2   Ablation Study: Complete Analysis

We conduct a systematic ablation study with six configurations to isolate the contribution of each architectural component in **XCTFormer**. All variants maintain identical data processing (patch length/stride), training procedures, and model parameters, except for the specific component being modified.

**Configuration Details**

1. **Full XCTFormer (Baseline)**: Complete architecture including CRAB module with learnable non-boolean mask (for datasets with ≤60 channels) or DeCoP (for datasets with >60 channels), cross-time and cross-channel attention, and our proposed attention activation function.

2. **W/o Learnable Mask**: Removes the learnable mask component. Attention masks are not converted to positive values and no element-wise multiplication is applied. The CRAB module remains unchanged otherwise.

3. **Standard Softmax Activation**: Replaces our proposed activation function with standard Transformer softmax while preserving CRAB and the learnable mask. Note that our `attention_dropout` rate parameter is replaced with the standard `dropout` argument commonly used in related work for fair comparison.

4. **Vanilla Transformer**: Substitutes CRAB (and DeCoP) with standard attention blocks following Vaswani et al. (2017).

5. **Sequence Modeling Only**: Retains only temporal self-attention within each channel, disabling cross-channel modeling (channel-independent processing). This configuration tests the necessity of modeling cross-channel relationships, mirroring approaches like PatchTST (Nie et al., 2023).

6. **Channel Modeling Only**: Preserves only cross-channel attention at each time step while removing temporal self-attention. This configuration tests the necessity of modeling temporal relationships, similar to designs that emphasize cross-variable mixing like iTransformer (Liu et al., 2024).

**Full Ablation Study Results**   Complete ablation study results for each time-series task are presented in Tables 18, 19, and 20. For long-term forecasting and imputation tasks, the results shown for each dataset represent averages across all prediction horizons and mask ratios, respectively.

Table 18: Ablation study results for Long-term Forecasting across different datasets, evaluated with different XCTFormer variations.

| | ETTh1 | | ETTh2 | | ETTm1 | | ETTm2 | | ECL$^p$ | | Traffic$^p$ | | Weather | | XCTFormer vs Others |
|---|---|---|---|---|---|---|---|---|---|---|---|---|---|---|---|
| | MSE | MAE | MSE | MAE | MSE | MAE | MSE | MAE | MSE | MAE | MSE | MAE | MSE | MAE | (%) |
| XCTFormer (Original) | 0.450 | 0.436 | **0.369** | **0.396** | **0.369** | **0.392** | **0.270** | **0.319** | **0.166** | **0.263** | **0.435** | 0.287 | **0.237** | **0.267** | - |
| W/o mask$^†$ | 0.453 | 0.440 | 0.383 | 0.405 | 0.379 | 0.394 | 0.282 | 0.326 | | | | | 0.256 | 0.280 | +2.7% |
| Original softmax activation | **0.443** | **0.435** | 0.390 | 0.409 | 0.410 | 0.413 | 0.280 | 0.326 | 0.224 | 0.317 | 0.520 | 0.368 | 0.247 | 0.278 | +8.0% |
| Vanilla transformer | 0.452 | 0.439 | 0.397 | 0.412 | 0.385 | 0.398 | 0.285 | 0.330 | 0.224 | 0.317 | 0.520 | 0.368 | 0.263 | 0.284 | +8.3% |
| Sequence modeling | 0.449 | 0.436 | 0.380 | 0.404 | 0.375 | 0.393 | 0.281 | 0.327 | 0.195 | 0.280 | 0.450 | **0.284** | 0.256 | 0.279 | +2.8% |
| Channel modeling | 0.461 | 0.450 | 0.388 | 0.409 | 0.378 | 0.393 | 0.278 | 0.321 | **0.166** | **0.263** | 0.476 | 0.329 | 0.239 | 0.269 | +3.4% |

$^p$ DeCoP was enabled for XCTFormer on this dataset. $^†$ W/o mask results exclude DeCoP datasets where the learnable mask is not applicable; % is recomputed over remaining datasets.

Table 19: Ablation study results for Imputation across different datasets, evaluated with different XCTFormer variations.

| | ETTm1 | | ETTm2 | | ETTh1 | | ETTh2 | | Weather | | ECL$^p$ | | XCTFormer vs Others |
|---|---|---|---|---|---|---|---|---|---|---|---|---|---|
| | MSE | MAE | MSE | MAE | MSE | MAE | MSE | MAE | MSE | MAE | MSE | MAE | (%) |
| XCTFormer (Original) | **0.029** | **0.113** | **0.024** | **0.092** | 0.087 | 0.201 | **0.046** | **0.144** | **0.031** | 0.050 | **0.046** | 0.141 | - |
| W/o mask$^\dagger$ | 0.041 | 0.132 | 0.029 | 0.100 | 0.092 | 0.206 | 0.064 | 0.169 | 0.031 | 0.047 | | | +13.9% |
| Original softmax activation | 0.032 | 0.117 | 0.027 | 0.099 | **0.078** | **0.192** | 0.065 | 0.178 | 0.040 | 0.075 | 0.077 | 0.196 | +14.7% |
| Vanilla transformer | 0.043 | 0.137 | 0.033 | 0.113 | 0.094 | 0.208 | 0.079 | 0.193 | 0.031 | 0.048 | 0.077 | 0.196 | +19.9% |
| Sequence modeling | 0.040 | 0.130 | 0.029 | 0.101 | 0.089 | 0.201 | 0.081 | 0.166 | **0.031** | **0.047** | **0.046** | **0.139** | +8.7% |
| Channel modeling | 0.065 | 0.174 | 0.044 | 0.136 | 0.203 | 0.305 | 0.060 | 0.167 | 0.041 | 0.072 | 0.075 | 0.191 | +34.6% |

$^p$ DeCoP was enabled for XCTFormer on this dataset. $^\dagger$ W/o mask results exclude DeCoP datasets where the learnable mask is not applicable; % is recomputed over remaining datasets.

Table 20: Ablation study results for Anomaly Detection across different datasets, evaluated with different XCTFormer variations.

| | PSM | SWaT | MSL | SMAP | SMD | XCTFormer vs Others |
|---|---|---|---|---|---|---|
| | F-Score | F-Score | F-Score | F-Score | F-Score | (%) |
| XCTFormer (Original) | **95.3** | 92.6 | 79.0 | **86.7** | 84.2 | - |
| W/o mask | 95.3 | 88.2 | 72.3 | 66.7 | 83.0 | +8.0% |
| Original softmax activation | 95.3 | 90.2 | 69.0 | 68.6 | **84.3** | +7.5% |
| Vanilla transformer | 95.3 | 93.1 | 71.8 | 66.9 | 83.2 | +6.7% |
| Sequence modeling | 95.3 | **93.5** | **79.4** | 67.5 | 83.9 | +4.4% |
| Channel modeling | 92.9 | 92.5 | 76.9 | 66.5 | 82.8 | +6.4% |

### E.3 Robustness Across Random Seeds: Complete Analysis

**Coefficient of Variation (CV) for a Single Metric** We quantify run-to-run stability using the coefficient of variation, a unitless measure of dispersion relative to the mean (Reed et al., 2002). For a metric with mean $\mu$ and standard deviation $\sigma$ across five seeds (2021 to 2025), we compute:

$$\text{CV}(\%) = 100 \cdot \frac{\sigma}{|\mu|}.$$

The coefficient of variation tells us how much results vary around their mean *relative* to the mean itself. Since CV is unitless, it enables comparison across datasets and metrics: smaller values indicate greater stability, while larger values indicate greater variability.

**Confidence score mapping** For intuitive interpretation, we report a complementary confidence score:

$$\text{Conf}(\%) = 100 - \text{CV}(\%).$$

This confidence score inverts the scale so that lower variability corresponds to higher confidence. For example, if CV = 3.2%, then Conf = 96.8%, indicating that repeated runs with identical setups produce very similar results.

**Full Seed Analysis Results** To enhance readability, we include only the averaged analysis table for all time-series tasks in the main paper, while the complete results are provided in Tables 21, 22 and 23. The confidence score presented for each dataset represents the average confidence score across all of its metrics.

Table 21: Standard deviation for XCTFormer on forecasting datasets, evaluated across five seeds (2021-2025). Results averaged over the four prediction lengths $\{96, 192, 336, 720\}$.

| Model | XCTFormer (Ours) | | Confidence Score |
|---|---|---|---|
| Dataset | MSE | MAE | Score % |
| ETTh1 | $0.449 \pm 0.002$ | $0.436 \pm 0.001$ | 99.7% |
| ETTh2 | $0.374 \pm 0.007$ | $0.399 \pm 0.004$ | 98.5% |
| ETTm1 | $0.371 \pm 0.003$ | $0.393 \pm 0.002$ | 99.3% |
| ETTm2 | $0.271 \pm 0.001$ | $0.319 \pm 0.001$ | 99.6% |
| $ECL^p$ | $0.176 \pm 0.007$ | $0.270 \pm 0.007$ | 96.6% |
| $Traffic^p$ | $0.435 \pm 0.001$ | $0.287 \pm 0.001$ | 99.7% |
| Weather | $0.237 \pm 0.001$ | $0.267 \pm 9.81e - 04$ | 99.5% |

$^p$ DeCoP was enabled for XCTFormer on this dataset.

Table 22: Results of the imputation task across datasets, evaluated across five seeds (2021-2025). We randomly mask $\{12.5\%, 25\%, 37.5\%, 50\%\}$ of the time points; the final results are averaged across these four masking ratios.

| Model | XCTFormer (Ours) | | Confidence Score |
|---|---|---|---|
| Dataset | MSE | MAE | Score % |
| ETTh1 | $0.090 \pm 0.002$ | $0.204 \pm 0.003$ | 98.0 % |
| ETTh2 | $0.052 \pm 0.013$ | $0.153 \pm 0.020$ | 80.9 % |
| ETTm1 | $0.031 \pm 0.004$ | $0.116 \pm 0.007$ | 90.2 % |
| ETTm2 | $0.026 \pm 0.003$ | $0.097 \pm 0.007$ | 90.8 % |
| ETT(Avg) | $0.049 \pm 0.006$ | $0.143 \pm 0.009$ | 91.1 % |
| $ECL^p$ | $0.051 \pm 0.008$ | $0.149 \pm 0.014$ | 87.6 % |
| Weather | $0.031 \pm 5.26e - 04$ | $0.049 \pm 0.002$ | 96.9 % |

$^p$ DeCoP was enabled for XCTFormer on this dataset.

Table 23: Results for the anomaly detection task (P, R, and F1 are precision, recall, and F1-score in %), evaluated across five seeds (2021-2025).

| Model | XCTFormer (Ours) | | | Confidence Score |
|---|---|---|---|---|
| Dataset | Precision | Recall | F1 | Score % |
| MSL | $87.84 \pm 1.73$ | $66.66 \pm 3.92$ | $75.77 \pm 3.18$ | 96.0% |
| PSM | $98.31 \pm 0.09$ | $93.05 \pm 0.57$ | $95.61 \pm 0.34$ | 99.6% |
| SMAP | $91.81 \pm 1.78$ | $64.59 \pm 14.53$ | $75.22 \pm 10.43$ | 87.2% |
| SMD | $87.01 \pm 0.25$ | $81.99 \pm 1.15$ | $84.42 \pm 0.69$ | 99.2% |
| SWaT | $91.94 \pm 0.49$ | $92.01 \pm 1.32$ | $91.98 \pm 0.86$ | 99.0% |

### E.4 Long-Term Forecasting Results

To improve readability, we present only the averaged table for long-term forecasting in the main paper and provide the full results here.

### E.5 Imputation Full Results

To improve readability, we present only the averaged table for imputation in the main paper and provide the full per-mask-ratio results for XCTFormer in Table 25.

Table 24: Long-term forecasting results comparison across multiple datasets and horizons. Synthetic[†] is evaluated under multivariate-to-single (MS) setting while all other datasets use multivariate (M). We compare extensive competitive models under different prediction lengths. *Avg* is averaged from all four prediction lengths, that $\{96, 192, 336, 720\}$.

| Dataset | Horizon | XCTFormer (Ours) MSE | MAE | TimeMixer++ (ICLR 2025) MSE | MAE | MTLinear (AISTATS 2025) MSE | MAE | Leddam (ICML 2024) MSE | MAE | TimeMixer (ICLR 2024) MSE | MAE | iTransformer (ICLR 2024) MSE | MAE | PatchTST (ICLR 2023) MSE | MAE | Crossformer (ICLR 2023) MSE | MAE | TiDE (TMLR 2023) MSE | MAE | TimesNet (ICLR 2023) MSE | MAE | DLinear (AAAI 2023) MSE | MAE | SCINet (NeurIPS 2022) MSE | MAE | FEDformer (ICML 2022) MSE | MAE | Autoformer (NeurIPS 2021) MSE | MAE |
|---|---|---|---|---|---|---|---|---|---|---|---|---|---|---|---|---|---|---|---|---|---|---|---|---|---|---|---|---|---|
| ETTm1 | 96 | **0.302** | **0.350** | **0.310** | **0.334** | 0.337 | 0.363 | 0.319 | 0.359 | 0.320 | 0.357 | 0.334 | 0.368 | 0.352 | 0.374 | 0.404 | 0.426 | 0.364 | 0.387 | 0.338 | 0.375 | 0.346 | 0.374 | 0.418 | 0.438 | 0.379 | 0.419 | 0.505 | 0.475 |
| ETTm1 | 192 | **0.354** | 0.382 | **0.348** | **0.362** | 0.379 | 0.387 | 0.369 | 0.383 | 0.361 | **0.381** | 0.390 | 0.393 | 0.374 | 0.387 | 0.450 | 0.451 | 0.398 | 0.404 | 0.374 | 0.387 | 0.382 | 0.391 | 0.439 | 0.450 | 0.426 | 0.441 | 0.553 | 0.496 |
| ETTm1 | 336 | **0.385** | 0.404 | **0.376** | **0.391** | 0.412 | 0.409 | 0.394 | **0.402** | 0.390 | 0.404 | 0.426 | 0.420 | 0.421 | 0.414 | 0.532 | 0.515 | 0.428 | 0.425 | 0.410 | 0.411 | 0.415 | 0.415 | 0.490 | 0.485 | 0.445 | 0.459 | 0.621 | 0.537 |
| ETTm1 | 720 | **0.435** | **0.433** | **0.440** | **0.423** | 0.468 | 0.443 | 0.460 | 0.442 | 0.454 | 0.441 | 0.491 | 0.459 | 0.462 | 0.449 | 0.666 | 0.589 | 0.487 | 0.461 | 0.478 | 0.450 | 0.473 | 0.451 | 0.595 | 0.550 | 0.543 | 0.490 | 0.671 | 0.561 |
| ETTm1 | Avg | **0.369** | **0.392** | **0.368** | **0.378** | 0.399 | 0.401 | 0.385 | 0.397 | 0.381 | 0.396 | 0.410 | 0.410 | 0.402 | 0.406 | 0.513 | 0.495 | 0.419 | 0.419 | 0.400 | 0.406 | 0.404 | 0.408 | 0.485 | 0.481 | 0.448 | 0.452 | 0.588 | 0.517 |
| ETTm2 | 96 | **0.168** | **0.252** | **0.170** | **0.245** | 0.175 | 0.254 | 0.176 | 0.257 | 0.175 | 0.258 | 0.180 | 0.264 | 0.183 | 0.270 | 0.287 | 0.366 | 0.207 | 0.305 | 0.187 | 0.267 | 0.193 | 0.293 | 0.286 | 0.377 | 0.203 | 0.287 | 0.255 | 0.339 |
| ETTm2 | 192 | **0.232** | **0.295** | **0.229** | **0.291** | 0.240 | 0.296 | 0.243 | 0.303 | 0.237 | 0.299 | 0.250 | 0.309 | 0.255 | 0.314 | 0.414 | 0.492 | 0.290 | 0.364 | 0.249 | 0.309 | 0.284 | 0.361 | 0.399 | 0.445 | 0.269 | 0.328 | 0.281 | 0.340 |
| ETTm2 | 336 | **0.289** | **0.332** | 0.303 | 0.343 | 0.301 | **0.335** | 0.303 | 0.341 | 0.298 | 0.340 | 0.311 | 0.348 | 0.309 | 0.347 | 0.597 | 0.542 | 0.377 | 0.422 | 0.321 | 0.351 | 0.382 | 0.429 | 0.637 | 0.591 | 0.325 | 0.366 | 0.339 | 0.372 |
| ETTm2 | 720 | **0.391** | **0.395** | **0.373** | 0.399 | 0.402 | **0.393** | 0.400 | 0.398 | **0.391** | 0.396 | 0.412 | 0.407 | 0.412 | 0.404 | 1.730 | 1.042 | 0.558 | 0.524 | 0.408 | 0.403 | 0.558 | 0.525 | 0.960 | 0.735 | 0.421 | 0.415 | 0.433 | 0.432 |
| ETTm2 | Avg | **0.270** | **0.319** | **0.269** | **0.320** | 0.279 | **0.320** | 0.280 | 0.325 | 0.275 | 0.323 | 0.288 | 0.332 | 0.290 | 0.334 | 0.757 | 0.611 | 0.358 | 0.404 | 0.291 | 0.333 | 0.354 | 0.402 | 0.571 | 0.537 | 0.304 | 0.349 | 0.327 | 0.371 |
| ETTh1 | 96 | 0.389 | 0.400 | **0.361** | 0.403 | 0.386 | **0.393** | 0.377 | **0.394** | 0.375 | 0.400 | 0.386 | 0.405 | 0.460 | 0.447 | 0.423 | 0.448 | 0.479 | 0.464 | 0.384 | 0.402 | 0.397 | 0.412 | 0.654 | 0.599 | 0.395 | 0.424 | 0.449 | 0.459 |
| ETTh1 | 192 | 0.440 | 0.429 | **0.416** | 0.441 | 0.439 | **0.421** | 0.424 | 0.422 | 0.429 | **0.421** | 0.441 | 0.512 | 0.477 | 0.428 | 0.470 | 0.473 | 0.525 | 0.492 | 0.436 | 0.429 | 0.446 | 0.441 | 0.719 | 0.631 | 0.469 | 0.470 | 0.500 | 0.482 |
| ETTh1 | 336 | 0.479 | 0.447 | **0.430** | **0.434** | 0.476 | 0.441 | 0.459 | 0.442 | 0.484 | 0.458 | 0.487 | 0.458 | 0.546 | 0.496 | 0.570 | 0.546 | 0.565 | 0.515 | 0.491 | 0.469 | 0.489 | 0.467 | 0.778 | 0.659 | 0.530 | 0.499 | 0.521 | 0.496 |
| ETTh1 | 720 | 0.490 | 0.468 | **0.467** | **0.451** | 0.472 | 0.460 | 0.463 | 0.459 | 0.498 | 0.482 | 0.503 | 0.491 | 0.544 | 0.517 | 0.653 | 0.621 | 0.594 | 0.558 | 0.521 | 0.500 | 0.513 | 0.510 | 0.836 | 0.699 | 0.598 | 0.544 | 0.514 | 0.512 |
| ETTh1 | Avg | 0.450 | 0.436 | **0.418** | 0.432 | 0.443 | **0.429** | 0.431 | 0.429 | 0.447 | 0.440 | 0.454 | 0.467 | 0.507 | 0.472 | 0.529 | 0.522 | 0.541 | 0.507 | 0.458 | 0.450 | 0.461 | 0.458 | 0.747 | 0.647 | 0.498 | 0.484 | 0.496 | 0.487 |
| ETTh2 | 96 | 0.295 | 0.342 | **0.276** | **0.328** | **0.288** | 0.336 | 0.292 | 0.343 | 0.289 | 0.341 | 0.297 | 0.349 | 0.308 | 0.355 | 0.745 | 0.584 | 0.400 | 0.440 | 0.340 | 0.374 | 0.340 | 0.394 | 0.707 | 0.621 | 0.358 | 0.397 | 0.346 | 0.388 |
| ETTh2 | 192 | 0.370 | 0.393 | **0.342** | **0.379** | 0.375 | **0.388** | 0.367 | 0.389 | 0.372 | 0.392 | 0.380 | 0.400 | 0.393 | 0.405 | 0.877 | 0.656 | 0.528 | 0.509 | 0.402 | 0.414 | 0.482 | 0.479 | 0.860 | 0.689 | 0.429 | 0.439 | 0.456 | 0.452 |
| ETTh2 | 336 | 0.402 | 0.417 | **0.346** | **0.398** | 0.412 | 0.423 | 0.412 | 0.424 | 0.386 | **0.414** | 0.428 | 0.432 | 0.427 | 0.436 | 1.043 | 0.731 | 0.643 | 0.571 | 0.452 | 0.452 | 0.591 | 0.541 | 1.000 | 0.744 | 0.496 | 0.487 | 0.482 | 0.486 |
| ETTh2 | 720 | **0.411** | **0.433** | **0.392** | **0.415** | 0.418 | 0.440 | 0.419 | 0.438 | 0.412 | 0.434 | 0.427 | 0.445 | 0.436 | 0.450 | 1.104 | 0.763 | 0.874 | 0.679 | 0.462 | 0.468 | 0.839 | 0.661 | 1.249 | 0.838 | 0.463 | 0.474 | 0.515 | 0.511 |
| ETTh2 | Avg | 0.369 | 0.396 | **0.339** | **0.380** | 0.373 | 0.397 | 0.372 | 0.398 | **0.365** | **0.395** | 0.383 | 0.407 | 0.391 | 0.411 | 0.942 | 0.683 | 0.611 | 0.550 | 0.414 | 0.427 | 0.563 | 0.519 | 0.954 | 0.723 | 0.436 | 0.449 | 0.450 | 0.459 |
| Weather | 96 | **0.153** | **0.199** | **0.155** | 0.205 | 0.159 | 0.211 | 0.156 | **0.202** | 0.163 | 0.209 | 0.174 | 0.214 | 0.186 | 0.227 | 0.195 | 0.271 | 0.202 | 0.261 | 0.172 | 0.220 | 0.195 | 0.252 | 0.221 | 0.306 | 0.217 | 0.296 | 0.266 | 0.336 |
| Weather | 192 | **0.199** | **0.242** | **0.201** | **0.245** | 0.202 | 0.252 | 0.207 | 0.250 | 0.208 | 0.250 | 0.221 | 0.254 | 0.234 | 0.265 | 0.209 | 0.277 | 0.242 | 0.298 | 0.219 | 0.261 | 0.237 | 0.295 | 0.261 | 0.340 | 0.276 | 0.336 | 0.307 | 0.367 |
| Weather | 336 | 0.257 | 0.286 | **0.237** | **0.265** | 0.259 | 0.294 | 0.262 | 0.291 | **0.251** | 0.287 | 0.278 | 0.296 | 0.284 | 0.301 | 0.273 | 0.332 | 0.287 | 0.335 | 0.280 | 0.306 | 0.282 | 0.331 | 0.309 | 0.378 | 0.339 | 0.380 | 0.359 | 0.395 |
| Weather | 720 | 0.339 | **0.340** | **0.312** | **0.334** | **0.332** | 0.346 | 0.343 | 0.343 | 0.339 | 0.341 | 0.358 | 0.347 | 0.356 | 0.349 | 0.379 | 0.401 | 0.351 | 0.386 | 0.365 | 0.359 | 0.345 | 0.382 | 0.377 | 0.427 | 0.403 | 0.428 | 0.419 | 0.428 |
| Weather | Avg | **0.237** | **0.267** | **0.226** | **0.262** | 0.238 | 0.276 | 0.242 | 0.272 | 0.240 | 0.272 | 0.258 | 0.278 | 0.265 | 0.285 | 0.264 | 0.320 | 0.270 | 0.320 | 0.259 | 0.286 | 0.265 | 0.315 | 0.292 | 0.363 | 0.309 | 0.360 | 0.338 | 0.382 |
| ECL[P] | 96 | **0.138** | 0.237 | **0.135** | **0.222** | 0.183 | 0.265 | 0.141 | **0.235** | 0.153 | 0.247 | 0.148 | 0.240 | 0.190 | 0.296 | 0.219 | 0.314 | 0.237 | 0.329 | 0.168 | 0.272 | 0.210 | 0.302 | 0.247 | 0.345 | 0.193 | 0.308 | 0.201 | 0.317 |
| ECL[P] | 192 | 0.164 | 0.261 | **0.147** | **0.235** | 0.183 | 0.268 | **0.159** | **0.252** | 0.166 | 0.256 | 0.162 | 0.253 | 0.199 | 0.304 | 0.231 | 0.322 | 0.236 | 0.330 | 0.184 | 0.322 | 0.210 | 0.305 | 0.257 | 0.355 | 0.201 | 0.315 | 0.222 | 0.334 |
| ECL[P] | 336 | 0.170 | 0.266 | **0.164** | **0.245** | 0.196 | 0.283 | 0.173 | 0.268 | 0.185 | 0.277 | 0.178 | 0.269 | 0.217 | 0.319 | 0.246 | 0.337 | 0.249 | 0.344 | 0.198 | 0.300 | 0.223 | 0.319 | 0.269 | 0.369 | 0.214 | 0.329 | 0.231 | 0.443 |
| ECL[P] | 720 | **0.190** | **0.286** | 0.212 | 0.310 | 0.231 | 0.317 | **0.201** | 0.295 | 0.225 | 0.310 | 0.225 | 0.317 | 0.258 | 0.352 | 0.280 | 0.363 | 0.284 | 0.373 | 0.220 | 0.320 | 0.258 | 0.350 | 0.299 | 0.390 | 0.246 | 0.355 | 0.254 | 0.361 |
| ECL[P] | Avg | **0.166** | **0.263** | **0.165** | **0.253** | 0.198 | 0.283 | 0.168 | **0.263** | 0.182 | 0.273 | 0.178 | 0.270 | 0.216 | 0.318 | 0.244 | 0.334 | 0.252 | 0.344 | 0.193 | 0.304 | 0.225 | 0.319 | 0.268 | 0.365 | 0.213 | 0.327 | 0.227 | 0.364 |
| Traffic[P] | 96 | 0.402 | 0.269 | **0.392** | **0.253** | 0.647 | 0.383 | 0.426 | 0.276 | 0.462 | 0.285 | **0.395** | **0.268** | 0.526 | 0.347 | 0.644 | 0.429 | 0.805 | 0.493 | 0.593 | 0.321 | 0.650 | 0.396 | 0.788 | 0.499 | 0.587 | 0.366 | 0.613 | 0.388 |
| Traffic[P] | 192 | 0.424 | 0.281 | **0.402** | **0.258** | 0.594 | 0.359 | 0.458 | 0.289 | 0.473 | 0.296 | **0.417** | **0.276** | 0.522 | 0.332 | 0.665 | 0.431 | 0.756 | 0.474 | 0.617 | 0.336 | 0.598 | 0.370 | 0.789 | 0.505 | 0.604 | 0.373 | 0.616 | 0.382 |
| Traffic[P] | 336 | 0.444 | 0.291 | **0.428** | **0.263** | 0.601 | 0.362 | 0.486 | 0.297 | 0.498 | 0.296 | **0.433** | **0.283** | 0.517 | 0.334 | 0.674 | 0.420 | 0.762 | 0.477 | 0.629 | 0.336 | 0.605 | 0.373 | 0.797 | 0.508 | 0.621 | 0.383 | 0.622 | 0.337 |
| Traffic[P] | 720 | 0.472 | 0.307 | **0.441** | **0.282** | 0.640 | 0.382 | 0.498 | 0.313 | 0.506 | 0.313 | **0.467** | **0.302** | 0.552 | 0.352 | 0.683 | 0.424 | 0.719 | 0.449 | 0.640 | 0.350 | 0.645 | 0.394 | 0.841 | 0.523 | 0.626 | 0.382 | 0.660 | 0.408 |
| Traffic[P] | Avg | 0.435 | 0.287 | **0.416** | **0.264** | 0.621 | 0.372 | 0.467 | 0.294 | 0.485 | 0.297 | **0.428** | **0.282** | 0.529 | 0.341 | 0.667 | 0.426 | 0.760 | 0.473 | 0.620 | 0.336 | 0.625 | 0.383 | 0.804 | 0.509 | 0.609 | 0.376 | 0.628 | 0.379 |
| Synthetic[†] | 96 | 0.038 | 0.151 | – | – | 0.670 | 0.640 | 0.043 | 0.159 | **0.025** | **0.123** | 0.217 | 0.362 | 0.222 | 0.365 | **0.018** | **0.106** | 0.836 | 0.746 | 0.554 | 0.597 | 0.516 | 0.577 | 0.130 | 0.277 | 1.537 | 1.006 | 0.573 | 0.605 |
| Synthetic[†] | 192 | **0.043** | **0.161** | – | – | 1.189 | 0.881 | 0.094 | 0.235 | 0.070 | 0.204 | 0.236 | 0.375 | 0.319 | 0.442 | **0.026** | **0.126** | 1.290 | 0.934 | 0.471 | 0.538 | 0.496 | 0.572 | 0.092 | 0.235 | 2.422 | 1.282 | 1.753 | 1.094 |
| Synthetic[†] | 336 | **0.045** | **0.165** | – | – | 1.388 | 0.975 | 0.095 | 0.234 | 0.053 | 0.178 | 0.282 | 0.411 | 0.319 | 0.441 | **0.027** | **0.129** | 1.476 | 1.016 | 0.404 | 0.499 | 0.482 | 0.565 | 0.085 | 0.225 | 4.065 | 1.663 | 1.743 | 1.079 |
| Synthetic[†] | 720 | **0.035** | **0.144** | – | – | 1.149 | 0.845 | 0.082 | 0.216 | 0.133 | 0.281 | 0.282 | 0.413 | 0.249 | 0.384 | **0.031** | **0.137** | 1.182 | 0.875 | 0.350 | 0.459 | 0.428 | 0.523 | 0.079 | 0.218 | 1.879 | 1.110 | 1.606 | 1.019 |
| Synthetic[†] | Avg | **0.040** | **0.155** | – | – | 1.099 | 0.835 | 0.079 | 0.211 | 0.070 | 0.197 | 0.254 | 0.390 | 0.277 | 0.408 | **0.025** | **0.124** | 1.196 | 0.893 | 0.445 | 0.523 | 0.480 | 0.559 | 0.097 | 0.239 | 2.476 | 1.265 | 1.419 | 0.949 |
| 1st Count | | 7 | 4 | 20 | 21 | 0 | 3 | 1 | 0 | 0 | 1 | 0 | 0 | 0 | 0 | 0 | 0 | 4 | 4 | 0 | 0 | 0 | 0 | 0 | 0 | 0 | 0 | 0 | 0 |
| Avg FLOPs | | 3.33E+09 | | – | | 7.59E+06 | | 8.64E+08 | | 8.72E+08 | | 1.34E+09 | | 8.28E+08 | | 2.31E+09 | | 1.31E+09 | | 1.23E+11 | | 1.01E+07 | | 1.41E+09 | | 2.01E+09 | | 5.26E+07 | |
| Avg Params | | 5.08E+06 | | – | | 7.71E+05 | | 2.85E+06 | | 1.34E+05 | | 2.30E+06 | | 7.10E+05 | | 7.87E+06 | | 8.53E+06 | | 5.74E+07 | | 6.50E+04 | | 7.14E+07 | | 1.69E+07 | | 2.15E+05 | |

[†] Evaluated under multivariate-to-single (MS) setting; hyperparameters chosen identical to ETTm1 for all models.
[1] Reported MTLinear results reflect the per-dataset best of MTNLinear and MTDLinear.
[P] DeCoP was enabled for XCTFormer on this dataset.

Table 25: Full imputation results for XCTFormer across all datasets and mask ratios. We randomly mask $\{12.5\%, 25\%, 37.5\%, 50\%\}$ of the time points in a time series of length 1024.

| Dataset | Mask 12.5% | | Mask 25% | | Mask 37.5% | | Mask 50% | | Avg | |
|---|---|---|---|---|---|---|---|---|---|---|
| | MSE | MAE | MSE | MAE | MSE | MAE | MSE | MAE | MSE | MAE |
| ETTh1 | 0.069 | 0.179 | 0.080 | 0.192 | 0.093 | 0.208 | 0.105 | 0.221 | 0.087 | 0.200 |
| ETTh2 | 0.045 | 0.143 | 0.065 | 0.168 | 0.068 | 0.181 | 0.060 | 0.165 | 0.059 | 0.164 |
| ETTm1 | 0.019 | 0.092 | 0.026 | 0.107 | 0.034 | 0.122 | 0.035 | 0.124 | 0.028 | 0.111 |
| ETTm2 | 0.020 | 0.083 | 0.022 | 0.089 | 0.025 | 0.095 | 0.028 | 0.101 | 0.024 | 0.092 |
| ETT(Avg) | 0.038 | 0.124 | 0.048 | 0.139 | 0.055 | 0.152 | 0.057 | 0.153 | 0.050 | 0.141 |
| ECL[p] | 0.037 | 0.124 | 0.043 | 0.135 | 0.049 | 0.146 | 0.056 | 0.157 | 0.046 | 0.140 |
| Weather | 0.029 | 0.047 | 0.031 | 0.048 | 0.033 | 0.049 | 0.036 | 0.052 | 0.032 | 0.049 |

[p] DeCoP was enabled for XCTFormer on this dataset.

Table 26: Full results for the anomaly detection task. The P, R, and F1 represent the precision, recall, and F1-score, (%) respectively. F1-score is the harmonic mean of precision and recall. A higher value of P, R and F1 indicates a better performance. Red indicates highest F1 score, blue indicates second highest F1 score.

| Datasets | | SMD | | | MSL | | | SMAP | | | SWaT | | | PSM | | | Avg F1 |
|---|---|---|---|---|---|---|---|---|---|---|---|---|---|---|---|---|---|---|
| Metrics | | P | R | F1 | P | R | F1 | P | R | F1 | P | R | F1 | P | R | F1 | (%) |
| LSTM | (1997) | 78.52 | 65.47 | 71.41 | 78.04 | 86.22 | 81.93 | 91.06 | 57.49 | 70.48 | 78.06 | 91.72 | 84.34 | 69.24 | 99.53 | 81.67 | 77.97 |
| Transformer | (2017) | 83.58 | 76.13 | 79.56 | 71.57 | 87.37 | 78.68 | 89.37 | 57.12 | 69.70 | 68.84 | 96.53 | 80.37 | 62.75 | 96.56 | 76.07 | 76.88 |
| LogTrans | (2019a) | 83.46 | 70.13 | 76.21 | 73.05 | 87.37 | 79.57 | 89.15 | 57.59 | 69.97 | 68.67 | 97.32 | 80.52 | 63.06 | 98.00 | 76.74 | 76.60 |
| TCN | (2019) | 84.06 | 79.07 | 81.49 | 75.11 | 82.44 | 78.60 | 86.90 | 59.23 | 70.45 | 76.59 | 95.71 | 85.09 | 54.59 | 99.77 | 70.57 | 77.24 |
| Reformer | (2020) | 82.58 | 69.24 | 75.32 | 85.51 | 83.31 | 84.40 | 90.91 | 57.44 | 70.40 | 72.50 | 96.53 | 82.80 | 59.93 | 95.38 | 73.61 | 77.31 |
| Informer | (2021) | 86.60 | 77.23 | 81.65 | 81.77 | 86.48 | 84.06 | 90.11 | 57.13 | 69.92 | 70.29 | 96.75 | 81.43 | 64.27 | 96.33 | 77.10 | 78.83 |
| Anomaly* | (2022) | 88.91 | 82.23 | 85.49 | 79.61 | 87.37 | 83.31 | 91.85 | 58.11 | 71.18 | 72.51 | 97.32 | 83.10 | 68.35 | 94.72 | 79.40 | 80.50 |
| Pyraformer | (2022b) | 85.61 | 80.61 | 83.04 | 83.81 | 85.93 | 84.86 | 92.54 | 57.71 | 71.09 | 87.92 | 96.00 | 91.78 | 71.67 | 96.02 | 82.08 | 82.57 |
| Autoformer | (2021) | 88.06 | 82.35 | 85.11 | 77.27 | 80.92 | 79.05 | 90.40 | 58.62 | 71.12 | 89.85 | 95.81 | 92.74 | 99.08 | 88.15 | 93.29 | 84.26 |
| LSSL | (2022) | 78.51 | 65.32 | 71.31 | 77.55 | 88.18 | 82.53 | 89.43 | 53.43 | 66.90 | 79.05 | 93.72 | 85.76 | 66.02 | 92.93 | 77.20 | 76.74 |
| Stationary | (2022c) | 88.33 | 81.21 | 84.62 | 68.55 | 89.14 | 77.50 | 89.37 | 59.02 | 71.09 | 68.03 | 96.75 | 79.88 | 97.82 | 96.76 | 97.29 | 82.08 |
| DLinear | (2023) | 83.62 | 71.52 | 77.10 | 84.34 | 85.42 | 84.88 | 92.32 | 55.41 | 69.26 | 80.91 | 95.30 | 87.52 | 98.28 | 89.26 | 93.55 | 82.46 |
| ETSformer | (2022) | 87.44 | 79.23 | 83.13 | 85.13 | 84.93 | 85.03 | 92.25 | 55.75 | 69.50 | 90.02 | 80.36 | 84.91 | 99.31 | 85.28 | 91.76 | 82.87 |
| LightTS | (2022) | 87.10 | 78.42 | 82.53 | 82.40 | 75.78 | 78.95 | 92.58 | 55.27 | 69.21 | 91.98 | 94.72 | 93.33 | 98.37 | 95.97 | 97.15 | 84.23 |
| FEDformer | (2022) | 87.95 | 82.39 | 85.08 | 77.14 | 80.07 | 78.57 | 90.47 | 58.10 | 70.76 | 90.17 | 96.42 | 93.19 | 97.31 | 97.16 | 97.23 | 84.97 |
| TimesNet | (2023) | 88.66 | 83.14 | 85.81 | 83.92 | 86.42 | 85.15 | 92.52 | 58.29 | 71.52 | 86.76 | 97.32 | 91.74 | 98.19 | 96.76 | 97.47 | 86.34 |
| TiDE | (2023) | 76.00 | 63.00 | 68.91 | 84.00 | 60.00 | 70.18 | 88.00 | 50.00 | 64.00 | 98.00 | 63.00 | 76.73 | 93.00 | 92.00 | 92.50 | 74.46 |
| iTransformer | (2024) | 78.45 | 65.10 | 71.15 | 86.15 | 62.65 | 72.54 | 90.67 | 52.96 | 66.87 | 99.96 | 65.55 | 79.18 | 95.65 | 94.69 | 95.17 | 76.98 |
| TimesMixer++ | (2025) | 88.59 | 84.50 | 86.50 | 89.73 | 82.23 | 85.82 | 93.47 | 60.02 | 73.10 | 92.96 | 94.33 | 94.64 | 98.33 | 96.90 | 97.60 | 87.47 |
| XCTFormer | (Ours) | 86.94 | 81.64 | 84.21 | 89.46 | 70.81 | 79.05 | 93.79 | 80.57 | 86.68 | 92.25 | 92.96 | 92.60 | 98.26 | 92.52 | 95.30 | 87.57 |

* The original paper of Anomaly Transformer (Xu et al., 2022) adopts the temporal association and reconstruction error as a joint anomaly criterion. For fair comparisons, we only use reconstruction error here.

### E.6 Anomaly Detection Full Results

To improve readability, we present only the averaged plot for anomaly detection in the main paper and provide the full results in Table 26.

### E.7 Computational Complexity

We report computational complexity in terms of floating-point operations (FLOPs) and trainable parameter counts. For a detailed description of the measurement methodology, see Section B.6. Tables 27 and 28 present the full per-dataset results.

Table 27: FLOPs (floating-point operations) per dataset, averaged over prediction lengths $\{96, 192, 336, 720\}$. Model and dataset ordering follows Table 24. See Section B.6 for measurement methodology.

| Model | ETTm1 | ETTm2 | ETTh1 | ETTh2 | Weather | ECL | Traffic | Synth.[†] | Avg |
|---|---|---|---|---|---|---|---|---|---|
| MTLinear | 3.4E+05 | 3.4E+05 | 3.4E+05 | 3.4E+05 | 1.0E+06 | 1.6E+07 | 4.2E+07 | 3.4E+05 | 7.59E+06 |
| DLinear | 4.5E+05 | 4.5E+05 | 4.5E+05 | 4.5E+05 | 1.4E+06 | 2.1E+07 | 5.6E+07 | 4.5E+05 | 1.01E+07 |
| Autoformer | 2.4E+07 | 2.4E+07 | 2.4E+07 | 2.4E+07 | 2.7E+07 | 8.4E+07 | 1.9E+08 | 2.4E+07 | 5.26E+07 |
| PatchTST | 3.8E+07 | 3.8E+07 | 1.9E+06 | 1.9E+06 | 1.1E+08 | 1.7E+09 | 4.7E+09 | 3.8E+07 | 8.28E+08 |
| Leddam | 3.5E+07 | 4.7E+07 | 4.7E+07 | 1.2E+07 | 3.5E+07 | 1.6E+09 | 5.1E+09 | 3.5E+07 | 8.64E+08 |
| TimeMixer | 1.6E+07 | 3.2E+07 | 1.6E+07 | 1.6E+07 | 6.0E+07 | 9.2E+08 | 5.9E+09 | 1.6E+07 | 8.72E+08 |
| TiDE | 9.2E+07 | 7.7E+07 | 2.5E+07 | 1.1E+08 | 1.2E+08 | 6.0E+09 | 3.9E+09 | 9.2E+07 | 1.31E+09 |
| iTransformer | 2.9E+06 | 2.9E+06 | 1.0E+07 | 2.9E+06 | 1.3E+08 | 1.9E+09 | 8.7E+09 | 2.9E+06 | 1.34E+09 |
| SCINet | 3.4E+06 | 3.4E+06 | 7.3E+05 | 7.3E+05 | 4.8E+06 | 4.7E+09 | 6.6E+09 | 3.4E+06 | 1.41E+09 |
| FEDformer | 1.8E+09 | 1.8E+09 | 1.8E+09 | 1.8E+09 | 1.8E+09 | 2.2E+09 | 3.1E+09 | 1.8E+09 | 2.01E+09 |
| Crossformer | 1.2E+09 | N/A | 9.7E+08 | N/A | 2.5E+09 | 2.6E+09 | 5.4E+09 | 1.2E+09 | 2.31E+09 |
| **XCTFormer** (Ours) | 3.3E+06 | 4.4E+07 | 4.0E+05 | 4.1E+06 | 4.9E+08 | 6.1E+09 | 2.0E+10 | 3.3E+06 | 3.33E+09 |
| TimesNet | 3.8E+09 | 1.9E+09 | 1.3E+09 | 2.5E+09 | 2.6E+09 | 3.2E+11 | 6.5E+11 | 3.8E+09 | 1.23E+11 |

[†] Synthetic uses the multivariate-to-single (MS) setting with ETTm1 hyperparameters.
 Crossformer values on ETTh2 and ETTm2 are N/A due to configuration incompatibility. TimeMixer++ is excluded as its code is not publicly available.

Table 28: Trainable parameter counts per dataset, averaged over prediction lengths $\{96, 192, 336, 720\}$. For MTLinear, values are the average of MTDLinear and MTNLinear. See Section B.6 for measurement methodology.

| Model | ETTm1 | ETTm2 | ETTh1 | ETTh2 | Weather | ECL | Traffic | Synth.[†] | Avg |
|---|---|---|---|---|---|---|---|---|---|
| DLinear | 6.5E+04 | 6.5E+04 | 6.5E+04 | 6.5E+04 | 6.5E+04 | 6.5E+04 | 6.5E+04 | 6.5E+04 | 6.50E+04 |
| TimeMixer | 1.2E+05 | 1.2E+05 | 1.2E+05 | 1.2E+05 | 1.5E+05 | 1.5E+05 | 1.7E+05 | 1.2E+05 | 1.34E+05 |
| Autoformer | 1.2E+05 | 1.2E+05 | 1.2E+05 | 1.2E+05 | 1.3E+05 | 3.2E+05 | 6.7E+05 | 1.2E+05 | 2.15E+05 |
| PatchTST | 9.2E+05 | 9.2E+05 | 8.2E+04 | 8.2E+04 | 9.2E+05 | 9.2E+05 | 9.2E+05 | 9.2E+05 | 7.10E+05 |
| MTLinear | 2.0E+05 | 2.4E+05 | 2.0E+05 | 2.4E+05 | 5.9E+05 | 2.2E+06 | 2.3E+06 | 2.0E+05 | 7.71E+05 |
| iTransformer | 2.6E+05 | 2.6E+05 | 9.0E+05 | 2.6E+05 | 5.0E+06 | 5.0E+06 | 6.5E+06 | 2.6E+05 | 2.30E+06 |
| Leddam | 2.9E+06 | 3.8E+06 | 3.8E+06 | 1.3E+06 | 1.3E+06 | 3.4E+06 | 3.4E+06 | 2.9E+06 | 2.85E+06 |
| **XCTFormer** (Ours) | 2.0E+05 | 1.3E+06 | 4.1E+04 | 1.7E+05 | 2.9E+06 | 3.8E+06 | 3.2E+07 | 2.0E+05 | 5.08E+06 |
| Crossformer | 1.1E+07 | N/A | 1.1E+07 | N/A | 1.1E+07 | 1.3E+06 | 1.9E+06 | 1.1E+07 | 7.87E+06 |
| TiDE | 9.6E+06 | 8.7E+06 | 2.4E+06 | 1.4E+07 | 4.3E+06 | 1.6E+07 | 3.4E+06 | 9.6E+06 | 8.53E+06 |
| FEDformer | 1.6E+07 | 1.6E+07 | 1.6E+07 | 1.6E+07 | 1.6E+07 | 1.8E+07 | 2.1E+07 | 1.6E+07 | 1.69E+07 |
| TimesNet | 2.7E+06 | 1.1E+06 | 6.3E+05 | 1.2E+06 | 1.2E+06 | 1.5E+08 | 3.0E+08 | 2.7E+06 | 5.74E+07 |
| SCINet | 2.8E+05 | 2.8E+05 | 7.2E+04 | 7.2E+04 | 3.1E+05 | 1.2E+08 | 4.5E+08 | 2.8E+05 | 7.14E+07 |

[†] Synthetic uses the multivariate-to-single (MS) setting with ETTm1 hyperparameters.
 Crossformer values on ETTh2 and ETTm2 are N/A due to configuration incompatibility. TimeMixer++ is excluded as its code is not publicly available.

