# OpenReview forum: "XCTFormer: Leveraging Cross-Channel and Cross-Time Dependencies for Enhanced Time-Series Analysis"
_TMLR — Accepted by TMLR_

### Review · Reviewer_H7Wf · 2025-11-04

**Summary Of Contributions:**

The authors aim to address the paradox in time-series analysis where CI models often outperform CD models, even though CD models often have more information to leverage. To achieve this, the authors propose XCTFormer, which is able to explicitly leverage cross-channel dependencies. This leverages three key architectural pieces including CRAB (Cross-Relational Attention Block) which significantly enhances performance, a data processing module, and an optional Dependency Compression Plugin (DeCoP). The results are generally better than almost all existing models for long-term forecasting, imputation, and anomaly detection.

**Audience:**

Yes

**Audience Explanation:**

The proposed XCTFormer generally achieves very strong results for time-series analysis and thus would be of great interest for the time-series analsysis community.

**Claims And Evidence:**

Yes

**Claims Explanation:**

- The results are overall very strong, and the motivation is also very strong.
- The ablations confirm the design choice regarding XCTFormer's changes to a standard baseline, with every changes helping performance
- The authors go above and beyond with ensuring the experiments hold across seeds and datasets, making a strong arugment for the proposed XCTFormer.

**Requested Changes:**

- It's unclear to me how much of the performance boost comes from the XCTFormer just being potentially more expensive.
    - Is it possible to compare the XCTFormer to the baselines, not just in terms of performance, but also in terms of parameters and FLOPs? Without this information, it is difficult to ensure a fair comparison is done.
    - This change is relatively critical for my recommendation, as without this information it's not possible to know how fair the comparisons made were.
- Is it possible to do an ablation regarding how much of the performance benefit was due to the data processing module?

---

> ### Author Response · Authors · 2026-01-13
> **Response to Reviewer H7Wf [Part1]**
>
> We would like to sincerely thank the reviewer for the valuable comments.
>
> Requested Changes:
>
> 1. Question:
>
> > It's unclear to me how much of the performance boost comes from the XCTFormer just being potentially more expensive.
> >1. Is it possible to compare the XCTFormer to the baselines, not just in terms of performance, but also in terms of parameters and FLOPs? Without this information, it is difficult to ensure a fair comparison is done.
> >2. This change is relatively critical for my recommendation, as without this information it's not possible to know how fair the comparisons made were.
>
> Answer:
>
> To ensure a fair comparison, we added summaries of parameter counts and FLOPs, computed under the *same experimental setting used throughout the paper*. For each baseline, we used the official implementations and hyperparameters from the authors’ original GitHub repositories when available. The tables below report the parameters/FLOPs per model and dataset, averaged across the different forecasting horizons.
>
> **Parameter analysis.** As shown in the table, XCTFormer is positioned fairly relative to other models, especially Transformer-based ones. In particular, it uses fewer parameters on average than iTransformer, LeDDAM, and CrossFormer.
>
>
> **Parameters Summary (sorted by Overall Avg, ascending)**
>
> | Model | ETTh1 | ETTh2 | ETTm1 | ETTm2 | Electricity | traffic | weather | Overall Avg |
> |---|---:|---:|---:|---:|---:|---:|---:|---:|
> | DLinear | 6.52E+04 | 6.52E+04 | 6.52E+04 | 6.52E+04 | 6.52E+04 | 6.52E+04 | 6.52E+04 | 6.52E+04 |
> | TimeMixer | 1.20E+05 | 1.20E+05 | 1.20E+05 | 1.22E+05 | 1.48E+05 | 1.58E+05 | 1.48E+05 | 1.34E+05 |
> | Autoformer | 1.21E+05 | 1.21E+05 | 1.21E+05 | 1.21E+05 | 3.22E+05 | 6.69E+05 | 1.30E+05 | 2.29E+05 |
> | PatchTST | 8.13E+04 | 8.13E+04 | 9.16E+04 | 9.16E+04 | 9.16E+05 | 9.16E+05 | 9.16E+05 | 6.78E+05 |
> | MTLinear | 1.96E+05 | 2.44E+05 | 1.96E+05 | 2.44E+05 | 2.15E+06 | 2.30E+06 | 5.87E+05 | 8.45E+05 |
> | **XCTFormer** | **3.33E+04** | **1.44E+05** | **1.47E+05** | **1.31E+06** | **3.04E+06** | **8.26E+06** | **2.49E+05** | **2.20E+06** |
> | iTransformer | 2.17E+06 | 2.55E+05 | 2.55E+05 | 2.55E+05 | 4.96E+06 | 6.53E+06 | 4.96E+06 | 2.77E+06 |
> | Leddam | 3.79E+06 | 1.25E+05 | 2.89E+06 | 3.78E+06 | 3.36E+06 | 3.37E+06 | 1.25E+05 | 2.81E+06 |
> | Crossformer | 1.11E+07 | N/A | 1.11E+07 | N/A | 7.09E+05 | 7.10E+05 | 1.11E+07 | 6.94E+06 |
> | FEDformer | 1.05E+07 | 1.05E+07 | 1.05E+07 | 1.05E+07 | 1.21E+07 | 1.49E+07 | 1.06E+07 | 1.14E+07 |
> | TimesNet | 6.29E+05 | 1.22E+06 | 2.68E+06 | 1.07E+06 | 1.50E+08 | 3.02E+08 | 1.22E+06 | 6.56E+07 |
>
>
> **FLOPs analysis.** XCTFormer incurs a relatively larger FLOPs budget because it explicitly models all pairwise cross-channel and cross-time dependencies. Nevertheless, its computational cost remains comparable to that of other Transformer-based models. Additionally, both parameter usage and FLOPs scale with dataset dimensionality: when applied to smaller datasets, XCTFormer uses marginally fewer parameters and requires fewer FLOPs.
>
> **FLOPs Summary (sorted by Overall Avg, ascending)**
>
> | Model | ETTh1 | ETTh2 | ETTm1 | ETTm2 | Electricity | traffic | weather | Overall Avg |
> |---|---:|---:|---:|---:|---:|---:|---:|---:|
> | MTLinear | 3.39E+05 | 3.39E+05 | 3.39E+05 | 3.39E+05 | 1.55E+07 | 4.17E+07 | 1.02E+06 | 8.52E+06 |
> | DLinear | 4.52E+05 | 4.52E+05 | 4.52E+05 | 4.52E+05 | 2.07E+07 | 5.57E+07 | 1.36E+06 | 1.14E+07 |
> | Autoformer | 2.45E+07 | 2.45E+07 | 2.45E+07 | 2.45E+07 | 8.43E+07 | 1.87E+08 | 2.72E+07 | 5.67E+07 |
> | Leddam | 4.66E+07 | 1.15E+07 | 3.45E+07 | 4.65E+07 | 1.46E+09 | 3.91E+09 | 3.46E+07 | 7.92E+08 |
> | PatchTST | 1.80E+06 | 1.80E+06 | 3.71E+07 | 3.71E+07 | 1.70E+09 | 4.57E+09 | 1.11E+08 | 9.22E+08 |
> | TimeMixer | 1.58E+07 | 1.58E+07 | 1.58E+07 | 3.16E+07 | 9.24E+08 | 5.91E+09 | 6.04E+07 | 9.97E+08 |
> | iTransformer | 2.38E+07 | 2.80E+06 | 2.80E+06 | 2.80E+06 | 1.61E+09 | 5.66E+09 | 1.24E+08 | 1.06E+09 |
> | FEDformer | 2.08E+09 | 2.08E+09 | 2.08E+09 | 2.08E+09 | 2.56E+09 | 3.38E+09 | 2.10E+09 | 2.33E+09 |
> | Crossformer | 9.53E+08 | N/A | 1.19E+09 | N/A | 2.45E+09 | 5.04E+09 | 2.44E+09 | 2.41E+09 |
> | **XCTFormer** | **2.85E+05** | **2.76E+06** | **2.37E+06** | **4.05E+07** | **5.52E+09** | **1.58E+10** | **3.95E+08** | **3.11E+09** |
> | TimesNet | 1.27E+09 | 2.54E+09 | 3.79E+09 | 1.94E+09 | 3.24E+11 | 6.49E+11 | 2.55E+09 | 1.41E+11 |

---

> > ### Comment · Reviewer_H7Wf · 2026-01-24
> >
> > Thanks for the information. The results/information generally seem to answer my questions and confirm that XCTFormer is worth the FLOP/parameter budget. To make the results even clearer, is it possible to add the FLOP and parameter information to Tables 1 and 2 in the main paper? That would make it easier for readers to determine the FLOP/parameter tradeoffs more directly.
> >
> > Aside from this, I am content with the author's response.

---

> > > ### Author Response · Authors · 2026-01-26
> > > **Response to Reviewer H7Wf [Part1]**
> > >
> > > We care deeply about presenting our work as transparently and accurately as possible. After submitting our rebuttal, we noticed that the method we used to report parameter counts and FLOPs was not fully accurate.
> > >
> > > Specifically, we computed parameters/FLOPs using thop.profile. This tool accounts primarily for supported built-in modules (e.g., Linear/Conv layers) and may omit custom learnable tensors or operations that are not registered as standard modules. As a result, a few meaningful learnable components in our implementation (e.g., a learnable mask) were not included in the previously reported parameter totals, leading to an underestimation of our parameter count.
> > >
> > > Accordingly, we have recomputed the computational cost using a more reliable approach: we now count trainable parameters by summing all parameters with gradients enabled, and we compute FLOPs using fvcore under the same input shapes and experimental settings used throughout the paper. The updated table is provided below.
> > >
> > > **FLOPs Summary Table (sorted by Overall Avg)**
> > >
> > > | Model | ETTh1 | ETTh2 | ETTm1 | ETTm2 | Electricity | Traffic | Weather | Overall Avg |
> > > |---|---:|---:|---:|---:|---:|---:|---:|---:|
> > > | MTNLinear | 2.26E+05 | 2.26E+05 | 2.26E+05 | 2.26E+05 | 1.04E+07 | 2.78E+07 | 6.77E+05 | 5.68E+06 |
> > > | MTLinear | 3.39E+05 | 3.39E+05 | 3.39E+05 | 3.39E+05 | 1.55E+07 | 4.17E+07 | 1.02E+06 | 8.52E+06 |
> > > | MTDLinear | 4.52E+05 | 4.52E+05 | 4.52E+05 | 4.52E+05 | 2.07E+07 | 5.56E+07 | 1.35E+06 | 1.14E+07 |
> > > | DLinear | 4.52E+05 | 4.52E+05 | 4.52E+05 | 4.52E+05 | 2.07E+07 | 5.56E+07 | 1.35E+06 | 1.14E+07 |
> > > | Autoformer | 2.44E+07 | 2.44E+07 | 2.44E+07 | 2.44E+07 | 8.42E+07 | 1.87E+08 | 2.71E+07 | 5.66E+07 |
> > > | PatchTST | 1.90E+06 | 1.90E+06 | 3.79E+07 | 3.79E+07 | 1.74E+09 | 4.67E+09 | 1.14E+08 | 9.43E+08 |
> > > | Leddam | 4.67E+07 | 1.16E+07 | 3.46E+07 | 4.66E+07 | 1.62E+09 | 5.06E+09 | 3.49E+07 | 9.79E+08 |
> > > | TimeMixer | 1.57E+07 | 1.57E+07 | 1.57E+07 | 3.15E+07 | 9.21E+08 | 5.90E+09 | 6.02E+07 | 9.94E+08 |
> > > | iTransformer | 1.01E+07 | 2.87E+06 | 2.87E+06 | 2.87E+06 | 1.94E+09 | 8.73E+09 | 1.26E+08 | 1.54E+09 |
> > > | SCINet | 7.25E+05 | 7.25E+05 | 3.37E+06 | 3.37E+06 | 4.69E+09 | 6.63E+09 | 4.79E+06 | 1.62E+09 |
> > > | FEDformer | 1.76E+09 | 1.76E+09 | 1.76E+09 | 1.76E+09 | 2.24E+09 | 3.06E+09 | 1.78E+09 | 2.02E+09 |
> > > | Crossformer | 9.66E+08 | N/A | 1.22E+09 | N/A | 2.63E+09 | 5.37E+09 | 2.48E+09 | 2.53E+09 |
> > > | **XCTFormer** | **3.99E+05** | **4.05E+06** | **3.28E+06** | **4.37E+07** | **6.07E+09** | **2.03E+10** | **4.90E+08** | **3.84E+09** |
> > > | TimesNet | 1.28E+09 | 2.54E+09 | 3.76E+09 | 1.94E+09 | 3.24E+11 | 6.49E+11 | 2.55E+09 | 1.41E+11 |
> > >
> > >
> > > **Parameters Summary Table (sorted by Overall Avg)**
> > >
> > > | Model | ETTh1 | ETTh2 | ETTm1 | ETTm2 | Electricity | Traffic | Weather | Overall Avg |
> > > |---|---:|---:|---:|---:|---:|---:|---:|---:|
> > > | DLinear | 6.52E+04 | 6.52E+04 | 6.52E+04 | 6.52E+04 | 6.52E+04 | 6.52E+04 | 6.52E+04 | 6.52E+04 |
> > > | TimeMixer | 1.20E+05 | 1.20E+05 | 1.20E+05 | 1.22E+05 | 1.51E+05 | 1.65E+05 | 1.49E+05 | 1.35E+05 |
> > > | Autoformer | 1.21E+05 | 1.21E+05 | 1.21E+05 | 1.21E+05 | 3.22E+05 | 6.69E+05 | 1.30E+05 | 2.29E+05 |
> > > | MTNLinear | 1.30E+05 | 1.63E+05 | 1.30E+05 | 1.63E+05 | 1.43E+06 | 1.53E+06 | 3.91E+05 | 5.63E+05 |
> > > | PatchTST | 8.15E+04 | 8.15E+04 | 9.18E+05 | 9.18E+05 | 9.18E+05 | 9.18E+05 | 9.18E+05 | 6.79E+05 |
> > > | MTLinear | 1.96E+05 | 2.44E+05 | 1.96E+05 | 2.44E+05 | 2.15E+06 | 2.30E+06 | 5.87E+05 | 8.45E+05 |
> > > | MTDLinear | 2.61E+05 | 3.26E+05 | 2.61E+05 | 3.26E+05 | 2.87E+06 | 3.06E+06 | 7.82E+05 | 1.13E+06 |
> > > | iTransformer | 9.03E+05 | 2.55E+05 | 2.55E+05 | 2.55E+05 | 4.96E+06 | 6.53E+06 | 4.96E+06 | 2.59E+06 |
> > > | Leddam | 3.80E+06 | 1.25E+06 | 2.90E+06 | 3.78E+06 | 3.36E+06 | 3.37E+06 | 1.26E+06 | 2.82E+06 |
> > > | Crossformer | 1.13E+07 | N/A | 1.14E+07 | N/A | 1.32E+06 | 1.86E+06 | 1.14E+07 | 7.47E+06 |
> > > | **XCTFormer** | **4.05E+04** | **1.66E+05** | **2.04E+05** | **1.32E+06** | **4.53E+06** | **5.59E+07** | **2.87E+06** | **9.29E+06** |
> > > | FEDformer | 1.63E+07 | 1.63E+07 | 1.63E+07 | 1.63E+07 | 1.79E+07 | 2.07E+07 | 1.64E+07 | 1.72E+07 |
> > > | TimesNet | 6.29E+05 | 1.22E+06 | 2.68E+06 | 1.07E+06 | 1.50E+08 | 3.02E+08 | 1.22E+06 | 6.56E+07 |
> > > | SCINet | 7.17E+04 | 7.17E+04 | 2.84E+05 | 2.84E+05 | 1.24E+08 | 4.46E+08 | 3.09E+05 | 8.16E+07 |
> > >
> > >
> > > These corrections do not materially alter the overall picture: the FLOPs remain largely unchanged, while the parameter count is higher than what we previously reported. The parameter count still outperforms a few baselines, but it is less competitive than in our earlier table. Compared to iTransformer, our parameter count is generally similar. The main exception is Traffic, where we use a larger setup, which increases the overall average. The observed pattern, in which both parameters and FLOPs increase with dataset dimensionality, remains.

---

> > > ### Author Response · Authors · 2026-01-26
> > > **Response to Reviewer H7Wf [Part2]**
> > >
> > > Thank you for the suggestion. We will add the FLOPs/parameter results alongside Table 1 to enable a clearer comparison in the revised version.
> > >
> > > Regarding Table 2 (imputation), we unfortunately cannot provide a fully comparable FLOPs/parameter table. The imputation results are taken directly from the TimeMixer++ paper, but the exact hyperparameter configurations used to obtain the reported numbers for TimeMixer++ and several baselines are not available. Moreover, the TimeMixer++ code is not publicly available to the best of our knowledge. Without these details, any FLOP/parameter estimates would be incomplete and potentially misleading.

---

> ### Author Response · Authors · 2026-01-13
> **Response to Reviewer H7Wf [Part2]**
>
> 2. Question:
>
> > Is it possible to do an ablation regarding how much of the performance benefit was due to the data processing module?
>
> Answer:
>  Our data processing pipeline builds on the processing unit introduced in PatchTST, with an additional component required for XCTFormer. The base pipeline includes patching, linear projection, and a learnable positional encoding. Because these elements are part of the standard PatchTST tokenization procedure, they are not expected to account for XCTFormer’s unique performance gains. The main method-specific addition is the flattening operation, which flattens the sequence and variate dimensions to enable attention over all pairwise time and channel interactions.
>
> An ablation of this flattening operation is already provided in our paper. Using the same backbone and training setup, we compare three modeling strategies: (1) full flattened attention over all pairwise connections (XCTFormer), (2) attention over the sequence dimension only (PatchTST-style), and (3) attention over the variate dimension only (iTransformer-style). This directly isolates the contribution of flattening and shows that modeling all pairwise dependencies is one of the drivers of the observed improvements.
>
> To further isolate the remaining components of the processing unit, we performed a new ablation that removes the positional encoding (identical to the learnable positional encoding used in PatchTST). The original XCTFormer variant performs about 1% better on average, suggesting that positional encoding provides a slight benefit, while the main gains are not primarily driven by this component.
>
> **Ablation study results across different tasks, evaluated with different XCTFormer variations.**
>
> | Model | Forecasting MSE | Forecasting MAE | Imputation MSE | Imputation MAE | Anomaly Precision | Anomaly Recall | Anomaly F-Score | XCTFormer vs Others (%) |
> |---|---:|---:|---:|---:|---:|---:|---:|---:|
> | **XCTFormer (Original)** | **0.328** | **0.337** | **0.044** | **0.124** | **92.1** | **83.7** | **87.6** | - |
> | _W/o Positional Encoding_ | _0.331_ | _0.339_ | _0.044_ | _0.125_ | _92.1_ | _81.3_ | _86.2_ | 1.0% |

---

### Review · Reviewer_92Lb · 2025-11-17

**Summary Of Contributions:**

This paper proposes new architectural components for transformer-based time series forecasting models. The goal is to enable modeling cross-channel and temporal dependencies simultaneously without impacting model performance. The key ideas in this paper are: a) learn a real-valued mask --- encoding the temporal/cross-channel dependencies of a token --- that can ultimately be used to scale the attention-logits; b) use signed attentions by replacing softmax with a different variety of row-normalization. The architecture is evaluated on 3 kinds of time-series tasks: (1) long-term forecasting, (2) imputation and (3) anomaly detection.

**Audience:**

Yes

**Audience Explanation:**

The ideas in this paper are useful for machine learning practitioners working in the area of time-series-forecasting/classification.

**Claims And Evidence:**

No

**Claims Explanation:**

Partially, no. The key claim in this paper is that the paper establishes state-of-the-art results in time-series related tasks --- namely, forecasting, imputation and anomaly detection. I disagree with the claim related to SOTA on forecasting because direct comparison of forecasting results in this paper with PatchTST results reveal that PatchTST has better or same results for almost all dataset/forecast pairs. I am open to being convinced otherwise and seek clarification from the authors on how to interpret this difference and how this method establishes a new state-of-art.

**Requested Changes:**

1. I request the authors to clarify how this paper is positioned with respect to other results reported in the literature and update the paper if needed.
2. DeCoP: While DeCoP is designed to address runtime complexity, it appears to me that it can also address overfitting due to spurious correlations. I probably missed this, but is there any analysis on how varying the k affects the results?
3. Hyperparameter-search space: Could you please clarify the hyperparameter-search space for this study? For example, what are the ranges/values that optuna searched over for Table 9? And, it appears that the hyperparameters are fixed for all forecasting horizons of a specific dataset?
4. Out of curiousity, how does this method perform on the Exchange dataset? PatchTST explicitly stated that this is a challenging problem for them but wondering if this method is robust?

---

> ### Author Response · Authors · 2026-01-13
> **Response to Reviewer 92Lb [Part1]**
>
> We would like to sincerely thank the reviewer for the careful reading and for raising these important points.
>
> 1. Question:
> > I request the authors to clarify how this paper is positioned with respect to other results reported in the literature and update the paper if needed.
>
> Answer:
>
> We would like to clarify that our paper does **not** claim to establish state-of-the-art (SoTA) performance in the *forecasting* task. Rather, our claim is that the proposed method improves or is competitive with strong, widely adopted baselines while simultaneously addressing multiple data domains. We carefully revised the manuscript to ensure that all statements accurately reflect our contributions and toned down any overstated claims.
>
> Regarding the comparison with PatchTST, we acknowledge that the forecasting results reported in the original PatchTST paper appear superior to ours. However, it is important to note that those results were obtained under a different experimental setting. In particular, the original PatchTST evaluations used a look-back window of 336. In contrast, our experiments use a look-back length of 96, which has become the modern, widely accepted setup in recent time-series forecasting papers (e.g., iTransformer, LeDDAM, TimeMixer, TimeMixer++).
>
> The PatchTST results reported in our tables are taken directly from the TimeMixer++ paper, which **either reports the original numbers from the corresponding papers or reproduces them when needed**, under aligned experimental settings. This procedure is designed to ensure fair and consistent comparison across methods. Under this commonly used evaluation setup, the observed performance differences therefore highlight the benefits of our approach.
>
> 2. Question:
> > DeCoP: While DeCoP is designed to address runtime complexity, it appears to me that it can also address overfitting due to spurious correlations. I probably missed this, but is there any analysis on how varying the k affects the results?
>
> Answer:
>
> We thank the reviewer for this thoughtful point. DeCoP is introduced primarily to reduce computation time and memory usage.  We had not considered that this compression could act as an implicit regularizer by limiting the capacity of the pairwise dependency representation, and we see it as a promising direction for future work.
>
>
> We conducted a sensitivity analysis on k and added it to the revised paper in Appendix C.2. The analysis shows that DeCoP’s compression rank k is dataset-dependent. In this experiment, we tested k from 2 to 256 and reported average MAE and MSE across forecasting horizons 96, 192, 336, and 720, and across imputation mask ratios 0.125, 0.25, 0.375, and 0.5. On Traffic, performance generally improves as k increases, suggesting that higher capacity compressed interactions help capture complex multivariate dependencies. In contrast, for Electricity, both forecasting and imputation, small to mid k values are often competitive or slightly better, indicating diminishing returns from larger k and a possible regularization benefit from stronger compression. Overall, k should be treated as a dataset-specific hyperparameter and tuned to balance accuracy against compute and memory cost.

---

> ### Author Response · Authors · 2026-01-13
> **Response to Reviewer 92Lb [Part2]**
>
> 3. Question:
> > Hyperparameter-search space: Could you please clarify the hyperparameter-search space for this study? For example, what are the ranges/values that optuna searched over for Table 9? And, it appears that the hyperparameters are fixed for all forecasting horizons of a specific dataset?
>
> Answer:
>
> We used Optuna over the following search ranges and included the full hyperparameter search space table in Appendix B.3 of the revised paper.
>
> **Hyperparameter Search Space**
>
> | Task | Datasets | lr | att_dropout | n_heads | e_layers | d_model | patch_len | dropout | fc_dropout | k |
> |---|---|---|---|---|---|---|---|---|---|---|
> | Long-term Forecast | ETTh1, ETTh2 | {5e-4, 1e-3, 5e-3, 1e-2} | [0.1, 0.8] (step 0.1) | {1, 2, 4} | [1, 3] | [4, 64] (step 2) | -- | [0.1, 0.3] (step 0.05) | [0.05, 0.3] (step 0.05) | -- |
> | Long-term Forecast | Others | {5e-4, 1e-3, 5e-3, 1e-2} | [0.1, 0.8] (step 0.1) | {1, 2, 4} | [1, 3] | [8, 256] (step 8) | -- | -- | -- | [64, 256] (step 64)† |
> | Imputation | All | {5e-4, 1e-3, 5e-3, 1e-2} | [0.1, 0.8] (step 0.1) | {1, 2, 4} | [1, 3] | [32, 256] (step 32) | {16, 64, 128} | -- | -- | [64, 256] (step 64)† |
> | Anomaly Detection | All | {5e-4, 1e-3, 5e-3, 1e-2} | [0.1, 0.8] (step 0.1) | {1, 2, 4} | [1, 3] | [8, 256] (step 8) | -- | -- | -- | -- |
>
> *Notation:* `[a, b]_s` denotes a range from `a` to `b` with step `s`; `{...}` denotes categorical choices; `--` indicates the parameter is not used.
> † `k` was searched only for the Electricity and Traffic datasets.
>
>
> Notes: We set the stride hyperparameter to patch_len / 2 and d_ff to 2 × d_model. We observed that our model overfits the ETTh1 and ETTh2 datasets during forecasting, so we reduced the search range for d_model and also tuned dropout rates.
>
> Related to the hyper-parameter score: Yes, the reported hyperparameters correspond to the selected configuration for each task and dataset. In other words, for a given task and dataset, we use a single hyperparameter setting that is shared across different forecasting horizons, or across different imputation mask ratios.
>
> 4. Question:
>
> > Out of curiousity, how does this method perform on the Exchange dataset? PatchTST explicitly stated that this is a challenging problem for them but wondering if this method is robust?
>
> Answer:
>
> Here is the performance of XCTFormer compared to baselines on the Exchange dataset:
>
> **Long-term Forecasting — Exchange (MSE / MAE)**
> (**Bold** = best, _italic_ = second-best)
>
> | Model | 96 MSE | 96 MAE | 192 MSE | 192 MAE | 336 MSE | 336 MAE | 720 MSE | 720 MAE | Avg MSE | Avg MAE |
> |---|---:|---:|---:|---:|---:|---:|---:|---:|---:|---:|
> | XCTFormer | _0.085_ | _0.203_ | 0.204 | 0.320 | 0.387 | 0.448 | 1.088 | 0.788 | 0.441 | 0.440 |
> | MTLinear | **0.084** | **0.201** | **0.173** | _0.300_ | _0.306_ | _0.412_ | **0.595** | **0.595** | **0.289** | **0.377** |
> | Leddam | -- | -- | -- | -- | -- | -- | -- | -- | -- | -- |
> | TimeMixer | 0.090 | 0.235 | 0.187 | 0.343 | 0.353 | 0.473 | 0.934 | 0.761 | 0.391 | 0.453 |
> | iTransformer | 0.086 | 0.206 | 0.177 | **0.299** | 0.331 | 0.417 | 0.847 | _0.691_ | 0.360 | _0.403_ |
> | PatchTST | 0.088 | 0.205 | _0.176_ | **0.299** | **0.301** | **0.397** | 0.901 | 0.714 | 0.366 | 0.404 |
> | Crossformer | 0.256 | 0.367 | 0.470 | 0.509 | 1.268 | 0.883 | 1.767 | 1.068 | 0.940 | 0.707 |
> | TiDE | 0.094 | 0.218 | 0.184 | 0.307 | 0.349 | 0.431 | 0.852 | 0.698 | 0.370 | 0.413 |
> | TimesNet | 0.107 | 0.234 | 0.226 | 0.344 | 0.367 | 0.448 | 0.964 | 0.746 | 0.416 | 0.443 |
> | DLinear | 0.088 | 0.218 | _0.176_ | 0.315 | 0.313 | 0.427 | _0.839_ | 0.695 | _0.354_ | 0.414 |
> | SCINet | 0.267 | 0.396 | 0.351 | 0.459 | 1.324 | 0.853 | 1.058 | 0.797 | 0.750 | 0.626 |
> | FEDformer | 0.148 | 0.278 | 0.271 | 0.315 | 0.460 | 0.427 | 1.195 | 0.695 | 0.518 | 0.429 |
> | Autoformer | 0.197 | 0.323 | 0.300 | 0.369 | 0.509 | 0.524 | 1.447 | 0.941 | 0.613 | 0.539 |
>
> As noted by the PatchTST authors, Exchange behaves close to a random walk, with weak predictable structure and few recurring patterns. It consists of daily exchange rates for eight major countries relative to the US dollar. In such a setting, there is a limited amount of learnable signal, and forecast error naturally increases with the prediction horizon. As a result, XCTFormer is not expected to perform well in such settings. In our experiments, it remains competitive with several transformer-based baselines and shows clear gains over CrossFormer, a closely related method that models cross-channel and cross-time relationships through a stage-based design

---

> > ### Comment · Reviewer_92Lb · 2026-02-04
> > **Thank you!**
> >
> > Dear authors,
> >
> > Thank you very much for your explanations/clarifications and additional experimental results.
> >
> > My thinking on PatchTST results is as follows:
> > * The lookback horizon (96 vs 336) should be considered an hyperparameter and is not really a different experimental setting. Practically speaking, we have much more than just past 336 samples at inference time --- for example, we have the entire training-period/val-period data. Ideally, we would like a forecasting model to dynamically adapt to all available historical observations but this isn't usually scalable/generalizable in practice.
> > * Secondly, given the vast number of hyperparameter choices for XCTFormer, it is only fair that PatchTST gets to at least select the lookback horizon!
> > * Finally, it appears that TimeMixer++ also has comparatively better results in almost all forecasting datasets. Is my interpretation correct? If so, please add TimeMixer++ to Table 1.
> >
> > My conclusion is that XCTFormer seems to offer good performance on other time-series tasks (e.g., anomaly-detection, imputation) but comes close to SOTA on forecasting.
> >
> > Optionally, the authors may consider developing a synthetic dataset --- where forecastability crucially depends on pairwise relationships --- to highlight the advantages of this method compared to rest. Please let me know your thoughts.
> >
> > Sincerely,
> > Reviewer 92Lb.
> >
> > PS: I deeply apologize for the delayed response: I was out of town for past 2 weeks.

---

> > > ### Comment · Action_Editor_CJdP · 2026-02-04
> > > **Thanks, authors you have time to respond**
> > >
> > > Thanks for this. Let’s continue the conversation. Authors, I will postpone making a final decision on this paper to give you sometime to address these assertions. Please do so as expediently as possible.

---

> > > ### Author Response · Authors · 2026-02-06
> > > **Response to Reviewer 92Lb [Part1]**
> > >
> > > We thank the reviewer for the thoughtful comments and respond to each point below.
> > >
> > > 1-2. Regarding the lookback parameter, we would like to emphasize that it should not be viewed as a hyperparameter. Instead, it should be regarded as part of the experimental setting that defines the forecasting task. Increasing the lookback window provides additional historical information and can materially change the problem’s difficulty. Therefore, comparing models evaluated with different lookback lengths does not constitute a fair or controlled comparison. In addition, in the forecasting literature, the lookback length is treated as part of the experimental setting and is kept fixed across methods within each benchmark table (including all baseline models we compare against). We follow this standard setup and evaluate only a single widely adopted setting. In particular, we did not test XCTFormer with lookback 336. All our main results use lookback 96, which is the most widely adopted default setting in recent literature.
> > >
> > > 3. Forecasting is a rapidly evolving research area, with new architectures and modeling paradigms introduced at nearly every major conference. Accordingly, we do not claim state-of-the-art forecasting accuracy and acknowledge that newer and stronger models may achieve better results. Our objective is to position XCTFormer against strong, widely accepted baselines in order to evaluate its effectiveness in addressing the proposed gap in cross-dimensional modeling. Notably, the development of XCTFormer was concurrent with the TimeMixer++ work, and improving XCTFormer’s performance to match or surpass the latest SoTA methods (e.g., TimeMixer++) would require additional research beyond the scope of this submission. Given that our contribution is not centered on establishing SoTA forecasting performance, we do not see additional scientific value in adding TimeMixer++ results to Table 1.

---

> ### Author Response · Authors · 2026-02-06
> **Response to Reviewer 92Lb [Part2]**
>
> **Experiment with a synthetic dataset**
>
> We thank the reviewer for this insightful suggestion. We carefully considered it and conducted the proposed experiment. Specifically, we constructed a synthetic dataset to emphasize settings in which forecastability depends jointly on cross-variate and lagged cross-time relationships.
>
> The dataset consists of seven channels. To isolate the effect of cross-channel dependence, we compute training and evaluation loss only on the 7th channel (the synthetic target), rather than averaging across all channels. This target channel is constructed entirely from lagged signals in the first four channels, so it is hardly predictable from its own history. Channels 5 and 6 are independent random walks that act as distractors, meant to probe whether a model latches onto spurious correlations.
>
> **Synthetic data**
>
> | **Index** | **Type** | **Parameters** |
> | --- | --- | --- |
> | **1** | Sine | amp=1.0, freq=0.02, noise=0.02 |
> | **2** | Sine | amp=3.0, freq=0.03, phase=1.0, noise=0.07 |
> | **3** | Sine | amp=2.0, freq=0.01, phase=1.0, noise=0.03 |
> | **4** | Sine | amp=5.0, freq=0.002, phase=0.5, noise=0.02 |
> | **5** | Random walk | step_std=0.1 (**distractor**) |
> | **6** | Random walk | step_std=0.15 (**distractor**) |
>
> **Constructed Channel**
>
> The raw multivariate signals are split into overlapping patches of length 16 with a stride of 8.
>
> - **Cross-Variate Pair Formation**
>     - For each target patch at index **k**, two **lagged cross-variate pairs** are constructed:
>         - **Pair 1:** patch of **var₀** at **k−1** blended with patch of **var₁** at **k−2**
>         - **Pair 2:** patch of **var₂** at **k−2** blended with patch of **var₃** at **k−3**
> - **Cyclic Linear Blending Schedule**
>     - Each pair is blended using a weight **w(k)** following a **cyclic linear blending schedule** across the patch index.
>     - The schedule has a **period of 20 patches**:
>         - **Linearly increases** from 0 → 1 over the first 10 **patches**
>         - **Linearly decreases** from 1 → 0 over the next 10 **patches**
>     - This induces a **smooth, periodic alternation** in source dominance.
> - **Pairwise Blending**
>     - For a given pair \((i,j)\), the blended patch is computed as:
>         - **blended(k) = w(k) · patchᵢ + (1 − w(k)) · patchⱼ**
> - **Target Patch Construction**
>     - The final target patch is an **equal-weight combination** of the two blended pairs:
>         - **target(k) = 0.5 · blended_pair1(k) + 0.5 · blended_pair2(k)**
>
> **Reconstruction to Time Series**
>
> The produced patches were combined into a continuous time series, where values at time steps covered by multiple patches were averaged. This setup creates a **periodically changing dependency pattern** where the **dominant source channels switch over time**, while the **random-walk channels remain non-causal distractors**. Consequently, accurate forecasting requires leveraging lagged cross-variate dependencies and pairwise patch-level interactions, which penalize univariate and channel-independent models and highlight the importance of cross-channel reasoning.
>
> **Hyper-parameters**
>
> Selecting model-specific hyperparameters in this setting is nontrivial and would require extensive tuning for each baseline. To ensure a fair comparison, since the synthetic dataset contains seven channels (matching ETTm1), we used the same hyperparameter configurations as for ETTm1 across all baseline models without **hyperparameter search**. All methods are evaluated with a fixed lookback length of 96 and a forecasting horizon of 96.
>
> Results:
>
> | Model | MSE | MAE | Params | FLOPs |
> | --- | --- | --- | --- | --- |
> | **Crossformer** | **0.0193** | **0.1088** | 11,232,048 | 545M |
> | **XCTFormer** | **0.0413** | **0.1558** | 111,424 | 1.7M |
> | **Leddam** | **0.0446** | **0.1640** | 4,355,318 | 56M |
> | TimeMixer | 0.0689 | 0.2012 | 75,497 | 11M |
> | SCINet | 0.0954 | 0.2359 | 106,560 | 1.6M |
> | iTransformer | 0.1656 | 0.3086 | 224,224 | 2.5M |
> | PatchTST | 0.2423 | 0.3831 | 548,704 | 34M |
> | DLinear | 0.5100 | 0.5726 | 18,624 | 130K |
> | TimesNet | 0.5785 | 0.5966 | 4,708,167 | 4,509M |
> | MTLinear | 0.6704 | 0.6408 | 55,872 | 65K |
> | FEDformer | 0.7024 | 0.6724 | 16,303,111 | 1,189M |
> | Autoformer | 0.7332 | 0.6823 | 10,535,943 | 1,189M |
>
> The reported test errors are obtained by selecting the checkpoint with the best validation performance.
>
> Overall, CrossFormer and XCTFormer lead the table, consistent with the fact that both explicitly model cross-time and cross-channel dependencies. Notably, XCTFormer attains the second-best MSE while using ~99.0% fewer parameters (111K vs. 11.2M) and ~99.7% fewer FLOPs (1.7M vs. 545M) than CrossFormer. This ranking suggests that the synthetic data achieves its intended goal of rewarding models that exploit lagged cross-variate structure. Finally, performing baseline-specific hyperparameter search on this synthetic setting could further improve performance.

---

> > ### Comment · Reviewer_92Lb · 2026-02-06
> > **Thank you!**
> >
> > Dear Authors,
> >
> > Thanks so much for additional experiments and sharing your views on lookback-length as a hyperparameter.
> >
> > On PatchTST: I still maintain my position that the entire dataset is available for use as lookback-window and fixing the lookback-length is really an architectural choice. If a longer lookback-length can help dramatically improve performance, why would one not use it? In my own experience, longer lookback window does not make it better for all models: many models overfit to spurious signals and only some can generalize. That said, I will not push for inclusion of official PatchTST results.
> >
> > On TimeMixer++: In my view, not having the best results across all rows in Table 1 does not disqualify this paper and is better to include TimeMixer++ results in all tables rather than some.
> >
> > On Synthetic data: The results on synthetic data are both interesting and useful enough to be added to Table 1: thanks so much for trying this out, really appreciate the dataset construction methodology as well as the comprehensive analysis!
> >
> > I do not have any further objections.
> >
> > Thank You!
> > Sincerely,
> > Reviewer 92Lb.

---

> > > ### Author Response · Authors · 2026-02-11
> > > **Response to Reviewer 92Lb**
> > >
> > > Thank you again for your constructive and thoughtful feedback.
> > >
> > > We will revise the manuscript to (i) include TimeMixer++ results consistently across all relevant benchmark tables, and (ii) incorporate the synthetic dataset results into Table 1:
> > >
> > > Regarding TimeMixer++, since its code is not publicly available, we are unable to run it on our synthetic dataset. Therefore, its results will be included in the real-world benchmarks but not in the synthetic experiment.
> > >
> > > Concerning the synthetic setting, full multivariate input is available to all models. However, the task is single-target forecasting (predicting only the constructed channel), specifically designed to verify pairwise cross-variate dependencies and robustness to spurious correlations. This differs from the real-world benchmark tables, which evaluate full multivariate forecasting. We will add the synthetic results while explicitly stating this distinction to ensure clarity and a fair interpretation.
> > >
> > > We sincerely appreciate your helpful suggestions throughout the review process.
> > >
> > > Sincerely,
> > > The Authors

---

### Review · Reviewer_F3Af · 2025-12-29

**Summary Of Contributions:**

### Summary
The authors propose a new model based on transformers that takes into account multivariate dependencies with token to token attention. To do this, the authors propose three modules: learnable masks, signed normalisation and a compression plugin for efficiency gains. The authors evaluate their model on a range of tasks and results suggest good performance, particularly on imputation.

### Strengths

1. It addresses a real gap in the literature. Alternatives to two stage approaches are useful and can potentially capture and retain more information through a principled solution
2. Evaluation tests three different task types on well established benchmarks. It also compares against a wide range of baselines.
3. Ablation study isolates the contribution of each module in the system.
4. Interpretability analysis is very interesting, and maybe should be part of the main text.

### Weaknesses

1. The claim that CrossFormer and CARD can/may lead to information loss at the stage interfaces is not backed up. No citations for prior work that show this or any empirical demonstration. The authors don’t claim this as fact, but this is the basis for the motivation. Without this ablation/comparison study we do not know if the improved performance comes from the additional modules that are introduced.
2. I am not convinced that the signed attention weights claim to “increase expressiveness” because softmax performs worse. It would be interesting to see the distribution of the weights (or how many are negative). Clipping the weights, rather than passing them through the softmax function, might give a better signal for the importance of negative weights, as the softmax brings its own benefits on top of clipping negatives.
3. Does the learnable mask not grow quadratically with patches * channels? Even with the compression, the mask would be quadratic, no?
4. Performance improvements on anomaly detection is marginal. SOTA performance isn’t a requirement, but the claims might need to be better reflect the actual performance.
5. High variance across many tasks reduces confidence. Particularly as no statistical significance tests of difference are done to compare the different methods. Seed analysis is useful but it serves a different purpose.
6. No sensitivity analysis for patch hparams. This seems like a very important hyper parameter but there is no indication of its effect on the performance or how this should be chosen for a given task. In the experiments some use length of 16 with stride 8 while others use 64 / 32. It’s not clear how or why these were chosen and what effect it has.
7. The paper is completely lacking a discussion or limitations section. The conclusion is very positive and does not mention any of the caveats of the approach. I would expect an honest limitations section that overview the key limitations that the authors have identified, I have mentioned some of them above. A discussion on some of the results (e.g. high variance) would be an improvement as well.
8. Not really a weakness but a mention or discussion of TS foundation models would be apt. Although they are a different paradigm, they are increasingly relevant in time series modelling.

**Audience:**

Yes

**Audience Explanation:**

It is suitable for TMLR and many readers will find it interesting.

**Broader Impact Concerns:**

None identified.

**Claims And Evidence:**

No

**Claims Explanation:**

Some claims lack support.

1. The claim to model “all possible cross-channel and cross-temporal relationships” is overstated as the attention mechanism is inherently pairwise. There is no modelling of higher order (>2) interactions. This should rather say “all pairwise …"
2. I feel the forecasting performance is overstated as well. It is hard to be sure of its superiority over other method since the margins are relatively small and no statistical significance test to actually test this.
3. Although not a direct claim, much of the motivation hinges on two stage methods losing information. The limitations of this assumption and lack of evidence should be more clearly stated, as a simple read of the paper does leave the impression this is a matter of fact.
4. The importance of negative weights (see above) is not fully explained with the softmax experiment. It lacks the proper experimental setup to demonstrate this effect because of the confounding.

**Requested Changes:**

Critical:
1. Validate information loss in two-stage methods (or adjust claims)
2. Perform statistical significance tests to check for difference (or adjust claims)
3. Check negative weights claims with more appropriate experimental set up
4. Add a discussion of limitation

Less critical:
1. Add sensitivity analysis of patch and stride hyperparams
2. Scalability analysis
3. Adjust the “all possible relationships” claim
4. Compare the computational cost and scalability of the methods

---

> ### Comment · Action_Editor_CJdP · 2025-12-30
> **Thank you!**
>
> Thank you for completing this review and doing so in such a thorough manner! I'm hoping that we'll have a great phase of discussion with the authors soon!

---

> ### Author Response · Authors · 2026-01-13
> **Response to Reviewer F3Af [Part1]**
>
> We would like to sincerely thank the reviewer for providing careful and insightful review of our paper. We particularly appreciate the positive feedback on the interpretability analysis of the learnable mask.
>
> Weaknesses:
>
> 1. Question:
>
> >The claim that CrossFormer and CARD can/may lead to information loss at the stage interfaces is not backed up. No citations for prior work that show this or any empirical demonstration. The authors don’t claim this as fact, but this is the basis for the motivation. Without this ablation/comparison study we do not know if the improved performance comes from the additional modules that are introduced.
>
> Answer:
>
> We agree with the reviewer that our original wording was overly suggestive in the absence of direct evidence.  We have toned down the corresponding claims in the revised paper.
>
> 2. Question:
>
> > I am not convinced that the signed attention weights claim to “increase expressiveness” because softmax performs worse. It would be interesting to see the distribution of the weights (or how many are negative). Clipping the weights, rather than passing them through the softmax function, might give a better signal for the importance of negative weights, as the softmax brings its own benefits on top of clipping negatives.
>
> Answer:
>
> Thank you for your thoughtful comment. Our motivation for allowing negative values follows the work of “More Expressive Attention with Negative Weights”^1, which argues that allowing negative attention can increase expressability and performance relative to standard softmax activation. To test this in our setting, we replaced the original softmax activation with our signed activation while keeping the training setup and hyperparameters fixed. As reported in the ablation study in the main paper, the signed activation improves performance by 9.7% over the softmax baseline.
>
> We conducted several additional analyses and included them in Appendix C.4. First, we examined the weight distribution produced by our activation layer. The plots show that the signed weights are approximately Gaussian and centered near zero, so roughly half are negative. We also provide an explanation for why this distribution emerges. Additionally, we performed an ablation to validate the effectiveness and contribution of negative weights. Specifically, we clip negative weights by applying a ReLU at the end of our proposed activation. We evaluate this variant across all three time-series tasks under the full set of experimental settings, including four forecasting horizons and four imputation mask ratios. Overall, XCTFormer consistently matches or outperforms the clipped variant, with a modest average improvement of about 1.3%.
>
> >^1 Lv et al. (2024). *More Expressive Attention with Negative Weights*. arXiv:2411.07176. https://arxiv.org/abs/2411.07176
>
> 3. Question:
>
> > Does the learnable mask not grow quadratically with patches * channels? Even with the compression, the mask would be quadratic, no?
>
> Answer:
>
> **Mask complexity.** Without DeCoP, the mask would indeed be quadratic. Vanilla self-attention forms an attention matrix of size N×N, where N = S·V (sequence tokens × channels). Any mask applied to these logits is therefore also N×N, so it scales as O(N²) = O((S·V)²).
>
> In our method, the mask is applied **after compression**, on the compressed attention logits. DeCoP computes a compressed attention matrix of size N×k with k ≪ N, without ever materializing the full N×N matrix. The learnable mask is defined in the same space (N×k), so its memory and parameter growth is O(Nk) = O(S·V·k), not quadratic. Intuitively, DeCoP compresses the token-to-token attention columns into k components, and the mask gates each token’s interaction with these k components rather than with all N tokens.
>
> 4. Question:
>
> > Performance improvements on anomaly detection is marginal. SOTA performance isn’t a requirement, but the claims might need to be better reflect the actual performance.
>
> Answer:
>
> Thank you for pointing this out. We agree that the gains from anomaly detection are relatively modest and that our current phrasing may overstate the improvement. We toned down the claims to better reflect the empirical results.

---

> ### Author Response · Authors · 2026-01-13
> **Response to Reviewer F3Af [Part2]**
>
> 5. Question:
>
> > High variance across many tasks reduces confidence. Particularly as no statistical significance tests of difference are done to compare the different methods. Seed analysis is useful but it serves a different purpose.
>
> Answer:
>
> We agree that high variance can reduce confidence and that statistical testing can strengthen comparisons. In the original submission, we followed the standard evaluation setup in multivariate time-series forecasting, where methods are compared primarily through main performance tables: test errors are reported under a shared experimental setup (fixed look-back and prediction horizon), models are selected based on validation loss, and conclusions are drawn from these point estimates. Consistent with this practice, many recent baselines we compare against do not report significance tests (e.g., iTransformer, LeDDAM, PatchTST); among forecasting papers we are aware of, TimeMixer and TimeMixer++ are the only ones that explicitly include them.
>
> Following your suggestion, and to align with the comparison protocol used in TimeMixer++, we performed statistical significance tests against our two strongest baselines, LeDDAM and iTransformer, under the same evaluation protocol. Concretely, we ran both our model and each baseline 5 times with different random seeds (2021 to 2025), using the baseline hyperparameters from their official GitHub repositories. All models were evaluated across four prediction horizons [96, 192, 336, 720] for each dataset. Against LeDDAM, we observe significant gains on most datasets. On Electricity and ETTh2, the results are inconclusive, and on ETTh1, LeDDAM remains stronger. Compared with iTransformer, we also observe significant gains across most datasets. On Electricity the result is inconclusive, and on Traffic iTransformer remains stronger. Given these mixed results, we have toned down some of our claims in the revised manuscript to better reflect our findings.
>
> **Statistical comparison of XCTFormer vs LeDDAM on forecasting datasets.**
> Averaged over prediction lengths {96, 192, 336, 720} across five seeds (2021–2025).
> Confidence from Welch’s t-test (99.9%: p<0.001, 99%: p<0.01, 95%: p<0.05). **Bold** = statistically significant better performance.
>
> | Dataset | XCTFormer MSE | XCTFormer MAE | LeDDAM MSE | LeDDAM MAE | Confidence |
> |---|---:|---:|---:|---:|---:|
> | ETTh1 | 0.449 ± 0.002 | 0.436 ± 0.001 | **0.436 ± 0.0073** | **0.432 ± 0.0034** | 95% |
> | ETTh2 | 0.374 ± 0.007 | 0.399 ± 0.004 | 0.374 ± 0.0019 | 0.398 ± 0.0007 | n.s. |
> | ETTm1 | **0.371 ± 0.003** | **0.393 ± 0.002** | 0.388 ± 0.0034 | 0.398 ± 0.0023 | 99% |
> | ETTm2 | **0.271 ± 0.001** | **0.319 ± 0.001** | 0.282 ± 0.0019 | 0.326 ± 0.0009 | 99.9% |
> | Electricity | 0.176 ± 0.007 | 0.270 ± 0.007 | 0.171 ± 0.0042 | 0.264 ± 0.0027 | n.s. |
> | Traffic | **0.435 ± 0.001** | **0.287 ± 0.001** | 0.468 ± 0.0075 | 0.294 ± 0.0052 | 95% |
> | Weather | **0.237 ± 0.001** | **0.267 ± 0.001** | 0.244 ± 0.0013 | 0.273 ± 0.0012 | 99.9% |
>
> *n.s.*: Not statistically significant (p ≥ 0.10).
>
>
> **Statistical comparison of XCTFormer vs iTransformer on forecasting datasets.**
> Averaged over prediction lengths {96, 192, 336, 720} across five seeds (2021–2025).
> Confidence from Welch’s t-test (99.9%: p<0.001, 99%: p<0.01, 95%: p<0.05). **Bold** = statistically significant better performance.
>
> | Dataset | XCTFormer MSE | XCTFormer MAE | iTransformer MSE | iTransformer MAE | Confidence |
> |---|---:|---:|---:|---:|---:|
> | ETTh1 | **0.449 ± 0.002** | **0.436 ± 0.001** | 0.457 ± 0.0014 | 0.449 ± 0.0013 | 99.9% |
> | ETTh2 | **0.374 ± 0.007** | **0.399 ± 0.004** | 0.383 ± 0.0022 | 0.407 ± 0.0011 | 95% |
> | ETTm1 | **0.371 ± 0.003** | **0.393 ± 0.002** | 0.408 ± 0.0022 | 0.412 ± 0.0013 | 99.9% |
> | ETTm2 | **0.271 ± 0.001** | **0.319 ± 0.001** | 0.291 ± 0.0010 | 0.335 ± 0.0011 | 99.9% |
> | Electricity | 0.176 ± 0.007 | 0.270 ± 0.007 | 0.176 ± 0.0037 | 0.267 ± 0.0026 | n.s. |
> | Traffic | 0.435 ± 0.001 | 0.287 ± 0.001 | **0.430 ± 0.0010** | **0.283 ± 0.0010** | 99.9% |
> | Weather | **0.237 ± 0.001** | **0.267 ± 0.001** | 0.260 ± 0.0014 | 0.281 ± 0.0016 | 99.9% |
>
> *n.s.*: Not statistically significant (p ≥ 0.10).

---

> ### Author Response · Authors · 2026-01-13
> **Response to Reviewer F3Af [Part3]**
>
> 6. Question:
>
> > No sensitivity analysis for patch hparams. This seems like a very important hyper parameter but there is no indication of its effect on the performance or how this should be chosen for a given task. In the experiments some use length of 16 with stride 8 while others use 64 / 32. It’s not clear how or why these were chosen and what effect it has.
>
>
> Answer:
>
> PatchTST conducted an explicit patch-length sensitivity analysis (“Varying Patch Length”) by fixing the lookback at L=336 and varying the patch length P, and reports that the forecasting error (MSE) does not vary significantly across patch sizes, indicating that performance is relatively robust to this hyperparameter. Following PatchTST’s best practice, we kept the patching configuration fixed in most of our experiments. For forecasting and anomaly detection, we used patch_len = 16 and stride = 8 throughout. For imputation, where the lookback was substantially larger (L = 1024 versus 96 in forecasting and 100 in anomaly detection), we tested a small set of patch_len values {16, 64, 128} with the corresponding stride = patch_len/2. Based on the validation error in our hyperparameter search, we selected patch_len = 64 and stride = 32 for some imputation datasets.
>
> We conducted an additional patch--length sensitivity experiment on our method and observed results similar to those reported in PatchTST. We fixed the stride to patch_len/2 and tested different patch_len values from 8 to 96 in increments of 8. We report the average validation and test losses for ETTm1 and Weather across forecasting horizons {96, 192, 336, 720}. Overall, performance varies only slightly across patch sizes, indicating that our method is robust to this hyperparameter. The effect is small on ETTm1 and somewhat more noticeable on Weather, but the differences remain limited to roughly 2 to 4 percent, suggesting that the preferred patching configuration can be dataset dependent without requiring precise tuning. The full analysis is provided in Appendix C.3 of the revised manuscript.
>
> 7. Question:
>
> > The paper is completely lacking a discussion or limitations section. The conclusion is very positive and does not mention any of the caveats of the approach. I would expect an honest limitations section that overview the key limitations that the authors have identified, I have mentioned some of them above. A discussion on some of the results (e.g. high variance) would be an improvement as well.
>
> Answer:
>
> We appreciate the reviewer raising this point, and we agree that a limitations and discussion section is important for presenting a balanced and transparent view of our method. In the revised manuscript, we added a dedicated limitations section in the main paper and updated the conclusion section accordingly.
>
> Summarizing the limitations section: XCTFormer explicitly models all pairwise time-channel dependencies via unified attention, which can improve performance but also introduces practical limitations. Flattening time-channel tokens makes attention quadratic in the number of tokens, increasing memory and runtime as the lookback length and channel count grow. DeCoP mitigates this cost by compressing attention to a linear form, but it still scales with sequence length and dimensionality and adds a small parameter overhead. Finally, the gains are not uniform across datasets and tasks. Some settings show smaller improvements or higher variance across runs. This suggests that the presented pairwise modeling strategy is sensitive to the underlying dependency structure and may offer limited benefits when cross-channel relations are weak or difficult to capture.
>
> 8. Question:
>
> > Not really a weakness but a mention or discussion of TS foundation models would be apt. Although they are a different paradigm, they are increasingly relevant in time series modelling.
>
>
> Answer:
>
> We added a short discussion to better position our work on channel-independent and channel-dependent methods in the context of time-series foundation models. This discussion is included in the Related Work section of the revised manuscript.
>
> ----------------------------------------------------------------------------------------------------------
>
> **Lack of support claims**
> We reviewed the manuscript and toned down several statements about forecasting and anomaly detection to better align with the presented results and analysis. We also revised overly strong claims. For example, we replaced the statement that XCTFormer models “all possible…” cross-channel and cross-temporal relationships with the more precise “all pairwise…” phrasing.

---

> ### Author Response · Authors · 2026-01-13
> **Response to Reviewer F3Af [Part4]**
>
> **Critical**:
>
> 1. Question:
>
> > Validate information loss in two-stage methods (or adjust claims)
>
> Answer:
>
> We adjusted claims in the revised paper.
>
> 2. Question:
>
> > Perform statistical significance tests to check for difference (or adjust claims)
>
> Answer:
>
> We performed statistical significance tests against the two strongest forecasting baselines and report the results in Appendix D.1. Based on these findings, we also toned down several claims in the revised paper to better reflect the observed performance.
>
> 3. Question:
>
> > Check negative weights claims with more appropriate experimental set up
>
> Answer:
>
> We included extended analysis in Appendix C.4 of the revised paper, showing that weights are normally distributed with approximately half of the values negative. We also performed an ablation study where we kept the proposed activation function with negative weights clipped, demonstrating that our original XCTFormer consistently matches or improves over the tested variant with an average modest gain of 1.3%.
>
> 4. Question:
>
> > Add a discussion of limitation
>
> Answer:
>
> We added a Limitations section at the end of the revised paper before Conclusions.
>
> **Less Critical**:
>
> 1. Question:
>
> > Add sensitivity analysis of patch and stride hyperparams
>
> Answer:
>
> We included analysis of patch and stride hyperparameters in Appendix C.3 of the revised paper, showing that these hyperparameters can be treated as dataset-specific and that our model is overall robust to these changes, with performance varying only 2–4%.
>
> 2. Question:
>
> > Scalability analysis
>
> Answer:
>
> XCTFormer is sensitive to input size because it explicitly models all pairwise time and channel relationships, leading to quadratic scaling. DeCoP mitigates this issue by compressing per-token dependencies, achieving approximately linear scaling. To verify this empirically, we conducted a scalability analysis to measure how the parameter count changes with input size. Since effective input scales with both data dimensionality (number of channels) and sequence length, we tested two settings: varying channel count with fixed sequence length, and varying sequence length with fixed channel count. The results confirm quadratic growth for XCTFormer, while XCTFormer with the DeCoP plugin exhibits approximately linear growth. The full scalability analysis is provided in Appendix C.5 of the revised paper. We are happy to include additional scalability analysis if needed.
>
> 3. Question:
>
> > Adjust the “all possible relationships” claim
>
> Answer: We adjusted this claim in the appropriate places in the revised version.
>
> 4. Question:
>
> > Compare the computational cost and scalability of the methods
>
> Answer:
>
> Please see our response regarding this concern to Reviewer 1.

---

### Comment · Action_Editor_CJdP · 2026-01-13
**Please engage in Author-Reviewer Discussions**

Hi everyone,

We're happy that we now have a set of responses + a revision to this paper from the authors. Reviewers, please continue the discussion with the authors and try to resolve any misunderstandings or points of clarification that may still be needed.

I know that you've been prompted to submit a recommendation on this paper, given the time since all three reviews were submitted. However, I do ask that you sufficiently engage with the authors prior to submitting your recommendations to me. I will not accept score recommendations without an honest effort to discuss with the authors.

Best,
Taylor

---

### Decision · Action_Editor_CJdP · 2026-02-13

**Recommendation:** Accept as is

**Audience:**

Yes

**Audience Explanation:**

The ideas in this paper are useful for machine learning practitioners working in the area of time-series-forecasting/classification.

**Claims And Evidence:**

Yes

**Claims Explanation:**

The paper has been substantially improved through the review process, primarily focused on addressing the reviewers concerns about the empirical evidence substantiating the claims that have been made. Initially there were mismatched or incomplete evidence to support the claims. This has been fixed, new experiments have been done and missing details added along with a discussion of limitations and further benchmarking.